# Enhancer adoption by an LTR retrotransposon generates viral-like particles, causing developmental limb phenotypes

Juliane Glaser [1] ✉, Giulia Cova [1,7], Beatrix Fauler[2], Cesar A. Prada-Medina[1], Virginie Stanislas [3], Mai H. Q. Phan[1], Robert Schöpflin[1,3], Yasmin Aktas[1], Martin Franke [1,8], Guillaume Andrey [1,9,10], Natalia Bartzoka[1,4], Christina Paliou[1,8], Verena Laupert[3], Wing-Lee Chan[4], Lars Wittler[5], Thorsten Mielke [2] & Stefan Mundlos [1,4,6] ✉

Transposable elements (TEs) are scattered across mammalian genomes. Silencing of TEs prevents harmful effects caused by either global activation leading to genome instability or insertional mutations disturbing gene transcription. However, whether the activation of a TE can cause disease without directly affecting gene expression is largely unknown. Here we show that a TE insertion can adopt nearby regulatory activity, resulting in the production of cell-type-specific viral-like particles (VLPs) that affect embryo formation. Failure to silence an LTR retrotransposon inserted upstream of the *Fgf8* gene results in their co-expression during mouse development. VLP assembly in the *Fgf8*-expressing cells of the developing limb triggers apoptotic cell death, resulting in a limb malformation resembling human ectrodactyly. The phenotype can be rescued by mutating the retrotransposon coding sequence, thus preventing its full endogenous retroviral cycle. Our findings illustrate that TE insertions can be incorporated into the local genomic regulatory landscape and that VLP production in post-implantation embryos can cause developmental defects.

About half of the mammalian genome consists of TEs[1]. These genetic elements can mobilize in the genome, causing insertional mutations and genomic instability[2]. To prevent such harmful effects, TEs are transcriptionally silenced by DNA methylation and other epigenetic marks in most somatic cells[3]. TE expression can be triggered by remodeling of the epigenome occurring in physiological and pathological contexts. During mammalian pre-implantation development, activation of several TE families was shown to participate in establishing pluripotency and embryonic genome activation required for development[4–6]. In cancer, neurological disorders or aging, aberrant TE activation may

have a role in disease progression[7–10]. Yet very little is known about how TE activation affects post-implantation development. Although this critical period of cellular differentiation and organ formation is not subject to global epigenetic changes, failure to establish epigenetic marks at specific TE sequences could occur.

Precise transcription during organogenesis is influenced by promoter–enhancer communication and the three-dimensional (3D) organization of the genome[11]. TEs can contain transcription factor binding sites, act as alternative promoters or enhancers and modify 3D chromatin architecture[12]. TE insertion in a gene can also affect

normal transcription processes (that is, splicing, polyadenylation and disruption of an exon or a regulatory element)[13]. However, how regulatory landscapes influence TE transcription is unknown. Here, we show that an unmethylated LTR retrotransposon insertion does not impair transcriptional activation but responds to the regulatory elements surrounding its insertion, according to the organization of the genome into topologically associating domains (TADs). Based on this result, we postulated that enhancer adoption by LTR elements could be a more common phenomenon, and we validated this concept at three developmental loci.

We focused on an endogenous retrovirus (ERV) element from the MusD (type D-related mouse provirus-like) family, probably originating from an infectious simian type D retrovirus[14]. MusD are evolutionarily young elements with about 100 full-length copies in the mouse genome and are polymorphic between mouse strains[15–17]. As with many TEs, their expression is high in the inner cell mass of stage E3.5 mouse embryos, but MusD elements are also active in post-implantation embryos from E7.5 to E13.5 in specific tissues[18,19], making them a good model to study the role of TEs during organogenesis. Using the example of the mouse dactylaplasia limb malformation, we show that enhancer adoption leads to the expression of a MusD element in the apical ectodermal ridge (AER) of the developing limb. This element assembles into VLPs, which cause the limb phenotype by triggering premature cell death of AER cells.

## Results

### Unmethylated MusD inserted at the *Fgf8* locus causes dactylaplasia

Dactylaplasia is a limb malformation caused by a spontaneous allele, named *Dac1J*, on mouse chromosome 19 (refs. [20,21]). The *Dac1J* allele was identified as a MusD LTR retrotransposon, inserted into the intergenic region between the *Fbwx4* and *Fgf8* genes[22] (Fig. 1a,b). The 7.4 kb-long MusD-*Dac1J* sequence contains a retroviral sequence with *gag*, *pro* and *pol* genes flanked by identical 5′ and 3′ LTR[22] but, like other MusD elements, lacks an *env* gene[14] (Fig. 1a). Dactylaplasia mutants show a severe ectrodactyly-type limb phenotype characterized by missing central digits in both forepaws and hindpaws and abnormal nail-like structures (Extended Data Fig. 1a,b). The phenotype is fully penetrant in homozygotes (Extended Data Fig. 1c), whereas heterozygote mutants show a variable phenotype with 76.2% (32 out of 42) and 85.7% (36 out of 42) affected forelimbs and hindlimbs, respectively (Extended Data Fig. 1d). Moreover, the phenotype is dependent on the presence of an epigenetic modifier (*Mdac*) that is polymorphic between mouse strains and has been refined to a region on mouse chromosome 13 (ref. [22]). Accordingly, the MusD-*Dac1J* insertion leads to dactylaplasia in 129sv mice but not in C57BL/6 mice (Fig. 1b,c), correlating with differential CpG DNA methylation on the 5′LTR promoter (Fig. 1d). The *Dac1J*-5′LTR methylation is constitutive among different germ layers of the embryo and in extra-embryonic tissues (Extended Data Fig. 1e) but does not affect methylation upstream and downstream the insertion (Fig. 1d). For simplicity, '*Dac1J*' and '*Dac1J*-BL6' will refer to mutant animals with the *Dac1J* insertion in the 129sv (exhibiting a phenotype) and the C57BL/6 (no phenotype) backgrounds, respectively.

To functionally prove that the dactylaplasia was caused by the MusD-*Dac1J* insertion at the *Fgf8* locus, we used CRISPR–Cas9 to delete it from the *Dac1J*/*Dac1J* line (Fig. 1e). E18.5 embryos with a deletion of MusD-*Dac1J* (ΔDac1J/ΔDac1J) show a complete rescue of the phenotype (Fig. 1f), confirming the causative nature of the LTR insertion. The deletion of the *Dac1J* did not affect the local CpG DNA methylation (Fig. 1g), and epigenetic profiling using publicly available data[23–25] suggests that neither the *Dac1J* insertion nor our deletion overlaps with an open chromatin region (Extended Data Fig. 1f).

### *Fgf8*-expressing cells are affected in *Dac1J* embryos

The locus contains six genes: *Btrc*, *Poll*, *Dpcd* and *Fbxw4* are ubiquitously and lowly expressed during limb development, whereas *Lbx1* and *Fgf8* are specifically expressed in the muscle progenitors and AER of the developing limbs, respectively. To elucidate whether the MusD insertion affects local gene activation, we performed bulk RNA sequencing (RNA-seq) from forelimbs at early E11.5, a stage in which the activation of all genes at the locus should be completed. Except for *Fgf8*, local gene expression is not affected at that stage (Extended Data Fig. 2a). *Fgf8* expression in AER of the developing limb ranges from E9.5 to E13.5, where it controls growth and patterning[26]. Whole-mount in situ hybridization confirmed *Fgf8* expression is affected at E11.5 in *Dac1J* embryos, but showed no difference with wild-type at E10.5 (Extended Data Fig. 2b).

To decipher how transcription is affected over time, we performed single-cell RNA-seq (scRNA-seq) from wild-type and *Dac1J*/*Dac1J* limb buds at E9.5, E10.5 and E11.5 (Fig. 2a). We annotated seven cell types (Fig. 2b, Extended Data Fig. 3a and Supplementary Table 1) that were present at similar proportions between wild-type and *Dac1J*/*Dac1J* at E9.5 and E10.5 (Fig. 2c and Extended Data Fig. 3b). However, at E11.5, a 82.6% loss of AER and dorso-ventral ectoderm cells was detected (Fig. 2c and Supplementary Note 1). We detected *Fgf8* expression specifically in the AER and dorso-ventral ectoderm cells and show that it was not affected at E9.5 and only mildly affected at E10.5 in *Dac1J*/*Dac1J* limbs compared to wild-type (Fig. 2d and Extended Data Fig. 3c). At E11.5, the very few AER cells in the *Dac1J*/*Dac1J* expressed *Fgf8* at a low level (Extended Data Figs. 2b, 3c and 4a). To address whether the observed *Fgf8* expression in *Dac1J*/*Dac1J* mice was directly caused by transcriptional changes or by the observed depletion of AER cells, we examined the expression of other AER genes. These genes show the same expression pattern as *Fgf8* with a proper activation at E9.5, a mild or no decrease at E10.5 and a decrease at E11.5 when the AER cells are lost (Extended Data Fig. 4b). Overall, our results indicate that AER cells rather than gene activation are affected by the MusD-*Dac1J* insertion.

### Co-expression of MusD-*Dac1J* and *Fgf8* in the developing limb

We confirmed that the activation and cell-type-specific expression of the other five genes at the locus was not affected in *Dac1J*/*Dac1J* compared to wild-type (Extended Data Figs. 3c and 4c and Supplementary Note 2). However, when scRNA-seq reads were mapped to a transcriptome containing the *Dac1J* sequence, we observed MusD-*Dac1J* expression specifically in *Dac1J*/*Dac1J* AER cells (Fig. 2e). MusD-*Dac1J* transcripts were detected at E9.5 and E10.5 but not at E11.5 when AER cells vanished (Fig. 2e and Fig. Extended Data Fig. 4a). Remarkably, nearly all AER cells expressing MusD-*Dac1J* co-expressed *Fgf8* (Fig. 2f), showing MusD-*Dac1J* transcription in the same pattern as *Fgf8*. These results indicate that MusD-*Dac1J* is co-expressed with *Fgf8* in the AER of the developing limb at E9.5 and E10.5, followed by a disappearance of most of the AER cells at E11.5.

### Active MusD-*Dac1J* leads to an ectopic chromatin loop

We next asked whether MusD-*Dac1J* co-expression with *Fgf8* is linked to changes at the 3D genome architecture level. The locus consists of two TADs[27] (Fig. 3a). TADs are self-interacting blocks containing genes with their regulatory elements and are insulated from each other by boundaries defined by the binding of cohesin and CTCF, or by active transcription[28–30]. MusD-*Dac1J* is located in the 175 kb TAD containing *Fbxw4* and *Fgf8* as well as previously characterized *Fgf8* enhancers (blue bars in Fig. 3a) driving its expression in the AER, the hindbrain–midbrain boundary, the branchial arches and the tailbud[27,31]. Recent data suggest that 3D genome organization can be shaped by retrotransposon-derived CTCF sites or their active transcription[32,33]. We subjected the MusD-*Dac1J* sequence to CTCF motif scanning and identified several binding sites (Fig. 3b and Extended Data Fig. 5a). To investigate whether MusD-*Dac1J* affects the 3D chromatin architecture at the locus, we performed capture-Hi-C (cHi-C) and CTCF chromatin immunoprecipitation with sequencing (ChIP–seq) from *Dac1J*, *Dac1J*-BL6 and wild-type embryonic limbs.

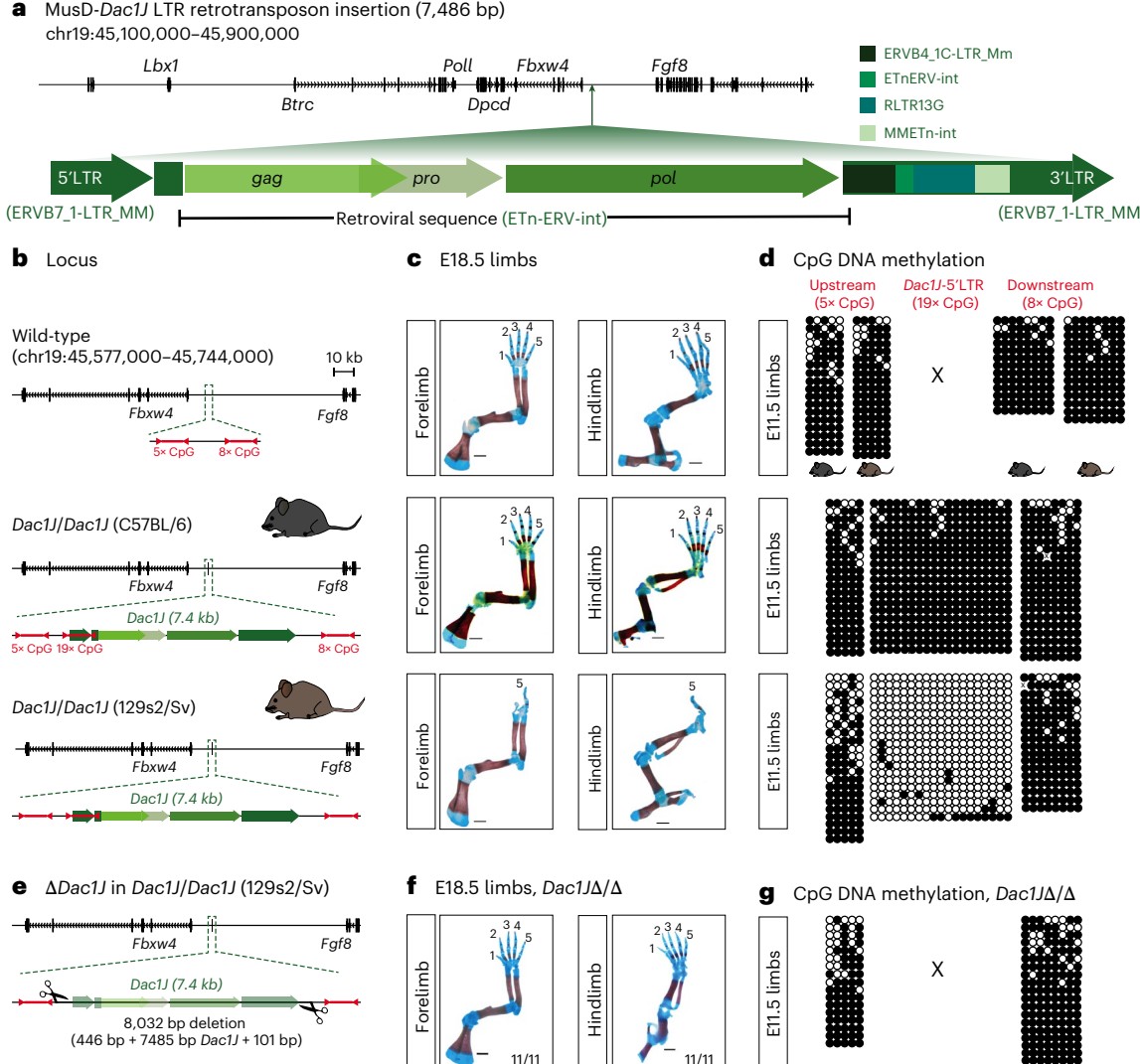

**Fig. 1 | Dactylaplasia limb malformation is caused by an unmethylated MusD insertion at the *Fgf8* locus. a**, Scheme of the 7,486 bp MusD-*Dac1J* inserted at the *Fgf8* locus (mm10, chr19:45,100,000–45,900,000). The full-length element contains retroviral (*gag*, *pro*, *pol*) and non-retroviral sequences flanked by 5′LTR and 3′LTR. RepeatMasker names are indicated in green. **b**, The mouse *Fbxw4*–*Fgf8* locus is shown in wild-type mice (top) or with the intergenic MusD-*Dac1J* insertion in *Dac1J*/*Dac1J* C57BL/6 (middle) or *Dac1J*/*Dac1J* 129s2/Sv (bottom). The number and localization of the CpG measured in **d** are indicated in red. **c**, Skeletal analysis of the forelimbs and hindlimbs of E18.5 wild-type (top), *Dac1J*/*Dac1J* C57BL/6 (middle) or 129s2/Sv (bottom) mice, stained with alcian blue (cartilage)

and alizarin red (bone). Scale bars, 1 mm. **d**, DNA methylation status of five CpGs upstream of the insertion, 19 CpGs in the 5′LTR (promoter) of *Dac1J* and eight CpGs downstream of the insertion in wild-type (C57BL/6 and 129s2/Sv, top), *Dac1J*/*Dac1J* C57BL/6 (middle) or 129s2/Sv (bottom) mice measured by bisulfite cloning and sequencing from E11.5 limbs. White circles, unmethylated CpGs; black circles, methylated CpGs. **e**, Schematic representation of the CRISPR–Cas9 *Dac1J* deletion in the *Dac1J*/*Dac1J* (129s2/Sv) line. **f**, Skeletal analysis of E18.5 *Dac1J* Δ/Δ (129s2/Sv) forelimbs and hindlimbs showing complete rescue. Scale bars, 1 mm. *n* = 11 out of 11 E18.5 showed a similar phenotype. **g**, CpG DNA methylation status in *Dac1J* Δ/Δ (129s2/Sv) E11.5 limbs as in **d**.

In the *Dac1J* mutants, the TAD structure was maintained, but an ectopic chromatin loop was formed at the MusD-*Dac1J* insertion site (Fig. 3c). Subtracting the cHi-C signal between mutants and wild-type showed gain of contact where the ectopic loop forms (Fig. 3d, red) and increased insulation within the *Fgf8* TAD in *Dac1J* mutants (Extended Data Fig. 5b, blue). The *Dac1J*-BL6 embryos did not show any change in chromatin conformation (Extended Data Fig. 5c). ChIP–seq from embryonic limbs suggests that binding of CTCF has a minor role in the formation of the ectopic chromatin loop, as weak ectopic binding in *Dac1J* mutants was observed strictly at E10.5 (Extended Data Fig. 5d). We next asked whether the ectopic chromatin loop formed by hypomethylated MusD-*Dac1J* exist in a tissue in which the region is inactive. cHi-C from *Dac1J* embryonic hearts, in which MusD-*Dac1J* 5′LTR is hypomethylated (Extended Data Fig. 5e), did not detect any ectopic chromatin loop (Fig. 3e,f).

Finally, we tested whether the MusD-*Dac1J* 5′LTR promoter (containing one CTCF site; Fig. 3b) would be sufficient to drive the ectopic chromatin loop. We generated a knock-in at the *Fgf8* locus with the MusD-*Dac1J* 5′LTR but replaced the retroviral sequence with a LacZ reporter gene (5LTR-LacZ-KI; Fig. 3b). When performing cHi-C from this mutant's embryonic limbs, we observed a similar ectopic loop as in the *Dac1J* embryos (Fig. 3g,h and Supplementary Note 3) with an unmethylated 5′LTR promoter (Extended Data Fig. 5e). This shows that the 5′LTR promoter is sufficient to recapitulate the main 3D conformation changes mediated by MusD-*Dac1J*, even when not bound by CTCF (Extended Data Fig. 5d). However, this ectopic contact without the full-length MusD-*Dac1J* retroviral sequence did not lead to any phenotype at E18.5 (Extended Data Fig. 5f). Overall, our results show that the active MusD-*Dac1J* promoter creates an ectopic chromatin loop that does not impact local gene activation or cause a phenotype.

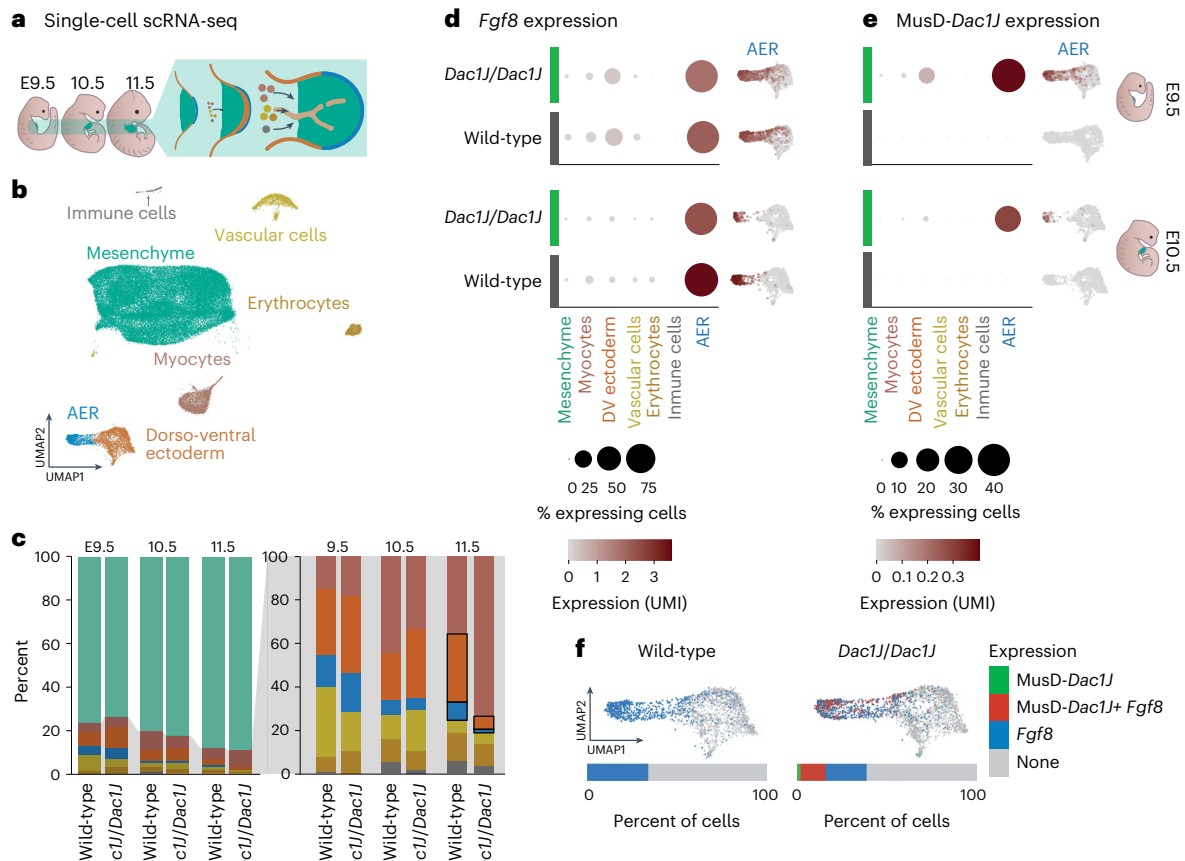

**Fig. 2 | MusD-*Dac1J* is expressed in *Fgf8*-expressing cells before their disappearance. a**, Schematic representation of the embryos used for limb bud scRNA-seq. **b**, Uniform manifold approximation and projection (UMAP) showing seven cell clusters identified through scRNA-seq of E9.5, E10.5 and E11.5 stage mouse forelimbs from wild-type and *Dac1J/Dac1J* embryos. **c**, Proportion of cell types within the forelimb population in E9.5, E10.5 and E11.5 wild-type and *Dac1J/Dac1J* embryos. The seven colors correspond to the seven cell types as represented in **b**. Gray shading represents a zoomed-in view without mesenchyme cells. It shows a strong decrease of AER and dorso-ventral cells in Dac1J/Dac1J at E11.5 compared to wild-type, as represented with black outlines.

Zoomed-in view without mesenchyme cells shows a strong decrease of AER and dorso-ventral (DV) cells in *Dac1J/Dac1J* embryos at E11.5 compared to wild-type. **d,e**, Dot plots showing expression and percentage of expressing cells for *Fgf8* (**d**) and MusD-*Dac1J* (**e**) in the seven forelimb cell cluster at E9.5 and E10.5 in wild-type (dark gray) and *Dac1J/Dac1J* (green). Unique molecular identifier (UMI) expression in the AER and dorso-ventral ectoderm cell type is also represented as a feature plot next to the respective dot plot for each genotype and stage. **f**, Wild-type (left) and *Dac1J/Dac1J* (right) AER and dorso-ventral cells colored depending on their status of expressing *Fgf8* and *Dac1J*. Percent of cells expressing either MusD-*Dac1J*, MusD-*Dac1J*+*Fgf8*, *Fgf8* or none of them is indicated.

## MusD-*Dac1J* adopts the regulatory activity of the *Fgf8* TAD

Our data suggest that the ectopic chromatin loop in *Dac1J* embryos is the consequence rather than the cause of active MusD-*Dac1J* transcription. Therefore, we wondered what drives MusD-*Dac1J* co-expression with *Fgf8*. We performed 4C-sequencing (4C-seq) with the MusD-*Dac1J* sequence as a viewpoint and detected more interaction within the region containing *Fgf8* enhancers in the *Dac1J* mutant than in the *Dac1J*-BL6 control, suggesting MusD-*Dac1J* 5′LTR compatibility with *Fgf8* enhancers (Fig. 3i). We took advantage of the 5LTR-LacZ-KI and generated a similar knock-in in which the LacZ reporter gene is driven by a β-globin minimal promoter instead of the 5′LTR (Fig. 3j). β-galactosidase staining in E11.5 embryos with either construct inserted in the *Fgf8* TAD both faithfully recapitulated the native *Fgf8* expression (Fig. 3j). This indicates that the MusD-*Dac1J* 5′LTR promoter acts as a sensor, adopting the regulatory information within the *Fgf8* TAD.

## Assembly of VLPs in *Dac1J* AER cells

Having shown that MusD-*Dac1J* uses the *Fgf8* regulatory landscape for its transcription, we asked how this is linked to the lack of AER at E11.5. *Dac1J* is a full-length MusD element with intact open reading frames for *gag*, *pro* and *pol* (Fig. 1a), almost identical to three autonomous

MusD elements[15] (Extended Data Fig. 6a and Supplementary Note 4). This prompted us to examine whether MusD-*Dac1J* produces retroviral proteins using a well-characterized Gag-MusD polyclonal antibody[15] for whole-embryo immunofluorescence assay. We detected Gag-MusD in all analyzed *Dac1J/Dac1J* embryos but no staining in the *Dac1J*-BL6 controls (Fig. 4a and Extended Data Fig. 6b–d). Cytoplasmic Gag-MusD was observed in tissues expressing *Fgf8*, such as the midbrain–hindbrain boundary, the branchial arches and the AER of both forelimbs and hindlimbs at E10.5, but not in the tailbud (Fig. 4a and Extended Data Fig. 6c). Gag-MusD persisted in the AER of *Dac1J* E11.0 embryos (Extended Data Fig. 6d). We thus analyzed the presence of VLPs in the AER cells. Remarkably, transmission electron microscopy (TEM) revealed the presence of cytoplasmic electron-dense particles of 70–90 nm, the reported size for MusD VLPs[34] (Fig. 4b). VLPs were detected both in clusters and as single particles nearby the nuclear membrane and in the cisternae of the endoplasmic reticulum (Fig. 4b). The presence of MusD-derived particles in the AER of *Dac1J* embryos was supported by immuno-gold TEM staining, which detected VLPs at the nuclear membrane labeled by Gag-MusD antibodies in several AER cells (Fig. 4c and Extended Data Fig. 6e–g). MusD-*Dac1J* VLPs were strictly intracellular and did not bud, as expected from an ERV

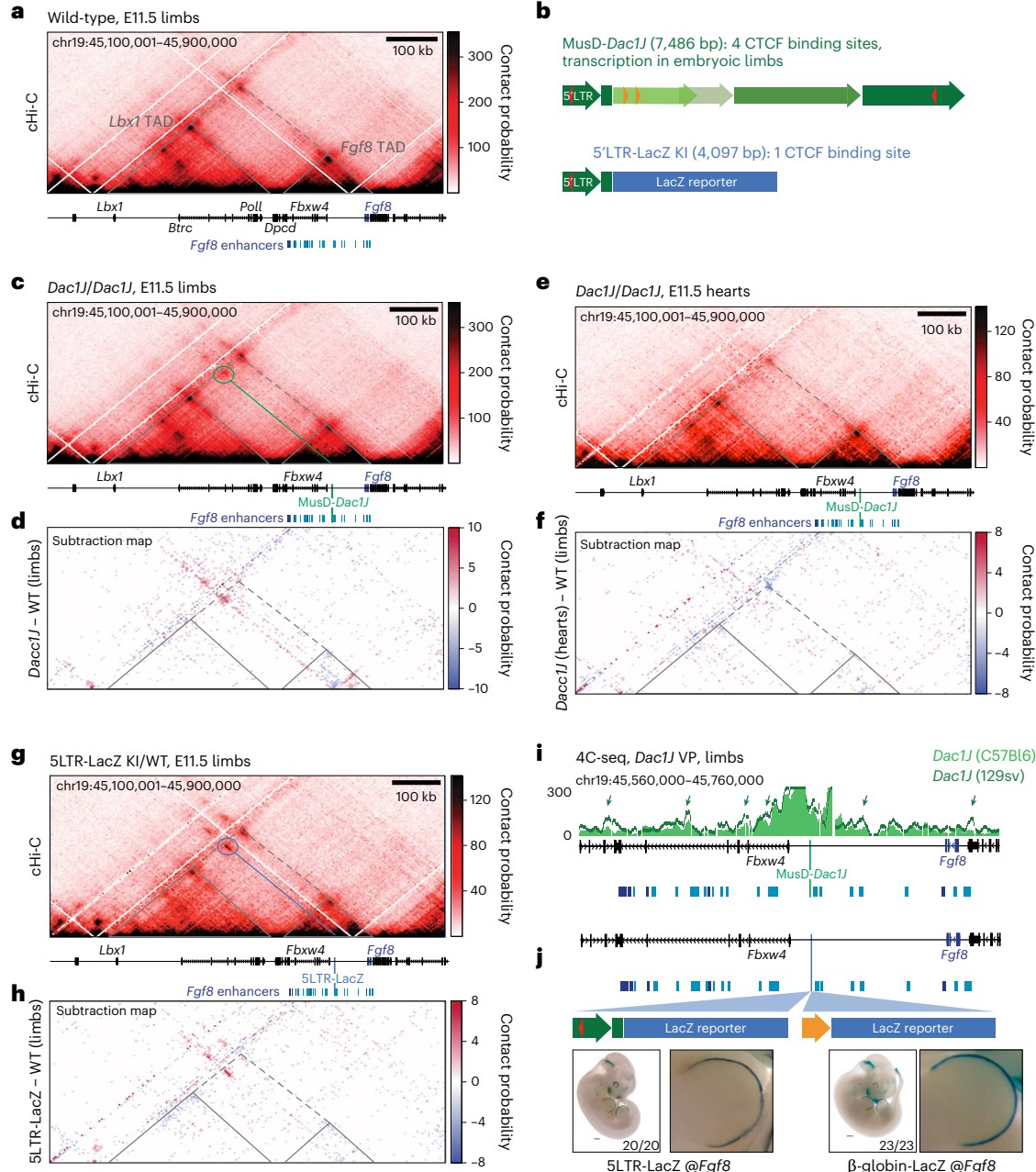

**Fig. 3 | MusD-*Dac1J* insertion leads to an ectopic chromatin loop. a**, cHi-C of the *Lbx1/Fgf8* locus (mm10, chr19:45,100,000–45,900,000) from wild-type E11.5 mouse limb buds. Data are shown as merged signals of *n* = 3 biological replicates and *n* = 1 technical replicate. The dashed lines indicate the *Lbx1* and *Fgf8* TADs. Published *Fgf8* enhancers are indicated in light blue and dark blue (AER enhancers). Color bar shows the KR-normalized contact probability (white: low; black: high). **b**, Schematic representation of the MusD-*DacJ* and the 5′LTR-LacZ with the number of CTCF sites with *P* < 10⁻⁴ (calculated using FIMO, based on a dynamic programming algorithm to convert log-odds scores into *P* values) indicated as orange (sense) and red (antisense) triangles. KI, knock-in. **c–h**, cHi-C of the *Lbx1/Fgf8* locus (mm10, chr19:45,100,000–45,900,000) from *Dac1J/Dac1J* E11.5 mouse limb buds (**c,d**), *Dac1J/Dac1J* E11.5 mouse embryonic hearts (**e,f**) and *Dac1J*-5LTR-LacZ E11.5 limb buds (**g,h**). Data show the cHi-C as merged signals of *n* = 2 biological replicates and *n* = 2 technical replicates (**c,d**), *n* = 2 biological replicates (**e,f**) and *n* = 3 biological replicates (**g,h**). Subtraction maps between mutants and wild-type (**d, f, h**) show gain (red) and loss (blue) of interaction in the mutant compared to the wild-type. The position of the MusD-*Dac1J* insertion is indicated in green. Scale bars in **a**, **c**, **e** and **g** show the KR-normalized contact probability (white: low; black: high). Scales bars in **d**, **f** and **h** show the *z*-scaled differential contact probability (red: gain; blue: loss). **i**, 4C-seq with a viewpoint (VP) on *Dac1J* showing *Dac1J/Dac1J* (129s2/Sv) and *Dac1J/Dac1J* (C57BL/6). Green arrows indicate a difference in 4C enrichments. In vivo-confirmed *Fgf8* enhancers are indicated in light blue and dark blue (specific AER enhancers), and the MusD-*Dac1J* insertion point is indicated in green. **j**, Representation of the *Dac1J*-5LTR-LacZ and β-globin-LacZ knock-ins at the *Fgf8* locus and their β-galactosidase staining on E11.5 embryos and forelimbs, zoomed in. Scale bars, 500 μm; *n* = 20 out of 20 and 23 out of 23 embryos showed similar staining.

lacking the *env* gene³⁴. Staining from wild-type embryos showed no signal (Extended Data Fig. 6h). This demonstrates the presence of retroviral proteins and particles in the AER cells of the developing *Dac1J* mutant embryos.

**MusD VLPs in the AER are associated with cell death**

To decipher how the presence of MusD VLPs during limb development is linked to the disappearance of AER cells and the phenotype, we examined TEM images of wild-type and *Dac1J* developing limbs at

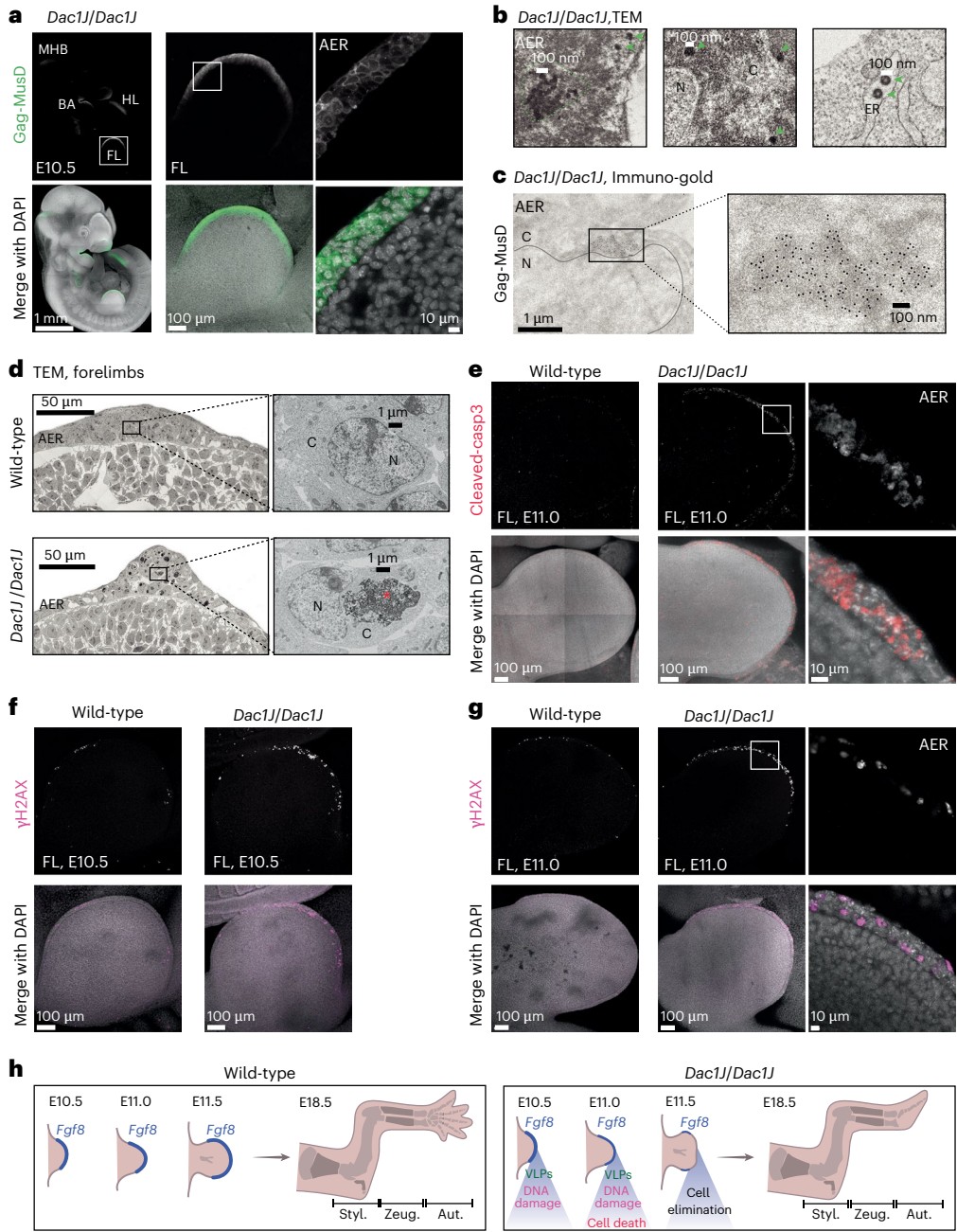

**Fig. 4 | Presence of MusD retroviral-like particles is associated with AER cell death. a**, Anti-Gag-MusD whole-mount immunofluorescence on E10.5 *Dac1J/Dac1J* embryos showing embryo (left), forelimb (middle) and AER (right). *n* = 5 out of 5 biological replicates were confirmed. FL, forelimb; HL, hindlimb; BA, branchial arches; MHB, midbrain–hindbrain boundary. **b**, TEM analysis on E11.0 *Dac1J/Dac1J* AER cells. Three different cells are shown. Green arrowheads and dotted lines indicate single and aggregated VLPs, respectively. *n* = 5 biological and *n* = 2 technical replicates. C, cytoplasm; N, nucleus; ER, endoplasmic reticulum. **c**, TEM analysis after immuno-gold labeling with anti-Gag-MusD on E11.0 *Dac1J/Dac1J* AER cell shows cytoplasmic aggregates of GAG. *n* = 2 biological replicates and *n* = 6 technical replicates were confirmed. **d**, TEM analysis on E11.0 wild-type and *Dac1J/Dac1J* forelimbs. Zoomed-in view (right) shows the structure of an AER cell. Red asterisk indicates an apoptotic body.

**e**, Anti-cleaved-Caspase-3 whole-mount immunofluorescence on E11.0 wild-type and *Dac1J/Dac1J* forelimb and AER. *n* = 4 out of 4 (wild-type) and *n* = 6 out of 6 (*Dac1J/Dac1J*) biological replicates were confirmed. **f,g**, Anti-γH2AX whole-mount immunofluorescence on wild-type and *Dac1J/Dac1J* E10.5 forelimbs (**f**) and E11.0 forelimbs and AER (**g**). *n* = 3 out of 3 biological replicates were confirmed. **h**, Schematic representation of wild-type and *Dac1J/Dac1J* limbs at E10.5, E11.0, E11.5 and E18.5. Although the AER of the limb bud is not morphologically affected in the *Dac1J/Dac1J* at E10.5, VLPs are expressed, and we detected increased DNA damage. Half a developmental day later, more DNA damage and apoptosis occur. By E11.5, most of the AER cells have been eliminated by cell death. Consequently, at E18.5, the autopod of the *Dac1J/Dac1J* embryos is severely affected by a lack of metacarpal structures, but the stylopod and zeugopod have developed similarly to the wild-type embryos. Styl., Stylopod; Zeug., Zeugopod; Aut., Autopod.

E11.0. Numerous *Dac1J* AER cells contain up to four (or occasionally more) apoptotic bodies and phagocytes (Fig. 4d and Extended Data Fig. 6i,j). Apoptotic cell death was confirmed by immuno-staining with a cleaved-Caspase-3 antibody, which shows Caspase-3 in its active

form in AER cells of both forelimbs and hindlimbs and lower signal in a few cells of the branchial arches but no staining in wild-type embryos (Fig. 4e and Extended Data Fig. 7a). We did not detect cleaved-Caspase-3 at E10.5, but γH2AX staining revealed elevated levels of DNA damage

in the AER compared to wild-type (Fig. 4f). This was even more pronounced at E11.0, concomitant to the activation of Caspase-3 (Fig. 4g and Extended Data Fig. 7b). Fluorescence in situ hybridization staining of the developing limb bud using an *Fgf8* mRNA probe supported proper AER morphology at E10.0 and E10.5 in *Dac1J* embryos (Extended Data Fig. 7c,d) despite few cells being possibly affected by the increased DNA damage (Fig. 4f). Typical apoptosis characteristics were observed in AER cells, such as blebbing and nuclear fragmentation, beginning at E11.0 and exacerbated at E11.5 (Extended Data Fig. 7e,f). According to our transcriptomic data, *Fgf8* expression globally decreased after E10.5 owing to AER cells undergoing DNA damage and apoptosis (Extended Data Fig. 7c–f). Early E11.5 *Dac1J* forelimbs showed a significant difference for apoptosis-related and limb patterning genes (Extended Data Fig. 8a,b). However, we did not detect activation of the interferon pathway (Extended Data Fig. 8c and Supplementary Table 2), suggesting that *env*-less MusD VLPs are not infectious. Overall, our results show that the presence of VLPs in the AER is toxic, leading to DNA damage and non-physiological apoptotic cell death during development (Fig. 4h). This reinforces our finding that dactylaplasia is caused by a loss of AER cells between E10.5 and E11.5 rather than the misregulation of *Fgf8*.

### Lack of MusD-*Dac1J* reverse transcription partially rescues dactylaplasia

As an autonomous ERV element, MusD undergo mRNA splicing and translation into the group-specific antigen (Gag) that assembles into a capsid polyprotein, the protease (Pro) for maturation of the gene products and the polymerase (Pol) encoding for reverse transcriptase, RNase H and integrase (Fig. 5a). The MusD enzymatic machinery is also used by the related non-autonomous ETn elements for their retro-transposition[14] (Fig. 5a). Having shown that MusD-*Dac1J* is transcribed, translated and produces VLPs in the AER, we wondered which step of the endogenous retroviral cycle causes the phenotype. We reasoned that the MusD VLPs-mediated AER cell death is directed by either the retroviral RNA, the capsid-containing Gag polyprotein, the replication of the ETn elements or the reverse transcription of its mRNA. To decipher between these mechanisms, we engineered five mutants carrying modification in the MusD-*Dac1J* sequence (Fig. 5b). Two mutants have similar consequences on the *pol* gene: (1) '*Dac1J*-ΔPol', generated from the *Dac1J*/*Dac1J* line, contains a deletion of the 3′ part of MusD-*Dac1J* resulting in a shorter Pol protein (Fig. 5b and Extended Data Fig. 9a); and (2) '*Dac1J*-mut-pol' is a knock-in of the MusD-*Dac1J* with an out-of-frame 493 bp deletion in the *pol* gene, introducing a 2 bp frameshift resulting in a premature STOP codon as previously generated[15] (Fig. 5b). We verified that the Gag protein was correctly translated in this knock-in, showing that it recapitulates *Dac1J*-mut-pol transcription and translation as in the original *Dac1J* line (Fig. 5c and Extended Data Fig. 9b). The lack of a functional Pol protein leads to a major rescue of the dactylaplasia phenotype with a milder, not fully penetrant limb phenotype (Fig. 5d and Extended Data Fig. 9c). A total of 29.6% (32 out of 108) and 15.8% (24 out of 152) of the observed limbs were affected in *Dac1J*-Δ/ΔPol and *Dac1J*-mut-pol KI/KI, respectively. None of these animals showed the classical one-digit dactylaplasia phenotype (Fig. 5d).

As LTR elements, MusD and ETn have tRNA primer-binding sites (PBS) to initiate reverse transcription and copy their RNA into DNA to reinsert themselves in the genome[35] (Fig. 5a). The *Dac1J pol* mutants prevented the replication of both MusD-*Dac1J* itself and non-autonomous ETns. To decipher whether replication of ETn elements has a role in the phenotype, we engineered a knock-in of the full-length MusD-*Dac1J* with a disrupted PBS (Fig. 5b and Extended Data Fig. 9a). A coding-competent MusD element with a scrambled PBS is no longer able to prime its reverse transcription but can still replicate ETns[36] as it correctly expresses retroviral proteins (Fig. 5c and Extended Data Fig. 9b). *Dac1J*-mut-PBS mutants also showed a partial rescue of the dactylaplasia phenotype with 24% (21 out of 87) of observed limbs affected (Fig. 5d and Extended Data Fig. 9c). The partial rescue of the

MusD-*Dac1J* element with disrupted *pol* and PBS suggest that reverse transcription of the MusD-*Dac1J* RNA is a major cause of dactylaplasia. Yet the presence of Gag or retroviral RNA can drive limb phenotypes to a lesser extent.

### Lack of MusD-*Dac1J* Gag capsid fully rescues dactylaplasia

To decipher whether Gag or the retroviral RNA is responsible for these limb phenotypes, we generated two other knock-ins affecting the *gag* gene (Fig. 5b). Both the *gag* complete deletion (*Dac1J*-mut-gag) and partial deletion, leaving the matrix intact but deleting the capsid (CA) protein, (*Dac1J*-mut-gag(CA)) (Extended Data Fig. 9a) showed a full rescue of the phenotype with none of the observed limbs affected (Fig. 5d). We validated that the MusD-Gag polyclonal antibody recognizes a truncated (*Dac1J*-mut-gag(CA) mutant) but not an absent (*Dac1J*-mut-gag mutant) GAG (Fig. 5c and Extended Data Fig. 9b).

For all four knock-ins, none of the observed heterozygote animals show a phenotype. We also validated that the variable phenotypes do not depend on DNA methylation variability, as all animals show hypomethylated 5′LTR (Extended Data Fig. 9d,e). These observations suggest that the MusD-*Dac1J* Gag capsid is essential to drive a limb phenotype, but in itself is not sufficient to generate a phenotype as severe as the full-length element.

### MusD-*Dac1J* adopts the expression surrounding its insertion

Activation of the MusD-*Dac1J* in the AER during development is driven by adoption of the *Fgf8* enhancers. To decipher whether this was specific to the *Fgf8* regulatory domain or a more general mechanism, we inserted the 5′LTR-LacZ construct into the *Lbx1*, *Shh* and *Sox9* TADs (Fig. 6a and Extended Data Fig. 10a,b). When inserted in the *Lbx1* TAD (Fig. 6a), adjacent to but separated from the *Fgf8* domain by a TAD boundary (Fig. 3a), LacZ expression recapitulated the *Lbx1* expression in muscle progenitors of the developing limbs[37] (Fig. 6b,c). Similarly, 5′LTR-LacZ insertion in the *Shh* and the *Sox9* TADs sensed the surrounding expression in the developing notochord and the limb chondrocytes, respectively (Extended Data Fig. 10c–f and Supplementary Note 5). Altogether, this reveals yet undescribed properties of an unmethylated 5′LTR, adopting the regulatory activity present in the TAD where it has been inserted.

To investigate whether the production of the Gag-MusD polyprotein is specific to *Fgf8*-expressing cells or could also exist in different limb cells, we engineered knock-ins of MusD-*Dac1J* in its full-length and scrambled PBS forms in the *Lbx1* TAD (Fig. 6a). Gag-MusD staining was observed in the *Lbx1*-expressing cells of the developing limb (Fig. 6d and Extended Data Fig. 10g). This suggests that MusD VLPs can assemble in any embryonic cells when adopting the regulatory information from a developmental gene. The AER and branchial arches are embryonic epithelium, but *Lbx1*-expressing cells are from the muscle lineage. Gag-MusD expression in *Lbx1*-expressing cells was maintained at E11.5 (Fig. 6d), whereas at this stage, most of the AER cells are dead in the *Dac1J*/*Dac1J* mutants (Fig. 2c). When assessing cell death at E12.5, we did not observe muscle progenitor cells undergoing apoptosis (Extended Data Fig. 10h), and embryos at E18.5 did not show any striking morphological phenotype. This suggests that cells from a muscle lineage tolerated VLP production during development. These results indicate that MusD VLPs production during development affects embryonic cells differently and highlight that mouse post-implantation development can proceed with the presence of VLPs.

### Discussion

In this study, we uncover that an ERV element escaping epigenetic silencing adopts the regulatory activity of nearby developmental genes, resulting in time-specific and cell-type-specific expression of retroviral products during organogenesis. In the dactylaplasia mutant, VLP assembly in the AER of the developing limb affects cell survival at a critical time preceding digit formation, leading to a lack of digits in

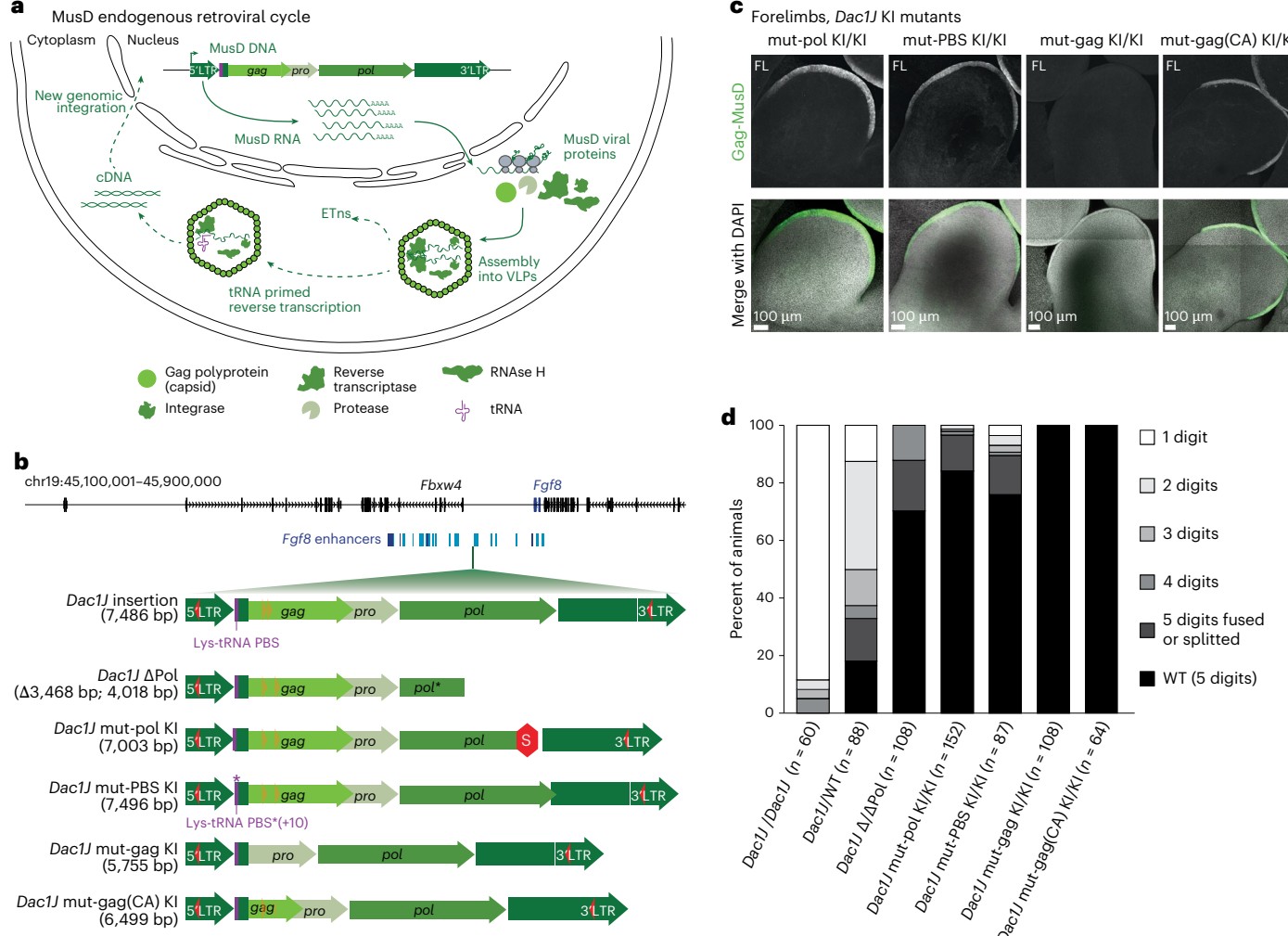

**Fig. 5 | Knock-ins of MusD-*Dac1J* carrying mutation rescue dactylaplasia phenotypes. a**, Schematic representation of the MusD endogenous retroviral cycle in a cell. MusD DNA insertion is transcribed into mRNA in the nucleus. MusD RNA is then translated into the Gag (producing the capsid and matrix), Pro (the protease) and Pol (producing the reverse transcriptase, RNase H and integrase) in the cytoplasm and assembled into VLPs. The assembled VLPs contain reverse transcriptase, integrase and RNase H and are formed by the Gag capsid, which was cleaved by the protease. The VLP can replicate the non-autonomous ETn retrotransposons but also undergo tRNA-primed reverse transcription to reintegrate in the genome through a cDNA intermediate (as indicated with the dashed arrows). **b**, Representation of the *Dac1J* insertion along with the five mutated versions. Red and orange triangles represent CTCF sites. Purple band shows the location of the Lys-tRNA primer binding site, as wild-type and mutated (*). Red hexagon represents a Stop codon. **c**, Anti-GAG-MusD whole-mount immunofluorescence on E10.5 knock-in mutants: mut-pol KI/KI, mut-PBS KI/KI, mut-gag KI/KI and mut-gag(CA) KI/KI forelimb. At least *n* = 3 biological replicates were confirmed. **d**, Histogram of the percentage of animals showing the six possible digit phenotype situations in the different mutants. *n* represents the number of limbs (forelimbs and hindlimbs) analyzed.

newborn mice. The MusD-*Dac1J* insertion is one of the few reported mutagenic LTR insertions not located in an intron of a gene[13]. Independent of *Dac1J*, a second MusD insertion (*Dac2J*) was reported in the intron of the *Fbxw4* gene, also causing dactylaplasia[38]. Given that *Dac2J* affects *Fbxw4* transcription and both insertions are nearby *Fgf8*, dactylaplasia was believed to be primarily caused by disruption of gene transcription[38–41].

Here, we show that the MusD-*Dac1J* insertion does not directly affect gene transcription. Instead, it adopts *Fgf8* regulatory elements as defined by the TAD boundaries. Similar results were obtained at other loci, indicating that adoption of local regulatory information by a TE element is likely to be a more common mechanism. *Fgf8* conditional inactivation in the limb leads to a milder phenotype than in the dactylaplasia mutants[42,43], which more closely resembles the double *Fgf8*; *Fgf4* inactivation[44]. The dactylaplasia phenotype is more comparable to the removal of AER at stage 25 in chick embryos, corresponding

to mouse E11.5, the stage at which AER cells are lost in *Dac1J*/*Dac1J* embryos[45]. These observations support our data showing that the morphological changes are caused by loss of AER cells at a critical developmental stage rather than changes in gene expression.

VLPs from LTR retrotransposons have been observed in pathological contexts when global epigenetic changes occur, resulting in the derepression of many elements from several families; for example, in the developing brain[46], tumor cells[47] or age-related senescence[9]. It is unclear whether this results from the activation of specific elements or the widespread activation of thousands of them. Here, we report the production of VLPs during organ formation in the mouse embryo as a result of epigenetic de-silencing of a single MusD element. The *Dac1J* element shows high sequence similarity with the identified MusD 'master' copies that are competent and autonomous for retrotransposition[15] (Extended Data Fig. 6a), suggesting that de novo integration of the MusD-*Dac1J* element could occur, leading to genomic

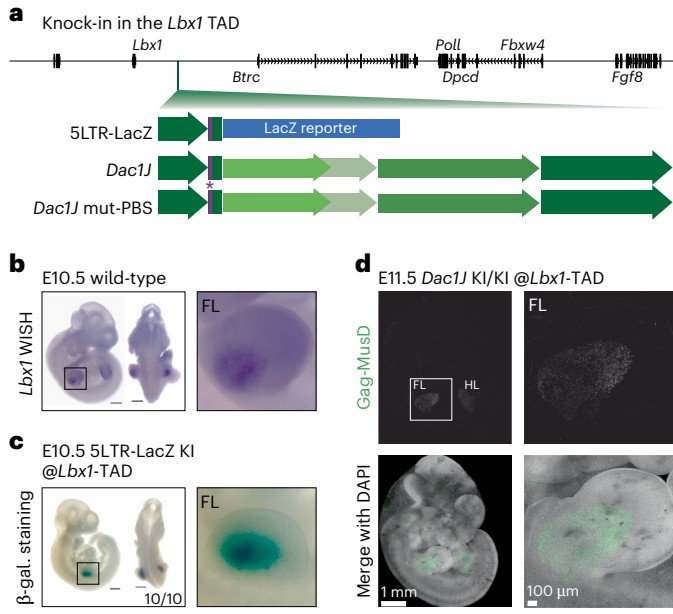

**Fig. 6 | MusD-*Dac1J* adopts the muscle progenitor expression when inserted in *Lbx1* TAD. a**, Representation of the three knock-in lines engineered in the *Lbx1* TAD. Purple band shows the location of the Lys-tRNA primer binding site, as wild-type and mutated (*). **b**, In situ hybridization for *Lbx1* in E10.5 wild-type showing whole embryos and forelimb. Scale bars, 500 µm. At least *n* = 3 biological replicates were confirmed. **c**, β-galactosidase staining on E10.5 *Dac1J*-5LTR-LacZ (*Lbx1* knock-in) showing whole embryos and forelimb. Scale bars, 500 µm. *n* = 10 out of 10 embryos show similar staining. **d**, Anti-Gag-MusD whole-mount immunofluorescence on E11.5 *Dac1J* KI/KI (*Lbx1* knock-in) showing whole embryo (left) and forelimb (right). At least *n* = 3 biological replicates were confirmed.

instability. However, detecting new insertion would be very challenging: MusD-*Dac1J*-expressing cells represent around 4% of all embryonic limb cells (Fig. 2c), and they die rapidly after VLP assembly.

The dactylaplasia phenotype is variable and not fully penetrant in mice carrying a heterozygous insertion of MusD-*Dac1J*, but all homozygotes display a severe phenotype, indicating a dose-dependent effect of the VLPs. By contrast, *Dac1J* mutants carrying mutations in the *pol* gene or the PBS exhibited a mild and low-penetrant phenotype. This remaining phenotype could be caused by the presence of the Gag capsid, as mutations in the *gag* gene show a complete rescue of the phenotype. However, we cannot fully exclude that *pol* genes from other active MusD elements contribute to the phenotype. Moreover, it is possible that inefficient reverse transcription taking place outside of the capsid also participates in the complete rescue of the *Dac1J* Gag mutant. Overall, this suggests that the main trigger of the apoptotic signaling is the Gag capsid, but it also demonstrates that Gag alone cannot drive complete dactylaplasia phenotypes.

Our findings raise the question of whether VLP-mediated cell death could happen in embryonic cells other than those of the AER. As a result of MusD-*Dac1J* expression, a few cells in the branchial arches of the *Dac1J* embryos undergo cell death. This does not seem to be sufficient to lead to any apparent morphological phenotype, probably because proliferating surrounding tissue can replace the few dying cells. We also showed that MusD-Gag is produced in *Lbx1*-expressing cells of the developing limb when MusD is inserted in the *Lbx1* TAD. Cell death was not detected in those cells, which could be explained by cell lineage specificity, genomic loci or dose-dependent effects. MusD-derived VLPs could also be beneficial for the embryo, as shown in humans, whereby blastocyst development proceeds with the presence of HERVK particles[48]. There are a growing number of studies showing Gag-derived proteins co-option[49–51]. Whether or not VLP production could also be

co-opted to benefit mouse development is yet to be studied, but recent work presents evidence that Gag proteins encoded by ERVs are essential for zebrafish and chicken embryonic development[52].

Identifying the causative variants in patients with congenital malformations remains challenging. Our data suggest that variable DNA methylation states at a single ERV can result in malformations induced by cell-type-specific and time-specific expression of the ERV in a pattern of the host gene. Unlike most human ERVs, HERVK (HML-2) retained copies with intact open reading frames for retroviral proteins and was shown to assemble into VLPs in the human early embryo[48]. Long-interspersed nuclear elements are active in humans, and their transcription was shown to be associated with immunogenic effects[53]. Aberrant activation of HERVK or long-interspersed nuclear elements in developmental cells might potentially be involved in developmental disorders that so far remain poorly understood.

## Online content

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

[1]RG Development & Disease, Max Planck Institute for Molecular Genetics, Berlin, Germany. [2]Microscopy and Cryo-electron Microscopy Service Group, Max Planck Institute for Molecular Genetics, Berlin, Germany. [3]Department of Computational Molecular Biology, Max Planck Institute for Molecular Genetics, Berlin, Germany. [4]Institute for Medical and Human Genetics, Charité-Universitätsmedizin Berlin, Berlin, Germany. [5]Department of Developmental Genetics, Max Planck Institute for Molecular Genetics, Berlin, Germany. [6]Berlin–Brandenburg Center for Regenerative Therapies, Charité-Universitätsmedizin Berlin, Berlin, Germany. [7]Present address: Department of Pathology, New York University School of Medicine, Langone Health Medical Center, New York, NY, USA. [8]Present address: Centro Andaluz de Biología del Desarrollo (CABD), Consejo Superior de Investigaciones Científicas/ Universidad Pablo de Olavide, Seville, Spain. [9]Present address: Department of Genetic Medicine and Development, Faculty of Medicine, University of Geneva, Geneva, Switzerland. [10]Present address: Institute of Genetics and Genomics in Geneva (iGE3), University of Geneva, Geneva, Switzerland. ✉e-mail: glaserj@ie-freiburg.mpg.de; mundlos@molgen.mpg.de

## Methods

The experiments were not randomized. The investigators were not blinded to allocation during experiments and outcome assessment. All animal procedures were conducted as approved by the local authorities (LAGeSo Berlin) under license numbers G0243/18 and G0176/19.

### Mice

For embryo isolation, mice were killed by cervical dislocation, and the uteri were dissected in PBS. All animal experiments followed all relevant guidelines and regulations. Mice were housed in a centrally controlled environment with a 12 h light, 12 h dark cycle, temperature of 20–22.2 °C and humidity of 30–50%. Routine bedding, food and water changes were performed. The *Dac1J* line (SM;NZB background) was obtained from the Jackson Laboratory (strain no. 002759) and backcrossed for at least ten generations with 129s2/Sv and C57BL/6 wild-type mice to obtain the *Dac1J*-129sv and *Dac1J*-BL/6 lines.

### Mouse embryonic stem cell targeting

Culture and genome editing of mouse embryonic stem (ES) cells were performed as previously described[27,54]. G4 (XY, 129sv × C57BL/6 F1 hybrid, wild-type) and *Dac1J*-129s2/Sv (derived in-house from mouse blastocysts) cells were used. In brief, mouse ES cells were seeded on a monolayer of CD1 feeder cells, and transfection was performed using FuGENE technology (Promega, cat. no. E5911). A total of 8 μg of each CRISPR construct containing the single guide RNA (sgRNA) of interest delivered through the pX459 vector and 4 μg of the DNA construct for homology-directed repair was used. After 24 h, cells were split onto DR4 puromycin-resistant feeders, and selection with puromycin was carried out for 2 days. Resistant-growing clones were then picked and grown into 96-well plates on CD1 feeders. All clones were genotyped by PCR (Supplementary Table 3). Positive clones were expanded and their genotyping was further confirmed by Sanger sequencing of PCR products. They were used for further experiments only if the successful modification could be verified. A list of sgRNAs (designed using the Benchling CRISPR guides tool) used for CRISPR–Cas9 genome editing is given in Supplementary Table 4. One of the sgRNAs is located in the repetitive MusD pol region; Supplementary Table 5 shows all off-targets for this sgRNA. All knock-in DNA vectors used for transfection were cloned using Gibson cloning to assemble the knock-in DNA and the homology arms targeting the *Fgf8*, *Lbx1*, *Shh* or *Sox9* loci. The mm10 coordinates of the homology arms used in the knock-in constructs can be found in Supplementary Table 6. The β-globin-LacZ construct contains the minimal β-globin promoter (57 bp) upstream of the LacZ gene (3,452 bp). The *Dac1J* mut-PBS sequence (7,496 bp) was synthesized and subsequently cloned into a pUC vector by Genewiz. *Dac1J* mut-pol, *Dac1J* mut-gag and *Dac1J* mut-gag(CA) were generated by Gibson cloning. The 5LTR-LacZ constructs contain the MusD-*Dac1J* 5′LTR (318 bp) and 5′UTR (331 bp, containing the same insertion in the Lys-tRNA PBS as in *Dac1J*-mutPBS) and the LacZ gene (3,452 bp).

### Generation of mutant mice

Mice were generated from genome-edited mouse ES cells by diploid or tetraploid aggregation[55]. Confirmed CRISPR–Cas9 mutant cell lines after expansion were seeded on CD1 feeders and grown for 2 days before aggregation. Female mice of the CD1 strain were used as foster mothers. The data collection was performed according to the stage of each sample, and investigators were not blind to genotype, given that mouse breeding and analysis required knowledge about the genotype at hand. For Δ*Dac1J*, Δ*Dac1J*-Pol (both from *Dac1J*-129sv mouse ES cells), β-globin-LacZ KI at *Fgf8*, 5LTR-LacZ KI at *Fgf8*, 5LTR-LacZ KI at *Lbx1*, 5LTR-LacZ KI at *Shh*, 5LTR-LacZ KI at *Sox9*, *Dac1J* mut-gag, *Dac1J* mut-gag(CA) and *Dac1J*-KI at *Lbx1* (all from G4 mouse ES cells), only embryos from tetraploid aggregation were generated for analysis at different embryonic stages. For *Dac1J*-mutPBS KI at *Fgf8*, *Dac1J*-mutPBS KI at *Lbx1* and *Dac1J*-mut-pol KI at *Fgf8* (all from G4 mouse ES cells),

adult mice were generated by diploid aggregation. Founder animals for each mouse line were backcrossed with 129sv wild-type animals to establish line stock. The selection of animals for analysis and breeding was random. Both males and females were used for analysis. For all MusD-*Dac1J* knock-in mice, the CpG DNA methylation status of the knocked-in 5LTR was assessed by bisulfite sequencing. Mice were used for the study only if the knocked-in 5LTR was hypomethylated. If the 5LTR of the *Dac1J* was methylated at more than 20%, animals were excluded from the study. The rationale for this exclusion is that a knock-in with hypermethylated *Dac1J*-5LTR will silence the transcript, preventing a conclusion on the effect of the mutated retroviral product.

### Skeletal preparation

E18.5 fetuses were killed and stored at −20 °C. On the first day of the staining protocol, E18.5 fetuses were kept in $H_2O$ for 1–2 h at room temperature (20–25 °C) and heat-shocked at 65 °C for 1 min. The skin, abdominal and thoracic viscera were gently removed using forceps. The fetuses were then fixed in 100% ethanol for at least 24 h. Alcian blue staining solution (150 mg alcian blue 8GX in 80% ethanol and 20% acetic acid) was used to stain the cartilage. After 24 h, fetuses were rinsed and post-fixed in 100% ethanol overnight, followed by 24 h incubation in 0.2% KOH in $H_2O$ for initial clearing. The next day, fetuses were incubated in alizarin red (50 mg l$^{-1}$ alizarin red S in 0.2% KOH) to stain the bones for 24 h. Rinsing and clearing were then carried out for 1–3 days using 0.2% KOH. The stained embryos were dissected in 25% glycerol, imaged using a Zeiss SteREO Discovery V12 microscope and Leica DFC420 digital camera and subsequently stored in 80% glycerol.

### β-galactosidase staining

Embryos were dissected in ice-cold PBS and fixed in 4% paraformaldehyde (PFA) in PBS for 20 min (E11.5 or younger) or 30 min (E12.5) and then washed several times in PBS. Embryos were subsequently washed two times for 15 min in wash solution (2 mM $MgCl_2$, 0.02% NP-40 and 0.01% $C_{24}H_{39}NaO_4$ in PBS) and finally incubated at 37 °C in X-gal solution (165 mg ml$^{-1}$ $K_4Fe(CN)_{63}H_2O$, 210 mg ml$^{-1}$ $K_3Fe(CN)_6$ and 40 mg ml$^{-1}$ X-gal diluted 1:50 in wash solution). Staining was assessed regularly, and the reaction was stopped when LacZ staining was clearly apparent (between 1 h and 12 h of staining). Samples were then washed several times with PBS and kept at 4 °C in PBS with a few drops of 4% PFA. Embryos were imaged using a Zeiss SteREO Discovery V12 microscope and a Leica DFC420 digital camera.

### DNA methylation analysis

Genomic DNA isolated from embryonic tissue was obtained following overnight lysis at 50 °C (100 mM Tris pH 8.0, 5 mM EDTA, 200 mM NaCl, 0.2% SDS and proteinase K). DNA was recovered by standard phenol–chloroform–isoamyl alcohol extraction and resuspended in water. Bisulfite conversion was performed on 1 μg of DNA using the EpiTect Bisulfite kit (Qiagen). Bisulfite-treated DNA was PCR-amplified (nested PCR), then either cloned and sequenced or analyzed by pyrosequencing. For cloning sequencing, at least eight clones were Sanger-sequenced and analyzed with BiQ Analyzer software[56] to analyze all 19 CpGs within the *Dac1J* 5′LTR region. Pyrosequencing, targeting CpG1–8, was performed on the PyroMark Q24 according to the manufacturer's instructions, and results were analyzed with the associated software. All bisulfite primers are listed in Supplementary Table 3.

### RNA-seq

Early E11.5 forelimb buds were microdissected from wild-type and mutant embryos in cold PBS and immediately snap-frozen for storage at −80 °C. Total RNA was extracted using the RNeasy Mini Kit according to the manufacturer's instructions. Samples were poly-A enriched, prepared into libraries using the Kapa HyperPrep Kit and sequenced on a NovaSeq 6000 with 75 bp or 100 bp paired-end reads. RNA-seq experiments were performed in triplicate. For processing, see Supplementary Methods.

## scRNA-seq

scRNA-seq experiments were performed in single replicates (except for the E9.5 *Dac1J/Dac1J* samples for which we had two replicates) as previously described[27]. In brief, E9.5, E10.5 and E11.5 limb buds of wild-type and *Dac1J/Dac1J* embryos were microdissected in ice-cold PBS. A single-cell suspension was obtained by incubating the tissue for 10 min at 37 °C in 200 µl Gibco trypsin-EDTA 0.05% (Thermo Fisher Scientific, cat. no. 25300054) supplemented with 20 µl 5% BSA. Trypsinization was then stopped by adding 400 µl of 5% BSA. Cells were then resuspended by pipetting, filtered using a 0.40 µm filter, washed once with 0.04% BSA, centrifuged (5 min at 150*g*) and resuspended in 0.04% BSA. The cell count was determined using an automated cell counter (Bio-Rad), and cells were subjected to scRNA-seq (10× Genomics, Chromium Single Cell 3′ v2), aiming for a target cell recovery of up to 10,000 sequenced cells per sequencing library. Single-cell libraries were generated according to the 10× Genomics instructions. Libraries were sequenced with a minimum of 230 million 75 bp paired-end reads according to standard protocols. For processing, see Supplementary Methods.

## cHi-C

The cHi-C protocol was performed as previously described[27]. In brief, E11.5 mouse limb buds were prepared in 1× PBS and dissociated with trypsin treatment. A total of $2.5–5 × 10^6$ nuclei were used for crosslinking, then snap-frozen and stored at −80 °C. Snap-frozen pellets were digested with DpnII, ligated and de-crosslinked. The final library was checked on agarose gel and subsequently used for cHi-C preparation. Libraries were sheared using a Covaris sonicator (duty cycle, 10%; intensity, 5; cycles per burst, 200; time, six cycles of 60 s each; set mode, frequency sweeping; temperature, 4–7 °C). Adaptors were added to the sheared DNA and amplified according to Agilent instructions for Illumina sequencing. The library was hybridized to the custom-designed SureSelect beads and indexed for sequencing following Agilent instructions. SureSelect enrichment probes were designed over the genomic interval chr19:44,365,510–46,325,510 (mm10) using the Agilent Sure-Design online tool (https://earray.chem.agilent.com/suredesign). The coordinates for generating cHi-C matrices were: chr19:44,370,000–46,330,000 (*Dac1J*_mm10 custom genome) and chr19:44,370,000–46,320,000 (5′LTRLacZ_mm10 custom genome). NovaSeq 6000 Illumina technology was used according to the standard protocols, with around 400 million 75 bp or 100 bp (NovaSeq 6000) paired-end reads per sample. For processing, see Supplementary Methods.

## 4C-seq

Mouse limb micro-dissection, cell-dissociation, crosslinking and nuclei extraction were performed as described above for cHi-C. The 4C library preparation was performed as previously described[57], with modifications described below. A total of two to five million cells were used as starting material for all 4C-seq libraries. Experiments for the MusD-*Dac1J* viewpoint in *Dac1J/Dac1J* (C57BL/6) and *Dac1J/Dac1J* (129sv/S2) samples (*n* = 3) were performed as singletons. All 4C-seq primer sequences are listed in Supplementary Table 3. DpnII and BfaI were used as primary and secondary restriction enzymes, respectively. Parallel inverse PCR reactions were performed to amplify from a total of 1.6 µg template per 4C library and viewpoint. Final 4C-seq libraries were indexed using TruSeq index primers for Illumina (NEB, E7335S) and NEBNext High-Fidelity 2× PCR Master Mix (NEB). Samples were sequenced with Illumina Hi-Seq technology according to standard protocols. For processing, see Supplementary Methods.

## ChIP–seq

**Tissue collection and cell fixation.** Embryonic limbs (32 for E10.5 and 15–20 for E11.5) were microdissected in ice-cold 1× PBS and pooled. Limbs were washed once with PBS solution and homogenized in 2 ml trypsin solution for 10 min at 37 °C. Trypsin was stopped by adding 5 ml of a 10% FCS–PBS solution. Then, samples were filtered through

a 40 µm cell strainer and complemented with 10% FCS–PBS solution. Formaldehyde (37%, diluted to a final 1%) was used to fix the samples for 10 min at room temperature; 1.425 M glycine was used to quench the fixation. The formaldehyde solution was removed by centrifugation (400*g*, 8 min), cells were washed in 10% FCS–PBS solution and centrifuged; the pellet was snap-frozen in liquid $N_2$ and stored at −80 °C.

**Chromatin shearing and ChIP–seq.** Chromatin immunoprecipitation was performed using the iDeal ChIP–seq Kit for Transcription Factors (Diagenode, C01010055) according to the manufacturer's instructions. In brief, fixed limbs were lysed and chromatin was sonicated in a size range of 200–500 bp using a Bioruptor Plus Sonication device (40 cycles, 30 s on, 30 s off, at high power setting) in the provided buffers. A total of 20 µg of sheared chromatin was used for CTCF immunoprecipitation with 1 µg of antibody (Diagenode, C15410210). Libraries were prepared using the Kapa HyperPrep Kit (Roche, 07962347001) and sequenced on a NovaSeq2 (E10.5) or Aviti (E11.5) sequencers. For processing, see Supplementary Methods.

## Whole-mount immunofluorescence

Embryos were collected in ice-cold PBS and fixed in 4% PFA for 1 h at 4 °C, washed three times in PBS and stored at 4 °C in PBS + 0.03% Na Azide until staining was performed. On the first day of the protocol, embryos were washed once with PBS for 5 min, permeabilized three times 20 min in 0.5% Triton-X–PBS (PBST) and blocked in 5% horse serum–PBST (blocking solution) overnight at 4 °C. Primary antibody incubation was performed in the blocking solution for 72 h at 4 °C under gentle rotation. Embryos were then washed three times in blocking solution and three times in PBST and incubated overnight in blocking solution at 4 °C with gentle rotation. The next day, the secondary antibody, diluted in blocking solution, was added, and embryos were incubated for 48 h at 4 °C with gentle rotation. Afterward, embryos were again washed three times in blocking solution and three times in PBST and incubated overnight in DAPI diluted in PBS (1:5,000; Sigma-Aldrich, D9542) at 4 °C with gentle rotation. On the last day, embryos were washed three times for 10 min each in PBS and post-fixed in 4% PFA at room temperature for 20 min. Before clearing, embryos were washed three times with 0.02 M phosphate buffer (0.025 M $NaH_2PO_4$ and 0.075 M $Na_2HPO_4$, pH 7.4). The clearing was performed by incubation in RIMS (13% Histodenz (Sigma-Aldrich D2158) in 0.02 M phosphate buffer) at 4 °C for at least 1 day. Whole-mount embryos were then imaged with a Zeiss LSM880 confocal laser-scanning microscope in Fast with Airyscan mode. The rabbit MusD-Gag antiserum was a gift from T. Heidemann's laboratory, as previously generated[34]. Anti-MusD-Gag (1:1,000) was used with goat anti-rabbit Alexa-fluorophore 488 (1:1,000; Invitrogen, A11008). Anti-cleaved-Caspase-3 (rabbit polyclonal, Cell Signaling Technology, 9661, Asp175, 1:400) and anti-γH2AX (rabbit monoclonal, Cell Signaling, mAb 9718) were used with a donkey anti-rabbit Alexa-fluorophore 568 (1:1,000; Invitrogen, A110042).

## TEM

E11.0 embryos were collected in ice-cold PBS and fixed in 2% PFA and 2.5% glutaraldehyde in PBS for 8 h with rotation at 4 °C and then kept in PBS at 4 °C overnight. After washing three times in PBS, limbs were dissected and post-fixed in 0.5% (w/v) osmium tetroxide (Science Services) in PBS for 1 h at room temperature and then washed four times in PBS for 20 min. Samples were incubated for 30 min in 100 mM HEPES buffer pH 7.4 containing 0.1% (w/v) tannic acid (Science Services), washed three times for 10 min in distilled water, contrasted in 2% (w/v) uranyl acetate (Sigma-Aldrich, Merck) for 1.5 h at room temperature and washed once in distilled water, followed by dehydration through a series of increasing ethanol concentrations (30% for 5 min, 50% for 10 min, 70%, 90% and 96% for 15 min each and finally three times for 10 min in 100% ethanol, respectively). Samples were then incubated for 5 min in a 1:1 mixture of propylene oxide (Sigma-Aldrich, Merck)

and absolute ethanol, followed by 5 min in a 1:1 mixture of propylene oxide (Fluka, Merck) and SPURR (Low Viscosity Spurr Kit, Ted Pella) and finally infiltrated with 100 % SPURR at 4 °C overnight. The SPURR mixture was renewed twice the following day. For polymerization, limbs were transferred into flat embedding molds, surrounded with fresh resin and incubated for 3 days at 60 °C. Then, 70 nm sections were cut using a Leica UC7 ultramicrotome equipped with a 3 mm diamond knife (Diatome) and placed on 3.05 mm Formvar Carbon Coated TEM Copper Slot Grids (Plano). Sections were post-contrasted using Uranyless EM-Stain and ready-to-use 3% Lead Citrate (Science Services). To visualize retrovirus-like particles in the apical ectodermal ridge, sections were imaged automatically using Leginon[58] on a Tecnai Spirit transmission electron microscope (FEI) operated at 120 kV, equipped with a 4k × 4k F416 CMOS camera (TVIPS). For full-limb pictures, acquired images (up to 657 micrographs per section through one limb bud) were then stitched to a single montage using the TrakEM2 plugin implemented in Fiji[59,60].

### Immuno-TEM

Embryos at E11.0 were collected in ice-cold PBS and fixed in 2% PFA and 0.2% glutaraldehyde in PBS for 1 h at 4 °C and washed three times in PBS. Limbs were dissected and stained for 30 min in 100 mM HEPES buffer pH 7.4 containing 0.1 % (w/v) tannic acid (Science Services), washed three times for 10 min in PBS and dehydrated in a series of increasing ethanol concentrations (30% for 5 min, 50% for 5 min, 70%, 90% and 96% for 10 min each and finally three times 10 min in 100% ethanol, respectively). Samples were then incubated for 30 min in a 1:1 mixture of 100% ethanol and London Resin Gold (Plano) and finally infiltrated with 100% LR Gold at 4 °C overnight. The following day, a fresh mixture of LR Gold was prepared by adding the accelerator Benzil (Plano) at a concentration of 0.2%. Specimens were incubated for two times for 2 h at room temperature and kept overnight at 4 °C, remaining in this mixture. Before polymerization, a fresh LR Gold solution containing 0.2% benzil was renewed for 4 h on the limbs 1 day later. Limbs were then transferred into flat embedding molds, surrounded with an LR Gold–benzil mixture and polymerized under a 100 W Black-Ray UV Lamp (Plano) for 2 days at 5 °C. Then, 80 nm sections were cut using a Leica UC7 ultramicrotome equipped with a 3 mm diamond knife (Diatome) and placed on 3.05 mm Formvar Carbon Coated TEM Copper Slot Grids (Plano). Grids were protected against dehydration and nonspecific binding by applying the grid on a drop of Aurion blocking solution for goat gold conjugates (Aurion) for 30 min at room temperature. Slices were then incubated with 10 μl of a 1:500 dilution of the rabbit anti-Gag-MusD primary antibody in Aurion blocking solution overnight at 4 °C in a humid chamber. The following day, grids were washed four times in buffer containing 20 mM TRIS and 0.9% NaCl. As secondary antibodies, goat anti-rabbit 10 nm gold conjugates (British BioCell) were used at 0.08 μg ml⁻¹ and incubated for 2 h at room temperature. The immune reaction was stopped by washing four times with TRIS–NaCl buffer. After a short dip into double-distilled water, grids were dried and kept for subsequent examination on a Tecnai Spirit transmission electron microscope (FEI). To visualize the localization of gold conjugates, limb bud sections were imaged at ×6,500 nominal magnification, applying a defocus of −2 μm with a pixel size of 1.7 nm.

### Reporting summary

Further information on research design is available in the Nature Portfolio Reporting Summary linked to this article.

### Data availability

The data generated in this study can be downloaded in raw and processed forms from the National Center for Biotechnology Information Gene Expression Omnibus (GEO) database and are available under accession code GSE246755 (containing SubSeries GSE246750, GSE246751, GSE246752, GSE246753 and GSE246754). Previously published data used in this study are accessible under GSE185774, GSE116794 and GSE84795. Data were mapped to the *Mus musculus* mm10 genome.

### Code availability

Newly generated code for the cHi-C data processing is available on GitHub (https://github.com/mikstapes/JGlaser-etal_Dac1J/) and Zenodo (https://doi.org/10.5281/zenodo.15528362)[61]. All other analyses were performed using previously published or developed tools, as indicated in the Methods section.

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

### Acknowledgements

This work was supported by a grant from the Deutsche Forschungsgemeinschaft (DFG) MU 880/16-1 to S.M. J.G. was supported by the HFSP postdoctoral fellowship (LT000465/2019-L). G.A. is supported by Swiss National Science Foundation grants PP00P3_176802 and PP00P3_210995-6. We thank the laboratory of T. Heidmann for generously giving us the antiserum rabbit MusD-Gag. We thank K. Macura, J. Fielder, C. Franke, N. Michaelis, U. Fisher and A. Stiege for technical support, as well as T. Aktas for scientific support. We thank the Schulz laboratory and I. Dunkel for the use of their pyrosequencer. We thank T. Aktas and W. Schwarzer for their thesis dissertation work. We would like to thank all the members of the Mundlos laboratory for their continuous support and stimulation. Finally, we thank D. Ibrahim, G. Cavalheiro, N. Benetti, K. Chudzik, P. Kurbel, A. Monaco and E. van Leen for their critical reading of the paper.

### Author contributions

J.G. and S.M. conceived the project. J.G. designed the experiments and generated transgenic mouse models with the help of G.C. and Y.A. L.W. performed morula aggregation. J.G. and G.C. performed the WISH experiments and the skeletal preparations. W.-L.C. performed the micro-CT. J.G. performed and analyzed the bisulfite-cloning sequencing, pyrosequencing and bulk RNA-seq. J.G. and M.F. performed the cHi-C and M.H.Q.P. and R.S. carried out the processing. ChIP–seq was performed by C.P., N.B. and J.G. and analyzed by J.G. Preparation of the scRNA-seq samples was performed by J.G., G.C., G.A. and M.F. and processed by C.A.P.-M. and V.S. 4C-seq was performed by M.F. and analyzed by V.L. J.G. performed the whole-mount immunofluorescence and HCR. B.F. and T.M. performed the TEM and immuno-EM experiments. J.G. and S.M.

wrote the paper with input from G.A., G.C. and M.F.; the paper was approved by all authors.

## Funding

## Competing interests

The authors declare no competing interests.

## Additional information

**Extended data** is available for this paper at https://doi.org/10.1038/s41588-025-02248-5.

**Correspondence and requests for materials** should be addressed to Juliane Glaser or Stefan Mundlos.

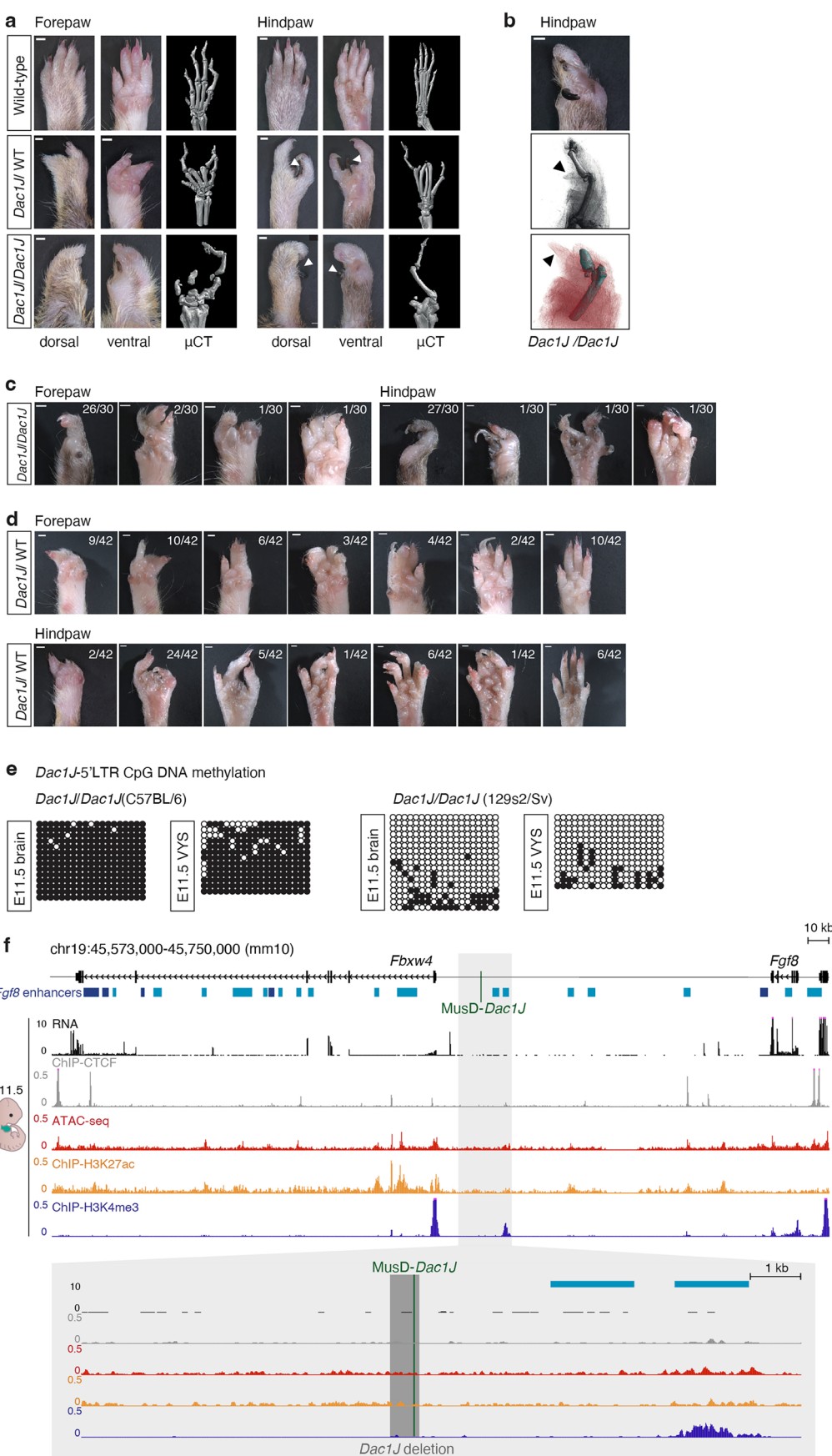

**Extended Data Fig. 1 | See next page for caption.**

**Extended Data Fig. 1 | Dactylaplasia phenotype and epigenetic polymorphism. a**, Dorsal, ventral and micro-computed tomography (μCT) views of wild-type, *Dac1J*/WT (129s2/Sv) and *Dac1J/Dac1J* (129s2/Sv) fore- and hind-paws from 7-month-old adults illustrating the phenotype in heterozygotes and homozygotes respectively. *n* = 2 biological replicates were analysed. White arrows indicate observed nail-like structures. **b**, ventral high-magnification view (top) and micro-computed tomography (μCT) (middle and bottom) of *Dac1J/Dac1J* (129s2/Sv) hind-paw illustrating a typical nail-like-structure (black arrow) which does not contain any bone. **c**, **d**, Ventral views of the various phenotypes observed in *Dac1J/Dac1J* (129s2/Sv) (**c**) and *Dac1J*/WT (129s2/Sv) (**d**) fore- and

hindpaws. *n* = x/x paws with a similar phenotype. For **a-d**, scale bars 1 mm. **e**, DNA methylation status of 19 CpGs from the 5′LTR (promoter) of the MusD-*Dac1J* insertion at the *Fgf8* locus in a C57BL/6 background (left) or 129s2/Sv background (right) measured by bisulfite cloning and sequencing from E11.5 brain, and visceral yolk sac. Constitutive low and high levels of DNA methylation of the MusD-*Dac1J* 5′LTR are observed in the 129s2/Sv and a C57BL/6 background respectively. White circles, unmethylated CpGs; black circles, methylated CpGs. **f**, Bulk RNA sequencing and epigenetic profiling (ATAC-seq, CTCF, H3K27ac and H3K4me ChIP-seq) from E11.5 wild-type forelimbs show that neither the MusD-*Dac1J* insertion nor its deletion seem to affect regulatory elements.

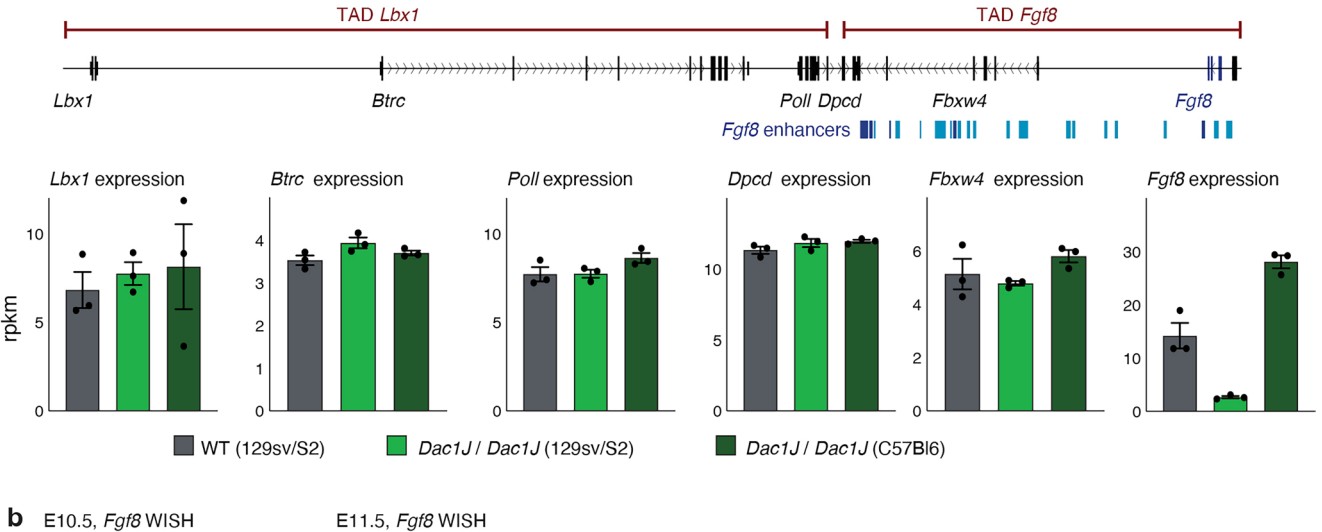

**a** Early E11.5 forelimbs, bulk RNA-sequencing

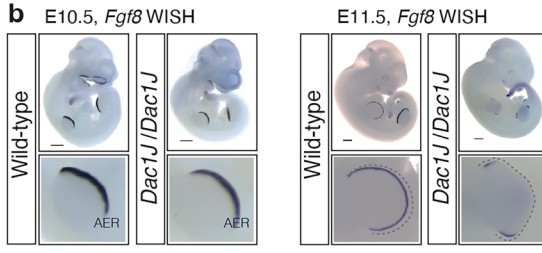

**Extended Data Fig. 2 | Local gene expression changes in *Dac1J/Dac1J* limb buds. a**, Rpkm expression of the 6 genes ate the *Lbx1-Fgf8* locus from bulk RNA-sequencing from early E11.5 forelimbs wild type (grey), *Dac1J/Dac1J* (129s2/Sv) (light green), and *Dac1J/Dac1J* (C57BL/6) (dark green). Data are shown as mean ± s.e.m. of *n* = 3 biological replicates. **b**, In situ hybridization for *Fgf8* at E10.5 (left) and in E11.5 (right) wild-type and *Dac1J/Dac1J* showing whole embryos and forelimbs. Dotted lines draw the shape of the forelimb. AER, apical ectoderm ridge. Scale bars 500um, at least *n* = 3 embryos were analysed per genotype.

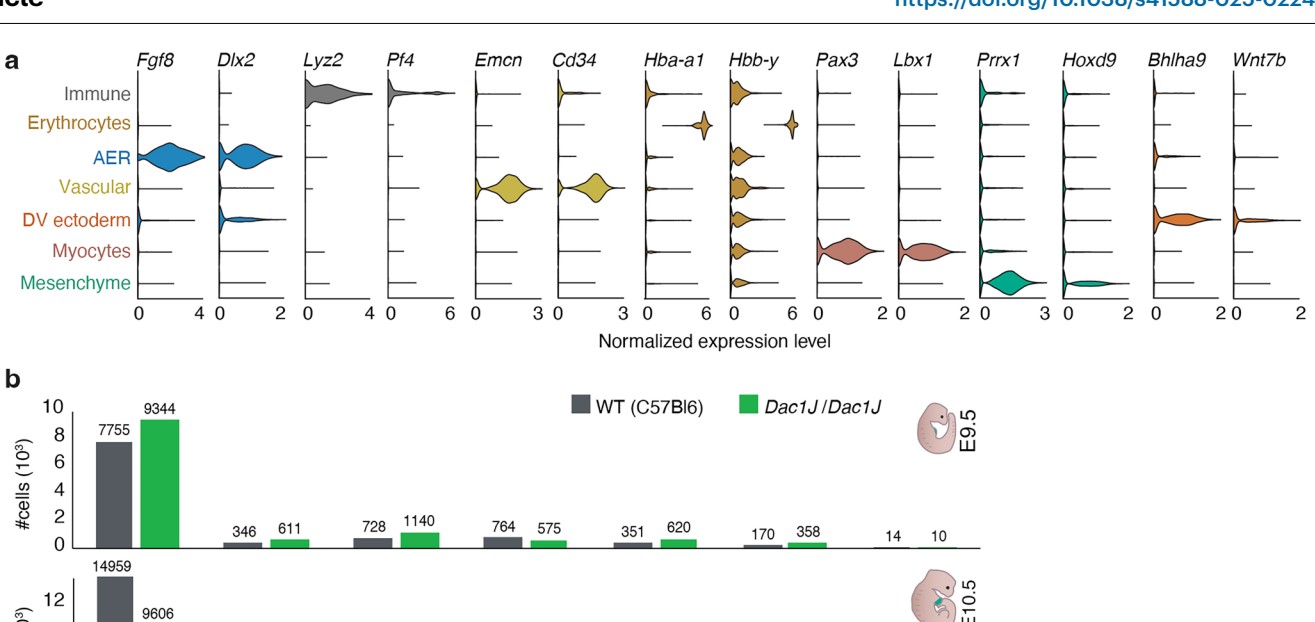

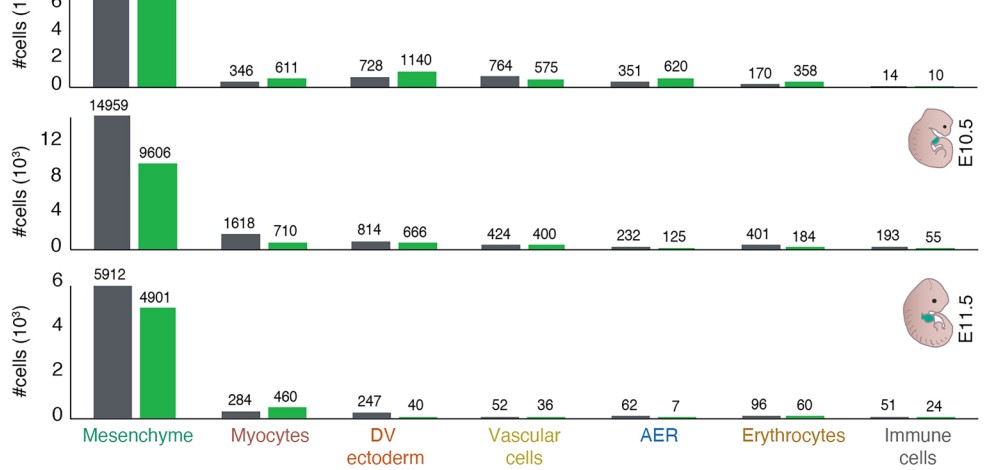

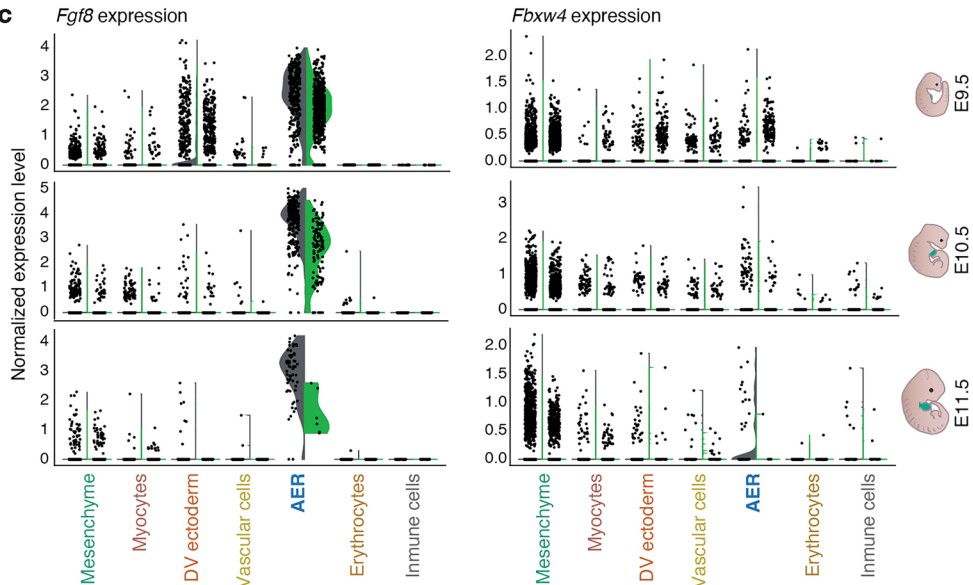

**Extended Data Fig. 3 | Gene expression changes in *Dac1J/Dac1J* AER. a**, Violin plot from E9.5-E11.5 single-cell data showing expression of two marker genes per cluster. **b**, Histogram showing cell number in each cell cluster from wild-type and *Dac1J/Dac1J* at E9.5, E10.5, and E11.5. **c**, Violin plot depicting the expression of *Fgf8* (left) and *Fbxw4* (right) between wild-type (grey) and mutant (green) in each cluster, at the three tested embryonic stages. Each dot represents a cell.

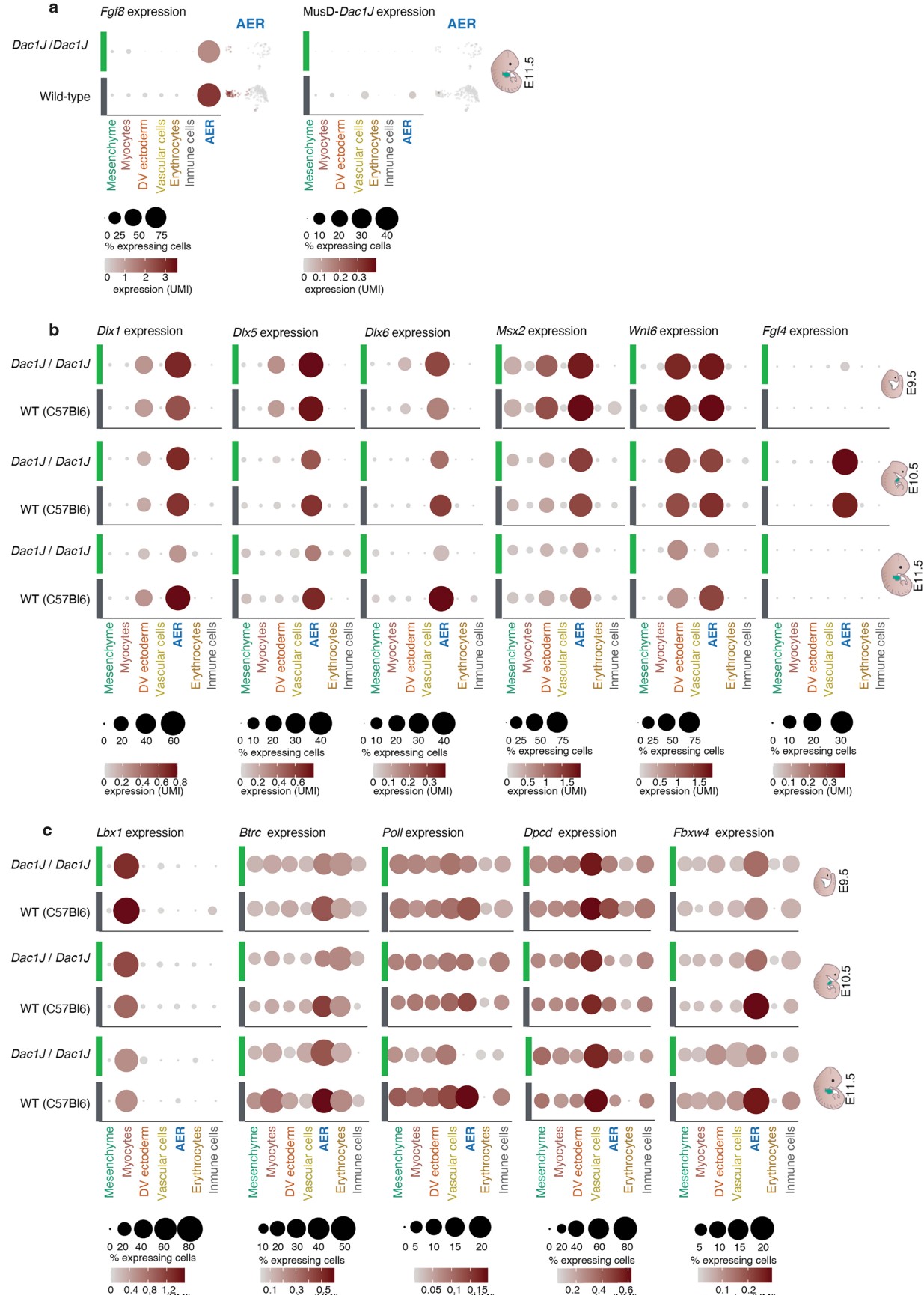

**Extended Data Fig. 4 | Single-cell expression of AER genes and genes at the *Lbx1-Fgf8* locus. a–c,** Dot plots showing expression and percentage of expressing cells for *Fgf8* and MusD-*Dac1J* at E11.5 (**a**), 6 AER genes (*Dlx1, Dlx5,* *Dlx6, Msx2, Wnt6,* and *Fgf4*) at E9.5, E10.5 and E11.5 (**b**), and the 5 genes at the *Lbx1-Fgf8* locus at E9.5, E10.5 and E11.5 (**c**). Data are shown in the 7 forelimb cell cluster in wild-type (grey) and *Dac1J/Dac1J* (green).

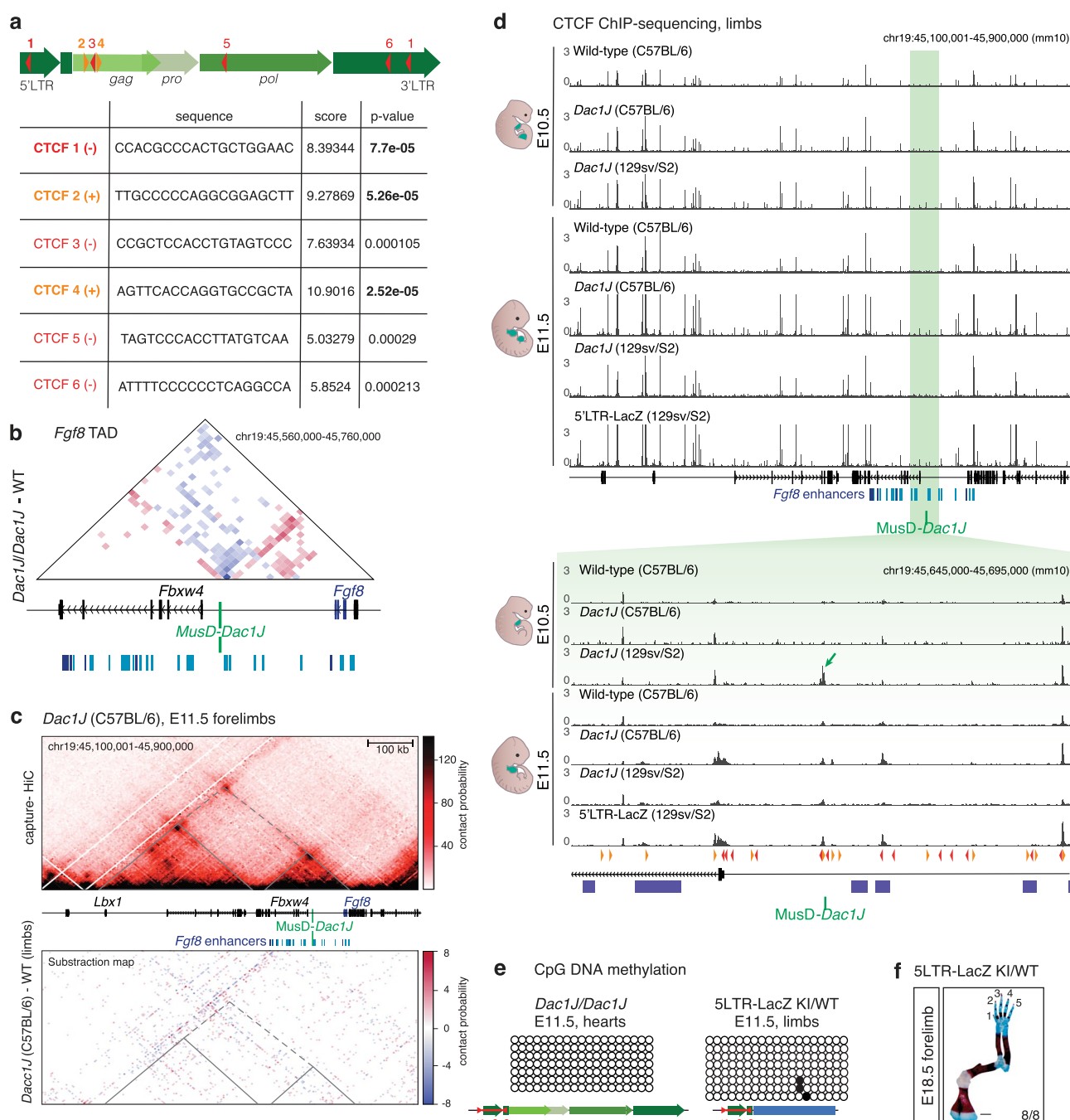

**Extended Data Fig. 5 | 3D conformation changes at the *Fgf8* locus. a**, Sequences of the 6 CTCFs binding sites detected in the MusD-*Dac1J* as identified with the FIMO (Find Individual Motif Occurrences) suite. The program uses a dynamic programming algorithm to convert log-odds scores into *p*-values, assuming a zero-order background model. *p*-value < 10⁻⁴ is considered as significant and marked in bold. FIMO score and *p*-value are indicated for each binding site. Orange and red triangles represent sense and antisense CTCF sites respectively. **b**, Zoom-in from the Capture-Hi-C subtraction map between wild-type and *Dac1J/Dac1J* (129s2/Sv) in Fig. 3d showing the *Fgf8* TAD. **c**, Capture-Hi-C of the *Lbx1/Fgf8*

locus (*mm10, chr19: 45,100,000-45,900,000*) from *Dac1J-Bl6* E11.5 mouse limbs buds. Data show the c-HiC as merged signals of *n* = 2 biological and 2 technical replicates. Subtraction maps between mutants and wild-type show gain (red) and loss (blue) of interaction in the mutant compared to the wild-type. **d**, CTCF ChIP-sequencing from forelimbs at E10.5 and E11.5 showing tracks at the *Fgf8* locus. **e**, CpG DNA methylation status of 19 CpGs in the MusD-*Dac1J* 5'LTR from E11.5 *Dac1J/Dac1J* (129s2/Sv) hearts (left) and E11.5 5'LTR-LacZ KI/WT limbs (right). **f**, Skeletal analysis of E18.5 5LTR-LacZ KI +/- forelimbs stained with alcian blue (cartilage) and alizarin red (bone). Scale bars 1 mm, *n* = 8/8 show a similar phenotype.

**a** Amino acid alignment between autonomous MusD element and MusD-*Dac1J*

| MusD name *(from Ribet et al. 2004)* | mm10 coordinates | GAG alignment to *Dac1J*-GAG (576a.a.) | | | POL alignment to *Dac1J*-POL (778a.a.) | | | Present in 129Sv/s2 genome |
|---|---|---|---|---|---|---|---|---|
| | | Identity | Similarity | Gaps | Identity | Similarity | Gaps | |
| MusD-1 | chr13:61,950,909-61,958,393 | 100% | 100% | 0 | 99.87% | 99.87% | 0 | No |
| MusD-2 | chr2:144,148,955-144,156,442 | 100% | 100% | 0 | 99.87% | 99.87% | 0 | Yes |
| MusD-6 | chr16:91,248,302-91,255,791 | 97.74% | 98.78% | 0 | 98.46% | 99.10% | 0 | No |

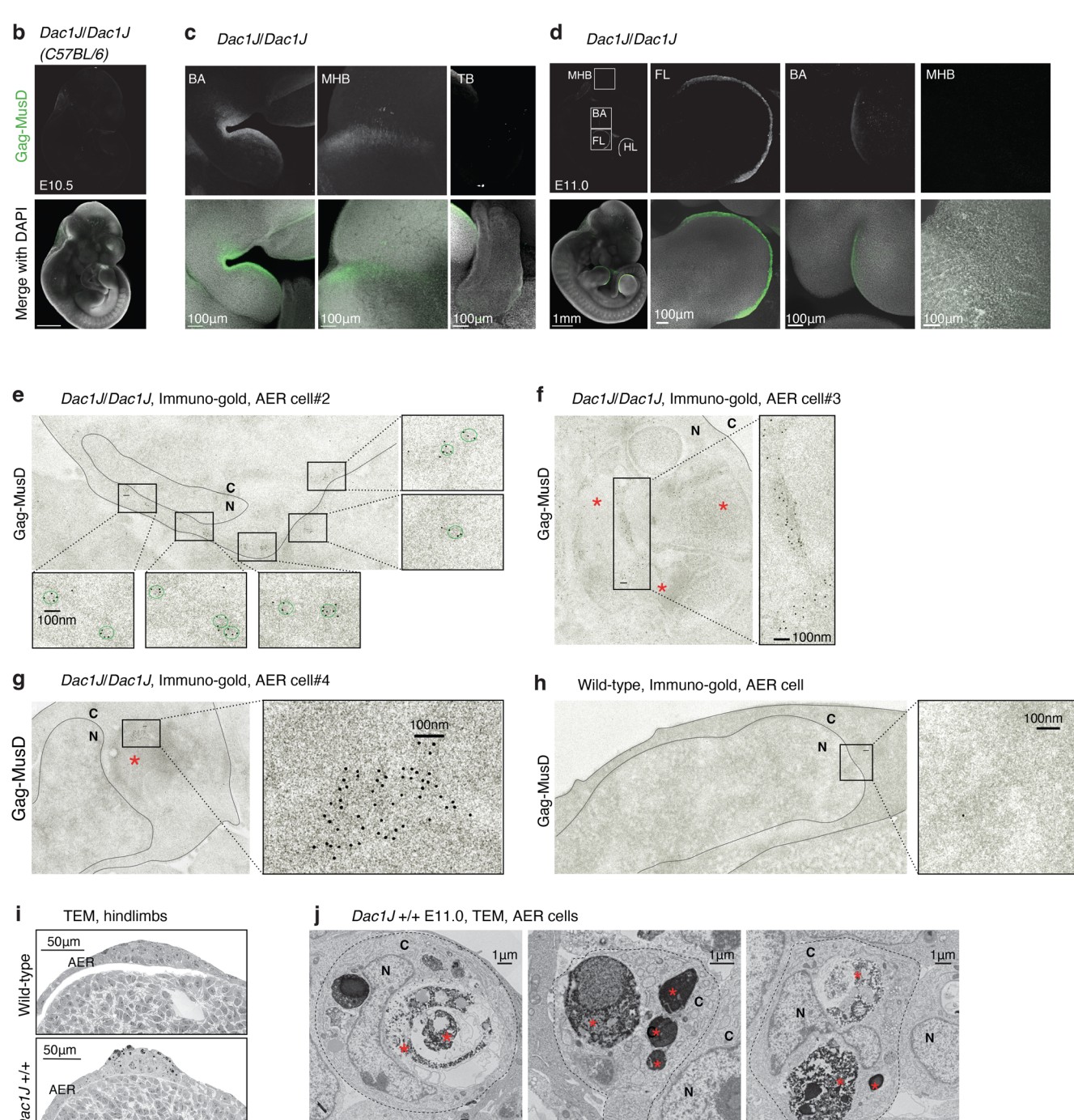

**Extended Data Fig. 6 | See next page for caption.**

**Extended Data Fig. 6 | Gag-MusD expression and VLPs in the *Dac1J* mutant embryos. a**, Table showing the percentage of amino-acid identity between MusD-*Dac1J* and the three MusD (MusD-1, 2, and 6) identified as autonomous for retro-transposition in *Ribet* et al. *2004*. Identity, similarity, and gaps of amino-acids are indicated for both GAG and POL proteins. **b**, anti-GAG-MusD whole-mount immuno-fluorescence on E10.5 *Dac1J/Dac1J* (C57BL/6) embryos showing whole embryo with no staining. Scale bars 1 mm. *n* = 5/5 biological replicates were confirmed. **c**, **d**, anti-GAG-MusD whole-mount immuno-fluorescence on E10.5 (**c**) and E11.0 (**d**) *Dac1J/Dac1J* (129s2/Sv) embryos showing branchial arches (BA), midbrain-hindbrain boundary (MHB), tailbud (TB), whole embryo and limbs (FL, forelimbs; HL hindlimbs). Scale bar 100 nm. *n* = 5/5 (E10.5) and *n* = 4/4 (E11.0) biological replicates were confirmed. **e**–**g**, TEM analysis after immuno-gold labeling with anti-GAG-MusD antibody on E11.0 *Dac1J/Dac1J* AER cells shows cytoplasmic aggregates of GAG. Scale bar 100 nm. *n* = 2 biological replicates and *n* = 6 technical replicates were confirmed. C, cytoplasm; N, nucleus. Green circle represents one VLP with Gag capsid. Red Asterix indicates an apoptotic body. **h**, TEM analysis after immuno-gold labeling with anti-GAG-MusD antibody on E11.0 wild-type control AER cell shows no staining. *n* = 2 biological replicates were confirmed. **i**, **j**, TEM analysis on E11.0 *Dac1J/Dac1J* hindlimbs (**i**) and zoom-in view on 4 forelimb AER cells showing apoptotic bodies and phagocytes (**j**). Scale bars 50um and 1um.

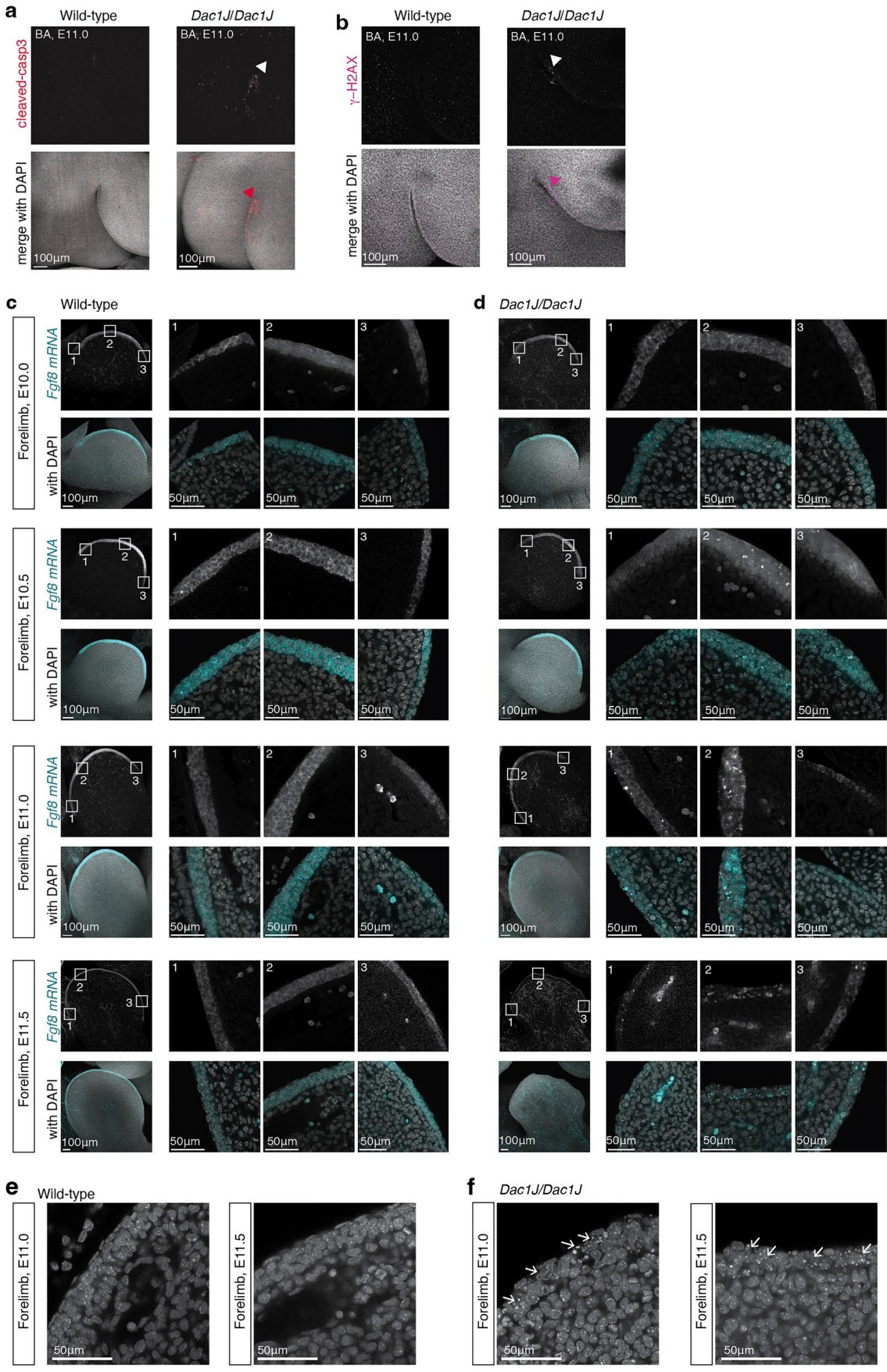

**Extended Data Fig. 7 | See next page for caption.**

**Extended Data Fig. 7 | Apoptotic cell death in the *Dac1J* embryos.**
**a**, **b**, anti-cleaved-Caspase3 (**a**) and anti-gamma-H2AX (**b**) whole-mount immuno-fluorescence on E11.0 wild-type and *Dac1J/Dac1J* showing branchial arches area. Scale bars 100um. At least *n* = 3 biological replicates were confirmed. **c**–**f**, HCR in situ hybridization with an *Fgf8* mRNA probe on showing embryonic forelimbs at E10.0 (30 somites), E10.5 (35 somites), E11.0 (40 somites) and E11.5 (46 somites) in wild-type (**c**) and *Dac1J/Dac1J* (**d**) embryos. Scale bars 100um and 50um. **e**, **f**, DAPI staining showing the AER area of forelimbs in wild-type (**e**) and *Dac1J/Dac1J* (**f**) embryos at E11.0 (40 somites) and E11.5 (46 somites). *n* = 2 biological replicates were confirmed for each genotype and stage. Scale bars 50um.

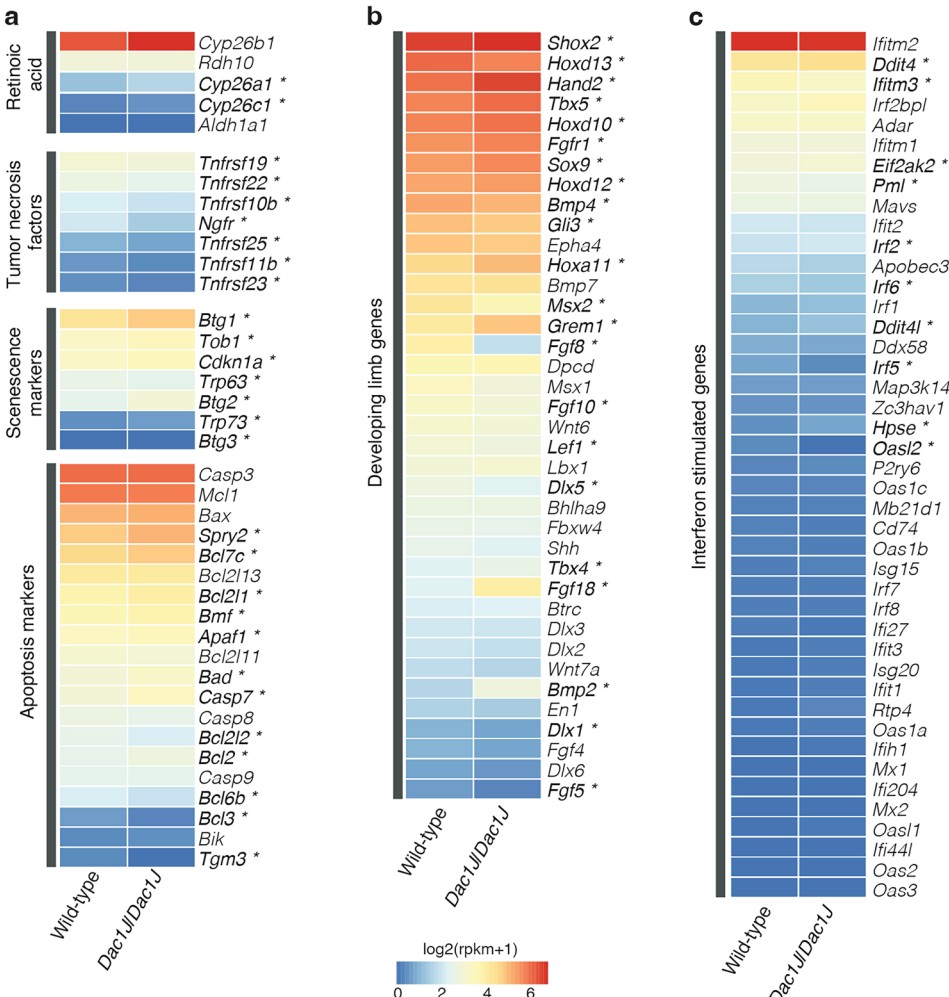

**Extended Data Fig. 8 | Gene expression changes in response to VLPs production in forelimbs. a–c,** Heat-map showing log2 (rpkm+1) of selected apoptosis (**a**), developing limb (**b**), and interferon-stimulated (**c**) genes.

Genes showing significant (*p*-value < 0.05) expression changes in *Dac1J/Dac1J* compared to wild-type are indicated with an asterisk. *P*-value was calculated using the DEseq2 package with a Wald test.

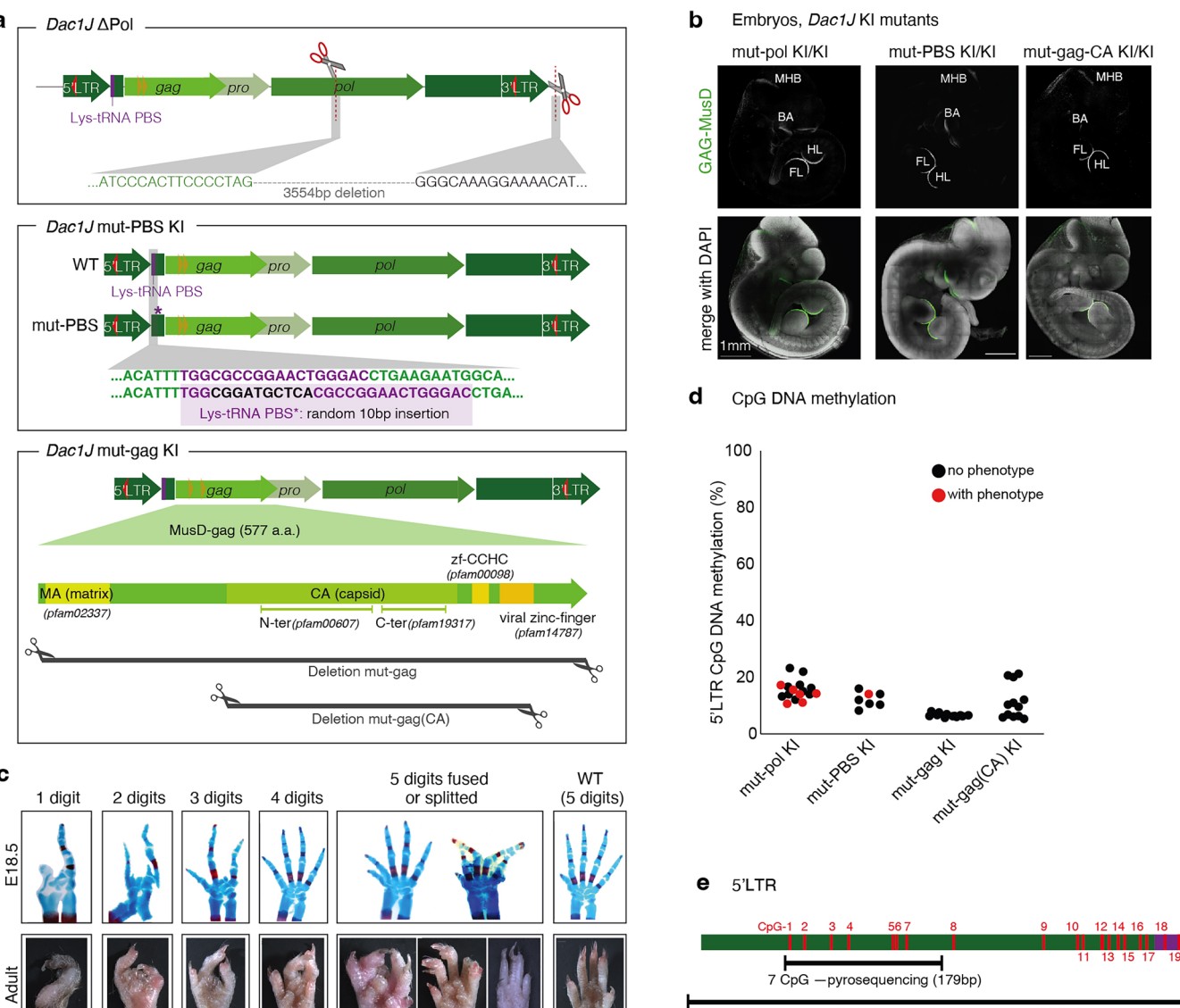

**Extended Data Fig. 9 | Knock-ins of MusD-*Dac1J* carrying mutations.**
**a**, Detail of the 3554 bp deletion of the Pol gene and 3' sequence (top), the 10 bp insertion of random DNA in the *Dac1J*-mutPBS knock-in (middle), and the two Gag mutations (bottom). In the Δ/ΔPol-*Dac1J* AER cells, Gag and Pro assemble into a VLP lacking reverse transcriptase. This incomplete VLP can neither replicate ETns, nor undergo tRNA primed reverse transcriptase of its RNA. In the *Dac1J*-mutPBS knock-in, the 10 bp insertion in the Lys tRNA primer binding site (PBS) leads to a scrambled PBS. In the *Dac1J*-mutPBS AER cells, Gag, Pro, and Pol assemble into VLP. The intact Pol allows possible replication of ETn elements, but the MusD VLPs are unable to undergo tRNA-primed reverse transcription. In the mut-gag knock-in, the entire *gag* gene is deleted, but a complete pro and pol protein can be assembled. In the mut-gag(CA) knock-in, the capsid and the zinc-finger domain are deleted so that the matrix (MA) protein of the gag polyprotein can be produced. **b**, anti-GAG-MusD whole-mount immuno-fluorescence on E11.5

mut-pol KI/KI (left), mut-PBS KI/KI (middle), and mut-gag(CA) KI/KI (right) whole embryos. Scale bars 1 mm. At least *n* = 3 biological replicates were confirmed. FL, forelimb; HL, hindlimb; BA, branchial arches; MHB, mid-hindbrain boundary. **c**, Representation of E18.5 skeletal analysis and adult limbs illustrating the six different observed phenotypes indicated in Fig. 5d. **d**, CpG DNA methylation at the 5'LTR promoter of the *Dac1J* insertion measured by pyrosequencing in *Dac1J* mut-pol KI, mut-PBS KI, mut-gag KI and mut-gag(CA) KI E18.5 embryos. Each dot represents the average CpG DNA methylation measured over 7 CpGs in one individual with a phenotype (red dot) or without a phenotype (black dot). Overall, no difference in 5'LTR DNA methylation is observed between the animals with or without a phenotype. **e**, Schematic representation of the 5'LTR CpGs content and the CpGs measured by pyrosequencing or bisulfite cloning sequencing.

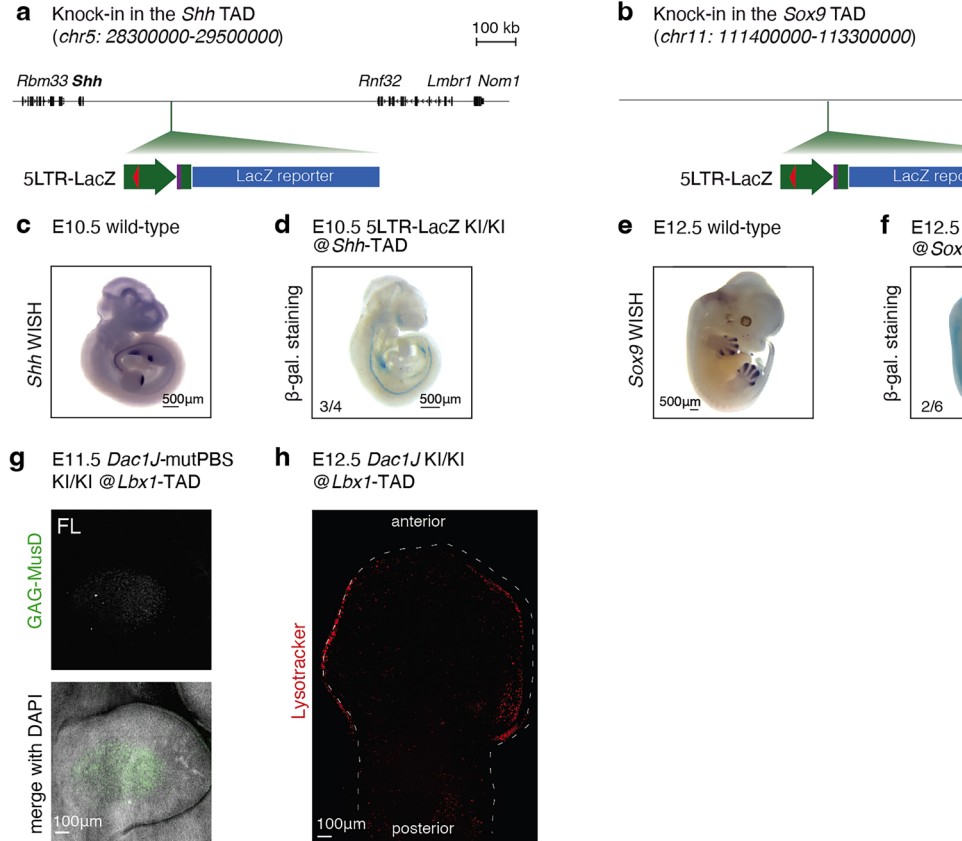

**Extended Data Fig. 10 | MusD-*Dac1J* adopts the expression of genes surrounding its insertion. a**, **b** Schematic representation of the *Dac1J*-5LTR-LacZ knock-in at the *Shh* (**a**) and *Sox9* (**b**) locus. **c**, whole mount *in situ* hybridization of *Shh* from E10.5 wild-type embryo. Scale bar 500um. *n* = 3 biological replicates were confirmed. **d**, beta-galactosidase staining on E10.0 *Dac1J*-5LTR-LacZ (*Shh* knock-in) showing whole embryos. Scale bars 500um, *n* = 3/4 embryos show similar staining. **e**, whole mount *in situ* hybridization of *Sox9* from E12.5 wild-type embryo. Scale bar 500um. *n* = 3 biological replicates were confirmed.

**f**, beta-galactosidase staining on E12.5 *Dac1J*-5LTR-LacZ (*Sox9* knock-in) showing whole embryos. Scale bars 500um, *n* = 2/6 embryos show similar staining. **g**, anti-GAG-MusD whole-mount immuno-fluorescence on E11.5 *Dac1J*-mutPBS KI/KI (*Lbx1* knock-in) showing forelimb staining. Scale bars 100um. At least *n* = 3 embryos were tested. **h**, Lysotracker staining of an E12.5 *Dac1J* KI/KI (*Lbx1* knock-in) forelimb. Expected AER staining for that stage is observed but no muscle cell progenitor staining is detected. *n* = 3 embryos were analysed. Scale bars 100 um.

# Reporting Summary

## Statistics

For all statistical analyses, confirm that the following items are present in the figure legend, table legend, main text, or Methods section.

| n/a | Confirmed | |
|---|---|---|
| ☐ | ☒ | The exact sample size (*n*) for each experimental group/condition, given as a discrete number and unit of measurement |
| ☐ | ☒ | A statement on whether measurements were taken from distinct samples or whether the same sample was measured repeatedly |
| ☐ | ☒ | The statistical test(s) used AND whether they are one- or two-sided<br>*Only common tests should be described solely by name; describe more complex techniques in the Methods section.* |
| ☒ | ☐ | A description of all covariates tested |
| ☒ | ☐ | A description of any assumptions or corrections, such as tests of normality and adjustment for multiple comparisons |
| ☐ | ☒ | A full description of the statistical parameters including central tendency (e.g. means) or other basic estimates (e.g. regression coefficient) AND variation (e.g. standard deviation) or associated estimates of uncertainty (e.g. confidence intervals) |
| ☐ | ☒ | For null hypothesis testing, the test statistic (e.g. *F*, *t*, *r*) with confidence intervals, effect sizes, degrees of freedom and *P* value noted<br>*Give P values as exact values whenever suitable.* |
| ☒ | ☐ | For Bayesian analysis, information on the choice of priors and Markov chain Monte Carlo settings |
| ☒ | ☐ | For hierarchical and complex designs, identification of the appropriate level for tests and full reporting of outcomes |
| ☒ | ☐ | Estimates of effect sizes (e.g. Cohen's *d*, Pearson's *r*), indicating how they were calculated |

*Our web collection on statistics for biologists contains articles on many of the points above.*

## Software and code

Policy information about availability of computer code

| Data collection | All analyses were performed using previously published or developed tools, as indicated in the methods section. No custom software was used to collect the data in this study. |
|---|---|
| Data analysis | - single-cell RNA-sequencing: Computational analysis of the sequenced samples was done with Cell Ranger and the Seurat package v.3 (10x Genomics lnc.). Mapping and preprocessing were done with Cell Ranger default parameters version 3.0.2. We estimated the phase of each cell by assigning a score based on the cell expression of G2/M and S phase markers using the "CellCycleScoring" Seurat function. UMI counts were normalized using scTransform. We built a common latent cell representation across samples by integrating the sample-wise top 50 cell principal components based on the tap 1000 highly variable genes using the Seurat CCA method. ). We clustered cells by first constructing a Shared Nearest Neighbor (SNN) Graph based on the Euclidean distance in the first 20 integrated principal components space using the "FindNeighbors" function with k.param set to 20. Cell clusters were defined using the Louvain algorithm as a modularity optimization technique implemented in the function "FindCluster" with the resolution parameter set to 0.2. Visualization of gene expression was computed after a new scTransform normalization run on the merged raw count assays regressing out for cell cycle and sex effect as previously described. For each cluster, conserved markers between mutant and wild types were identified using the Seurat "FindConservedMarkers" function and were then used for cell-type annotation.<br><br>- Bulk RNA-sequencing: Reads were mapped to the mouse reference genome (mm10) using the STAR mapper(splice junctions based on RefSeq; options: –alignIntronMin 20–alignIntronMax 500000–outFilterMultimapNmax 5–outFilterMismatchNmax 10–outFilterMismatchNoverLmax 0.1). Reads were subsequently used for expression analysis via the Cufflinks package (version 2.2.1; default settings). Transcripts of each sample were assembled using Cufflinks provided with reference gene annotations from Ensembl. The resulting assemblies were then merged via Cuffmerge. Heatmap results were visualized with the R package pheatmap. |

- ChIP-sequencing: Reads were mapped to the mouse reference genome (mm10) using bowtie2 mapper. SAMtools was used for filtering, sorting, and removing duplicates, and deepTools for generating coverage tracks.

- 4C-sequencing: Reads were pre-processed and mapped to the mm10 reference genome using BWA. The viewpoint and adjacent fragments 1.5 kb upstream and downstream were removed and a window of 10 fragments was chosen to normalize the data per million mapped reads (RPM).

- Capture-HiC: Read mapping and further filtering for each library were carried out separately using the HiCUP pipeline v0.8.3 with Bowtie2 v2.5.0 as the aligner and no size selection or filling. Binned and Knight-Ruiz (KR) normalized contact matrices from merged biological replicates were then generated with Juicer tools v1.22.0 for the region of interest. Subtraction and matrices visualization at 5kb resolution were done using a custom Python script via the FANC Python API (available on GitHub (https://github.com/mikstapes/JGlaser-etal_Dac1J/).

- DNA methylation bisulfite-cloning- sequencing data were analyzed with the biQ Analyzer software (version 2.02, 2008)

-CTCF motif analysis was performed using the FIMO (Find Individual Motif Occurrences), MEME suite 5.5.4 (https://meme-suite.org/meme/tools/fimo)

- Image analysis was done using Fiji (Image J, version 2.1.0/1.54f, 2020)

For manuscripts utilizing custom algorithms or software that are central to the research but not yet described in published literature, software must be made available to editors and reviewers. We strongly encourage code deposition in a community repository (e.g. GitHub). See the Nature Portfolio guidelines for submitting code & software for further information.

## Data

Policy information about availability of data

All manuscripts must include a data availability statement. This statement should provide the following information, where applicable:
- Accession codes, unique identifiers, or web links for publicly available datasets
- A description of any restrictions on data availability
- For clinical datasets or third party data, please ensure that the statement adheres to our policy

All datasets generated in this study have been deposited in the Gene Expression Omnibus (GEO) database and are accessible under accession code GSE246755 (containing SubSeries GSE246750, GSE246751, GSE246752, GSE246753, GSE246754). Previously published data used in this study are accessible under GSE185774, GSE116794 and GSE84795. Data were mapped to the Mus musculus mm10 genome.

## Research involving human participants, their data, or biological material

Policy information about studies with human participants or human data. See also policy information about sex, gender (identity/presentation), and sexual orientation and race, ethnicity and racism.

| Reporting on sex and gender | N/A |
|---|---|
| Reporting on race, ethnicity, or other socially relevant groupings | N/A |
| Population characteristics | N/A |
| Recruitment | N/A |
| Ethics oversight | N/A |

Note that full information on the approval of the study protocol must also be provided in the manuscript.

# Field-specific reporting

Please select the one below that is the best fit for your research. If you are not sure, read the appropriate sections before making your selection.

☒ Life sciences   ☐ Behavioural & social sciences   ☐ Ecological, evolutionary & environmental sciences

For a reference copy of the document with all sections, see nature.com/documents/nr-reporting-summary-flat.pdf

# Life sciences study design

All studies must disclose on these points even when the disclosure is negative.

| Sample size | Samples size was determined according to previous knowledge from the lab and the community. For expression analysis (RNA and protein), a minimum of 2 biological replicates were used per genotype and per developmental stage to unsure reproducibility. For phenotype, a larger number of samples were used to allow for detecting non-fully penetrant phenotype which would occur in 10% of the mice. IF, TEM, WISH, and LacZ staining experiments were performed from at least 3 independent biological mouse embryos. FISH experiments were performed from 2 independent biological mouse embryos. Skeletal preparations were performed using at least 3 independent biological |
|---|---|

mouse embryos or pups. Micro-CT was performed from 2 adult mice per genotype. Phenotypic evaluation of the limbs were performed using at least 60 limbs (15 animals) per experiment. Capture Hi-C experiments were performed using 2 or 3 biological replicates or 2 biological and 2 technical replicates per experiment. RNA-seq analyses were performed using 3 biological replicates. scRNA-seq experiments were performed from one biological replicate.

| | |
|---|---|
| Data exclusions | Samples/animals were included/excluded according to the genotype. |
| Replication | All experiments, except for scRNA-seq which used only 1 replicate per genotype, were replicated at least 2 times. |
| Randomization | When generating a mouse line, two founder animals for each mouse line were used for establishing line stock with variable intercrosses between single founder and 129sv wild-type animals. If more founders were generated, those two founder were randomly selected as long as they could transmit the mutation to the F1 (germline transmission).<br>For allocation of samples into experimental groups, this was done according to their genotype (WT or mutants). |
| Blinding | Investigators were not blinded during experiments, the data collection was performed according to the stage of each sample since mouse breeding and analysis required knowledge about the genotype at hand. |

# Reporting for specific materials, systems and methods

We require information from authors about some types of materials, experimental systems and methods used in many studies. Here, indicate whether each material, system or method listed is relevant to your study. If you are not sure if a list item applies to your research, read the appropriate section before selecting a response.

## Materials & experimental systems

| n/a | Involved in the study |
|---|---|
| ☐ | ☒ Antibodies |
| ☐ | ☒ Eukaryotic cell lines |
| ☒ | ☐ Palaeontology and archaeology |
| ☐ | ☒ Animals and other organisms |
| ☒ | ☐ Clinical data |
| ☒ | ☐ Dual use research of concern |
| ☒ | ☐ Plants |

## Methods

| n/a | Involved in the study |
|---|---|
| ☐ | ☒ ChIP-seq |
| ☒ | ☐ Flow cytometry |
| ☒ | ☐ MRI-based neuroimaging |

## Antibodies

| | |
|---|---|
| Antibodies used | primary antibody anti-MusD-Gag was a gift from the Heidmann lab (Ribet et al. 2007) (25058 J77)<br>primary antibody anti-Cleaved-Caspase 3 (rabbit polyclonal, Cell Signaling Technology Cat#9661, Asp175)<br>primary antibody anti-gamma-H2AX (rabbit monoclonal, Cell signaling Technology Cat#9718, Ser139 20E3)<br>secondary antibody anti-rabbit Alexa-fluorophore 488 ( Invitrogen #A11008)<br>secondary anntibody anti-rabbit Alexa-fluorophore 568 ( Invitrogen #A110042) |
| Validation | The anti-MusD-Gag was previously tested by the Heidemann lab (Ribet et al 2007). When we received it, we tested its reactivity on HeLa (human), HEK-293 (human) and Neuro2A (mouse) cells transfected with a MusD expressing plasmid (phCMV-RU5-musD-6 from Ribet et al. 2004, gift from the Heidemann lab). We tested the anti-MusD-Gag antiibody (25058 J77) alongside with a negative control (serum from the same rabbit, 25058 J0) which did not show any signal.<br>The anti-Cleaved-Caspase3 and anti-gamma-H2AX antibodies were purchased from Cell signaling where the antibodies were certified as validated for immuno-fluorescence application in mouse samples and were cited by respectively 99 and 16 scientific papers for this application. The antibodies were first tested in wild-type samples were no apoptosis and DNA damage are expected to validate the lack of background staining. |

## Eukaryotic cell lines

Policy information about cell lines and Sex and Gender in Research

| | |
|---|---|
| Cell line source(s) | We used mouse embryonic stem cells (mESCs) from 129/SvxC57BL/6J F1 hybrid (G4) background. These cells were obtained from Dr. Anders Nagy (George et al., 2007). mESCs from the Dac1J-129sv background were derived in house from bastocyst. CD1 and DR4 feeder cell lines, produced from CD1 and DR4 transgenic embryos, were used to culture the G4 cells. |
| Authentication | Genetically modified mESCs were used to produce embryos using tetraploid and diploid aggregation. Genotyping confirmed the presence of the desired mutation. The pluripotent state of the ESCs used was authenticated by generation of highly chimeric, germ-line transmitting mice.<br>CD1 and DR4 mouse embryonic fibroblast (feeder) cell lines were not authenticated. They were directly produced from mouse embryos originating from DR4 and CD1 mice crosses, respectively. |
| Mycoplasma contamination | All the cell lines were tested and were negative for mycoplasma contamination. |
| Commonly misidentified lines<br>(See ICLAC register) | No commonly misidentified cell lines were used. |

# Animals and other research organisms

Policy information about studies involving animals; ARRIVE guidelines recommended for reporting animal research, and Sex and Gender in Research

| Laboratory animals | Mice from CD1, C57BL/6J, 129s2/Sv, or 129s2/SvxC57BL/6J hybrid backgrounds were used in our study. Males and females from embryonic days E9.5-E12.5 and E18.5 were used in our experiments. Routine bedding, food, and water changes were performed. Mice were housed in a centrally controlled environment with a 12-h light/12-h dark cycle, temperature of 20-22.2 Celsius , and humidity of 30-50%. All animal experiments followed all relevant guidelines and regulations. |
|---|---|
| Wild animals | This study did not involved wild animals. |
| Reporting on sex | Sex was not part of the study design. |
| Field-collected samples | This study did not involve samples collected from the field. |
| Ethics oversight | All animal procedures were conducted as approved by the local authorities (LAGeSo Berlin) under license numbers G0243/18, G0176/19, and G0098/23. |

Note that full information on the approval of the study protocol must also be provided in the manuscript.

# Plants

| Seed stocks | *Report on the source of all seed stocks or other plant material used. If applicable, state the seed stock centre and catalogue number. If plant specimens were collected from the field, describe the collection location, date and sampling procedures.* |
|---|---|
| Novel plant genotypes | *Describe the methods by which all novel plant genotypes were produced. This includes those generated by transgenic approaches, gene editing, chemical/radiation-based mutagenesis and hybridization. For transgenic lines, describe the transformation method, the number of independent lines analyzed and the generation upon which experiments were performed. For gene-edited lines, describe the editor used, the endogenous sequence targeted for editing, the targeting guide RNA sequence (if applicable) and how the editor was applied.* |
| Authentication | *Describe any authentication procedures for each seed stock used or novel genotype generated. Describe any experiments used to assess the effect of a mutation and, where applicable, how potential secondary effects (e.g. second site T-DNA insertions, mosiacism, off-target gene editing) were examined.* |

# ChIP-seq

## Data deposition

☒ Confirm that both raw and final processed data have been deposited in a public database such as GEO.

☐ Confirm that you have deposited or provided access to graph files (e.g. BED files) for the called peaks.

| Data access links<br>*May remain private before publication.* | https://www.ncbi.nlm.nih.gov/geo/query/acc.cgi?acc=GSE246751 (reviewers token: qrofmegqzlmljgl) |
|---|---|
| Files in database submission | GSM7876362  ChIPseq_CTCF_WT_E10.5_FLHL_Rep1<br>GSM7876363  ChIPseq_CTCF_WT_E10.5_FLHL_Rep2<br>GSM7876364  ChIPseq_CTCF_Dac1J129sv_E10.5_FLHL<br>GSM7876365  ChIPseq_CTCF_Dac1JBl6_E10.5_FLHL<br>GSM8641800  ChIPseq_CTCF_WT_E11.5_FLHL_Rep1<br>GSM8641801  ChIPseq_CTCF_WT_E11.5_FLHL_Rep2<br>GSM8641802  ChIPseq_CTCF_Dac1J129sv_E11.5_FLHL_Rep1<br>GSM8641803  ChIPseq_CTCF_Dac1J129sv_E11.5_FLHL_Rep2<br>GSM8641804  ChIPseq_CTCF_Dac1JBl6_E11.5_FLHL_Rep1<br>GSM8641805  ChIPseq_CTCF_Dac1JBl6_E11.5_FLHL_Rep2<br>GSM8641806  ChIPseq_CTCF_5LTR-LacZ_E11.5_FLHL_Rep1<br>GSM8641807  ChIPseq_CTCF_5LTR-LacZ_E11.5_FLHL_Rep2 |
| Genome browser session<br>(e.g. UCSC) | https://genome-euro.ucsc.edu/s/Juliane%20glaser/Glaser_et_al_CTCF_ChIPseq |

## Methodology

| Replicates | n=2 biological replicates were used for WT samples. |
|---|---|
| Sequencing depth | At least 20 millions read per samples were sequenced, using single-end sequencing. |
| Antibodies | CTCF antibody (C15410210; Diagenode) |

| | |
|---|---|
| Peak calling parameters | Not applicable. |
| Data quality | Not applicable. |
| Software | Reads were mapped to the mouse reference genome (mm10) using bowtie2 mapper. SAMtools was used for filtering, sorting, and removing duplicates, and deepTools for generating coverage tracks. |

