## [Peer Review File · Nature Genetics]

Enhancer adoption by an LTR retrotransposon generates viral-like particles causing developmental limb phenotypes

Corresponding Author: Professor Stefan Mundlos

Version 0:

Decision Letter:

21st Jun 2024

Dear Stefan,

Your Article, entitled "Enhancer adoption by an LTR retrotransposon generates viral-like particles causing developmental limb phenotypes", has now been seen by 4 referees. You will see from their comments below that while they find your work of interest, some important points are raised. We are interested in the possibility of publishing your study in Nature Genetics, but would like to consider your response to these concerns in the form of a revised manuscript before we make a final decision on publication.

Reviewer #1 notes that this work partially recapitulates what was described in a 2005 PNAS paper. The new data, while intriguing, point to a mechanism that may not be generalizable.

Reviewer #2 is clearly more positive. While they acknowledge that this example may not be common (this is unclear at this stage), they find it novel and interesting. Importantly, they highlight that you have not fully examined the potential contribution of the gag/gag-pol polyprotein.

Reviewer #3 is generally positive about this work but would like you to investigate the role (and underlying mechanisms) of apoptosis in more detail.

Reviewer #4 is also impressed by the work and doesn't raise any major concerns. Still, there are some suggestions for improvement.

We invite you to revise your manuscript taking into account all reviewer comments. Please highlight all changes in the manuscript text file. At this stage we will need you to upload a copy of the manuscript in MS Word .docx or similar editable format.

We are committed to providing a fair and constructive peer-review process. Do not hesitate to contact me if there are specific requests from the reviewers that you believe are technically impossible or unlikely to yield a meaningful outcome. I would be happy to discuss the reviewers' comments in detail to facilitate the revision process.

*2) If you have not done so already please begin to revise your manuscript so that it conforms to our Article format instructions, available

[here](http://www.nature.com/ng/authors/article_types/index.html).

*3) Include a revised version of any required Reporting Summary: <https://www.nature.com/documents/nr-reporting-summary.pdf>

Please be aware of our [guidelines on](https://www.nature.com/nature-research/editorial-policies/image-integrity)

digital image standards.

Link Redacted

We hope to receive your revised manuscript within 6 months. If you cannot send it within this time, please let us know.

Nature Genetics is committed to improving transparency in authorship. As part of our efforts in this direction, we are now requesting that all authors identified as 'corresponding author' on published papers create and link their Open Researcher and Contributor Identifier (ORCID) with their account on the Manuscript Tracking System (MTS), prior to acceptance. ORCID helps the scientific community achieve unambiguous attribution of all scholarly contributions. You can create and link your ORCID from the home page of the MTS by clicking on 'Modify my Springer Nature account'. For more information please visit please visit www.springernature.com/orcid.

Sincerely,

Tiago

Tiago Faial, PhD
Chief Editor
Nature Genetics
<https://orcid.org/0000-0003-0864-1200>

Reviewers' Comments:

Reviewer #1:

Remarks to the Author:

The manuscript by Glaser et al sets out to characterise the molecular basis of a spontaneously occurring mouse mutation Dactylaplasia, first mapped in 1995. Much of the early part of this study uses modern molecular sequencing techniques to confirm data from the 2005 study by Kano (PNAS 104 (48) 19034-19039)- that phenotypes results from insertion of a MusD TE, that the active (phenotype causing) allele is unmethylated while that from C57 is heavily methylated, that MusD elements are expressed ectopically at the apical ectodermal ridge of limb buds and in the mutants FGF8 expression in the AER is downregulated. The conclusion from the Kano paper was that decreased expression of Fgf8 in the Dactylaplasia mouse is thought to be caused by degeneration of the AER, however now, given that the insertion is close to FGF8 it is not unreasonable to suspect that chromatin conformation and regulation of FGF8 is being disrupted.

In the current study, nice Hi-C data shows an ectopic loop from the insertion into the neighbouring TAD which is absent in the inactive methylated allele. KI of the 5'LTR driving LacZ recapitulates this chromatin loop but not the digit phenotype showing that this alone isn't causative. 4Cseq does show peaks of increased interactions between the insertion and the region containing Fgf8 enhancers (but not always exactly matching the enhancers) and lacZ staining of KIs show an FGF8 like expression pattern, demonstrating the adoption of endogenous enhancers.

The phenotype of mutant embryos is similar to that obtained in late stage surgical AER removal in chicken and experiments in carefully staged embryos show that regulation of FGF8 (and other AER markers) is the same as in wildtype at E9.5 and E10.5 and only differ in E11.5 embryos, suggesting a rapid and late loss of AER cells. This was confirmed by IF for cleaved Caspase3, the presence of apoptotic bodies in EM and upregulation of cell death genes in RNA-seq). By making a series of KIs carrying mutant MusD-Dac1J sequences (2 without functioning pol genes, and one with a disrupted primer binding site) all of which largely rescue the mutant phenotypes, the authors are able to demonstrate that the observed cell death results from the replicative viral like particles, perhaps causing genome instability. However, this appears to be tissue-dependent, as a wildtype KI in the adjacent TAD under the control of Lbx1 leads to Gag protein expression but no apparent cell death. So, despite the possibility raised in the manuscript I'm unconvinced how generally applicable this mechanism may prove to be.

I'm not sure what more information is added by the insertion of the B-globin driven LacZ than is shown by the 5'LTR lacZ experiment?

Some of the figures suggest that the mutant hand plates are already an abnormal size and shape by E11. Can the authors comment on how/ when the observed differences fit with their molecular timeline?

The use of Dac $+/+$ and $+/-$ to reflect the presence or absence of the insertion is confusing (and in mouse nomenclature terms the wrong way round) and should be changed.

In the figures- the known Fgf8 enhancers are in two different shades of blue- but what this signifies is never explained.

Reviewer #2:

Remarks to the Author:

It has long been recognized that transposable elements (TEs) can be deleterious when their insertion causes genome instability or gene disruption. However, it remains unclear whether products of an active TE insertion per se can play pathological roles in addition to its genomic context. In this manuscript, Glaser et al. addresses this intriguing question with an affirmative answer. Specifically, they dissected the etiology of mouse dactylaplasia limb malformation caused by Dac1J allele – an insertion of an evolutionarily young TE MusD (type D-related mouse provirus-like) in the intergenic region near Fgf8. By examining the molecular role of MusD-Dac1J at various levels, they present multiple lines of evidence that MusD viral proteins are responsible for the dactylaplasia phenotype. The various animal models (a real Tour de Force) clearly corroborate the main claims and give validity for this phenomenon being a general mechanism in the certain developmental niche. This work is important due to how overlooked such a mechanism is, based on literature surrounding similar genetic elements. Indeed, the opposite is typically reported, where a regulatory information embedded in the LTR of a TE influences nearby gene expression, splicing, etc.

The highlight of this study is that it presents a compelling example of how TE per se, in a pathological context, affects mammalian development without being solely functionally dependent of its nearby genes. The icing on the case is the discovery of reverse transcription of MusD-Dac1J being the primary cause of the dactylaplasia phenotype, which broadens our understanding of how TE lifecycle impacts normal biology. It is unclear how often these criteria might be found in the human population, but this mechanism demonstrates a novel way of approaching the complex relationship between TE and host.

As much as I enjoyed reviewing the manuscript, my enthusiasm for the work is somewhat dampened by an imprudent mechanistic interpretation of how different retroviral components contribute to the dactylaplasia etiology. Specifically, while the study provides solid evidence that disrupting reverse transcription of MusD-Dac1J significantly rescues the dactylaplasia phenotype, it does not examine the functional contribution of gag/gag-pol polyprotein to the phenotype at all with the rigorous loss-of-function experiments done on reverse transcription. These results are insufficient to support one of the main propositions made in the manuscript that “full Dac1J-derived MusD VLPs with replication potential are required to cause Dactylaplasia phenotype.” (line 287-288) – both mice with full length and gag-pol only MusD-Dac1J show dactylaplasia phenotype, though to different degrees. Nevertheless, the data presented are of high quality, and the manuscript is generally well written. I understand the difficulties that associate with the often-multifaceted roles of TEs in physiological context, and the presented discovery and characterization are overall sufficient to warrant publication. Therefore, I would consider the manuscript ready for publication with a minor revision that addresses the data interpretation issue along with few suggestions to improve the manuscript.

Major comments:

1. In line 110-111, the MusD insertion is 7.4 kb while the deletion made in figure 1e and 1f is a 16 kb deletion. Since this mechanism relies on the local epigenetic environment there might be cause for concern that the surrounding region has been disturbed. I think my minor comment 2 and 4 about Mdac and local epigenetic profiling might be useful here. Also, it might be informative to do the same experiment post deletion. Another way of asking this would be to add two additional bisulfite cloning result. These two additional panels wouldn't perfectly match up to the current panels (LTR promoter region) but they would clue us into what is happening before > during (B6) > during (129) > after (deletion) of the insertion.
2. In figure 2d and 2e, only expression Fgf8 is shown. What about other genes that are proximal to Dac1J? It would be helpful to show expression profiles of nearby genes, especially those that are topologically associated with Dac1J.
3. In figure 3h, does the minimal β -globin promoter also introduce ectopic TAD similar to that of MusD 5' LTR? If not, does that mean that adaptation of nearby enhancer activity by 5' LTR is independent of chromatin topology?
4. In figure 5d and line 281, could the other 9 complete MusD copies be involved instead of ETns? Furthermore, are there other MusD copies that potentially express pol? It would be helpful to show the expression (or non-expression, if that is the case) of these copies. This is very relevant as it may explain the partially rescued phenotype and strengthen the claim that replication-competent VLPs are responsible for the etiology.
5. Main text line 286-288: “Altogether, the partial rescues of MusD-Dac1J element with disrupted Pol protein or tRNA binding site suggest that full Dac1J-derived MusD VLPs with replication potential are required to cause actylaplasia phenotype.” – this is not entirely correct. If Dac1J is solely responsible for the RT activity of MusD as concerned above, then the result does not suggest that intact VLPs are required for the dactylaplasia phenotype. Instead, the result suggests that reverse transcription of MusD accounts for most of the dactylaplasia phenotype, while GAG solely, although to a much lesser extent, still causes missing digits. I feel strongly that the distinction between VLP formation and reverse transcription needs to be addressed more carefully in this context, as study on HIV has shown that VLPs can form effectively with 80% truncated gag (Accola et al., J Virol. 2000).
6. Main text line 315-318: “Overall, our results indicate that the cell death response to MusD VLPs during development can

affect embryonic cells differently and highlight that mouse post-implantation development can also proceed with the presence of endogenous retrovirus capsid.” – same as above, the manuscript does not present evidence supporting that VLPs are responsible for cell death. The reviewer feels strongly that a knock-in experiment with gag mutated Dac1J is needed to make such statement.

7. General: the methylation status between 129 and B6 is very interesting and perhaps outside the scope of the present work. It would at least be helpful to speculate on what might be causing such a drastic discrepancy between strains in the discussion and whether something akin to the IAP element within the agouti locus (differential methylation pattern/metastable epiallele) could possibly be at play. Is it possible “stress” or some “environmental” biosensor is triggered?

Minor comments:

1. In figure 1a, what is the unlabeled region just upstream of the 3' LTR? Is it supposed to be Env? If so, it might be helpful to indicate that this is mutated or missing.
2. In line 103-104, the manuscript mentions “However, Dactylaplasia is subject to polymorphisms between mouse strains (Fig. 1c) as the MusD-Dac1J 5'LTR promoter exhibits differential DNA methylation in the limb21 (Fig. 1d).” For better readability, it would be helpful to expand the background a bit more by briefly introduce that methylation of Dac1J is dependent on the presence of MDac (Kano et al., Proc. Natl. Acad. Sci.).
3. In figure 1b, what strain is “wildtype” specifically?
4. In figure 1d, would it be possible to investigate the methylation status around the area where the MusD insertion would have been in the WT? Perhaps this could be a clue about the local epigenetic environment “pre-insertion”. Perhaps this can be done for both B6 and 129 just to see whether there was something different/unique about this location in these strains.
5. In figure 2a, when printed, the details of the limb bud are difficult to resolve. I think enlarging just this side of the image would help.
6. In figure 2b, similarly, I completely missed the “immune cells” population on my first look of this image. A little rearranging and enlarging will be helpful.
7. In line 122, AER has already been defined. No need to have full name here.
8. In figure 2c, text on the right is very hard to see. Please consider larger font. Keep consistent with the rest of the panel (same font and size).
9. In figure 2c and line 121-133, how would these populations change if you were to compare WT to another WT? What is the individual variance among different embryos? (Replicates?) I am not familiar with the nuances of single cell analysis and just wonder whether a replicate would demonstrate a similar change. Alternatively, is it possible to present the data without the Mesenchyme, as this signal dominates the profile, making the changes you are interested small and hard to visualize/interpret.
10. In line 139, was “E10.25” intentional?
11. In line 144, I was under the impression that mapping to a specific TE using single cell RNA-seq was difficult, but given the age of the insertion and about 100 nearly identical copies, is unique mapping even possible? Is this at the subfamily level?
12. In line 156, minor type? I think the word “and” was meant to mean something else. “are”?
13. In figure 3a, “Fgf8 enhancers” text is way too small.
14. In line 168, can you expand a little bit on how increased insulation is determined and what this might impact? I think indicating how this is determined on figure 3e and 3f would help people who are not too familiar with these types of graphs/outputs.
15. In figure 3e, Dac1J locus shows increased interaction asymmetrically spanning over Lbx1 and Fgf8 comparing with wildtype, while the same cannot be observed with 5' LTR-LacZ. Do the authors have an interpretation for this? Personally, I think it is interesting and worth mentioning briefly in the manuscript.
16. Figure 4d legend, shouldn't it be “Zoom-in view (right)”?
17. In figure 5a, would it be better to title this something like “MusD endogenous retroviral cycle” to distinguish this from a current “viral” cycle of an exogenous retrovirus? Also, the font for LTR, gag, pro, etc... are quite hard to see in the current version of the figure. It might be a quality issue when the figures were submitted but there is an overall drop in quality of the various texts (some letters are thinner or seem to be missing “ink”, if that makes sense).
18. In figure 6, do Dac1J KI/KI @ Lbx1-TAD embryos develop to full term? Do they show any phenotype?
19. In extended data figure 6, typo in title “Extebbed”.
20. In discussion, is there any link between this phenotype and the human condition, or would this be specifically associated with the 129 mouse line?

Reviewer #3:

Remarks to the Author:

In this manuscript, Glaser et al. characterize genetic determinants of limb malformation in a dactylaplasia mouse model and highlight the importance of transposon control during development. The authors show that in the 129sv mouse strain with the dactylaplasia phenotype (Dac1J), a MusD ERV element inserted at the Fbxw4-Fgf8 locus escapes DNA methylation and is activated in the apical ectodermal ridge (AER) cells via hijacking local cis-regulatory elements. The MusD activity does not appear to affect Fgf8 expression but instead leads to apoptosis of AER cells, which authors suggest is due to viral-like particles (VLPs) produced by the MusD. Consequently, deletion of the entire MusD element, mutation of its coding sequence or of the tRNA primer-binding site, all result in rescue of the limb phenotype.

Overall, this is a well-executed study, with results that are of interest to the developmental biology and transposable element fields. The results are clearly presented in figures that make understanding of the manuscript straightforward. The mouse genetics approach to the dissection of the MusD element viral cycle is very elegant, and clearly implicates ERV proviral sequences in mediating the limb phenotypes. I think this latter point is the most novel aspect of the work (rather than

adoption of local enhancers by the TE promoters, which is not surprising, and in fact a basis of many sensors of enhancer activity).

While I am generally positive about the study and its suitability for Nature Genetics readership, one quite unsatisfying point in the manuscript is its lack of insight into the mechanism behind the questions of why the MusD-expressing AER cells are undergoing apoptosis and why is the MusD activity temporarily decoupled from apoptosis (seems to occur at least a couple of days prior to the cell death). Given that loss of AER cell population due to MusD activity is the cornerstone of the manuscript, some insight into these questions would greatly strengthen the study. For example, at which point (if at all) following MusD expression do authors see DNA damage or innate immunity activated in AER? Is it restricted to AER cells within the developing limb? To what extent can the kinetics of the viral life cycle explain the observed delay? Some simple stainings may provide interesting, even if correlative, clues as to the underlying mechanism. In addition, there are several other points that deserve further study and clarification.

Major comments:

-The authors focus on *Fgf8*, but there is another gene closer to the MusD element, named *Fbxw4*, for which the impact of the MusD (*Dac1J*) presence has not been addressed, but which has been previously implicated in dactylaplasia-like phenotypes. Please confirm that expression of *Fbxw4* is not affected by MusD activity.

-Figure 1E: Why did the authors delete ~8 extra kb (3.3 kb upstream and 5.3 kb downstream) of the MusD element? Are these extra deleted DNA base pairs affecting the phenotype? Please show as a supplementary figure the epigenomic landscape (e.g. ATAC-seq, histone modifications) of the deleted genomic region to visualize if any cis-regulatory elements have been inadvertently deleted. WT data should suffice.

-Which markers were used to call the different cell clusters in Figure 2B? Please provide more details, especially for the AER cells. Which markers and specific expression levels are used as cut-off for considering a cell AER at the different stages?

-The authors should exclude a possibility that the depletion of the AER population at E11.5 is the consequence of a consistently lower *Fgf8* expression at E9.5 and E10.5. For example, in Figure 2D *Fgf8* expression appears lower in *Dac1j+/-* than in WT, though it could be because some cells have already been lost. Please show *Fgf8* (and *Fbxw4*) expression in MusD-positive cells using Featureplot and violin plots. This could help visualize the results in a more quantitative manner.

-How confident are the authors that the transcription in Fig. 2E and F is mostly (or solely) due to activity of the MusD element at the *Fgf8* locus? Is there evidence that other MusD elements are also active in these cells? Transposon mapping at the single insertion level is challenging with the 10x data.

-In Extended Data Figure 3 the authors perform bulk RNA-seq concluding that no other gene at the locus is affected. Showing browser captures is not enough to claim this. The authors should perform formal quantitative differential gene expression analysis to explore potential gene expression changes driving AER loss within and outside the TAD.

Minor comments:

-Line 76: Current reference 18 does not show expression at E3.5. Please add this reference too PMID: 2580305.

-Figure legend 1D, typo: promotor should be promoter.

-In Figure 3C and D, the authors show CTCF ChIP-seq. What is the numerical scale of the zoom-ins? It seems that WT and *Dac1j+/-* are displayed at different scales favoring a larger CTCF peak at the MusD element.

-MusD VLPs do not bud off the cells due to lack of envelope. Please indicate it somewhere in the text, as some readers may expect these particles to act in trans.

-The authors did not provide a token for the reviewers to assess the quality of the sequencing data deposited at GEO.

Reviewer #4:

Remarks to the Author:

Glaser et al investigated the mechanism by which a ERV insertion (*MusD-Dac1J*) near *Fgf8* gene causes Dactylaplasia, a genetic limb malformation. The study rejected the hypothesis that *Dac1J* disrupts *Fgf8* expression; rather, the authors discovered a novel mechanism where ERVs have the capability to take on the surround epigenomic landscape and express viral-like particles, which lead to cell death. The authors systematically show the role of *DAC1J* in limb malformation through the co-expression of *Fgf8* with *DAC1J* and the changes in TAD structure in mutant despite observing no transcriptional changes associated with this change. Since *Dac1J* has the machinery for autonomous proviruses they investigate viral-like particles (VLPs) and the potential implications. They found VLPs were associated with cell death and using mutated *DAC1J* they identified POL protein as a key driver in digit phenotype. They generalized their finding to show that MusD insertions can adapt the expression pattern of nearby genes using LTR-LacZ constructs.

Overall, reading this manuscript has been a treat for me. The authors presented an excellent scientific design and addressed many possible alternatives thoroughly. The authors not only made a novel discovery, but also defined a rigorous paradigm for testing hypothesis using genetics and genomics approaches. I have a few suggestions to hopefully help enhance the rigor of the science, which is already quite strong.

1. Because this study investigates a young transposable element that has multiple copies in the genome and is polymorphic, any experiment that requires sequence specificity (CRISPR guide RNA design, bisulfite sequencing primer design, RNA-seq read mapping, etc.) needs to be carefully designed and scrutinized. The authors should provide a clear description of these designs, including analysis of off-target effect. How do they ensure that they are measuring/targeting the exact *Dac1J* copy that they intend to measure/target?

2. The differential methylation of *Dac1J* in C57BL/6 and 129sv is a key evidence for the claim that unmethylated *Dac1J* is required for the phenotype. However, to establish a real causal relationship, DNA methylation could be manipulated directly

in C57BL/6 and 129sv to rule out other factors associated with their genetic background. Later in Fig 5b, the methylation status of the mutants should be provided. The assumption is that they are not methylated, but it should be confirmed.

3. To generalize the finding, the authors inserted the 5'LTR-LacZ construct into the Lbx1, Shh, and Sox9 regulatory domains and showed that the LacZ expression recapitulated those genes' expression. It is important to measure 3D genome changes in these contexts, in order to generalize the principle. Does the 5'LTR-LacZ insertion also form an ectopic chromatin loop like the 5'LTR-LacZ insertion upstream of Fgf8 gene?
4. Considering the limit number of cells be captured and presented using scRNA-seq and its relatively large variations in detecting cell numbers, it would be better if the authors can estimate the number of AER cells using another approach such as FACS or immunostaining. This could provide more direct and quantitative evidence on AER cell apoptosis at different time points.
5. According to scRNA-seq data, AER cells only present a very small population at E10.5 and E11.5, which indicate that most of the CHIP-seq and cHi-C signal are captured from other cell types (i.e., mesenchymal cells), which means that MusD-Dac1J are likely to be epigenetically activated in those cells. However there is no strong expression of MusD-Dac1J transcripts as well as production of VLP particles in other cell types. Are there any explanations for that?
6. Extended data Fig. 2 showed a large cell number decrease for immune cells at E10.5 and cell number decrease for DV ectoderm at E11.5. Why cell number decreased in Dac1J +/+? Are those cells affected by the Dac1J insertion, although Dac1J did not express in those cells?
7. Fig. 3a-b showed the Hi-C data from wild-type and Dac1J +/+ E11.5 mouse limb buds. However, the Fig.3c-d used the ChIP-seq tracks from E10.5 mouse limb buds. Please used the ChIP-seq tracks from same stages as the Hi-C data to support the TADs.
8. To measure the methylation status of 5' LTR promoter from Dac1J, the authors measured 19 CpGs of 5' LTR in Fig. 1, 21 CpGs of 5' LTR in Extended Data Fig. 4 and 7 CpGs in Extended Data Fig. 8. Please make these consistent.
9. The proper name of the B6 mice is "C57BL/6", not "C57Bl6". Also, "wild type" is used rather leisurely throughout the manuscript. Specific genetic background should be provided.

Version 1:

Decision Letter:

Our ref: NG-A65483R

24th Jan 2025

Dear Stefan,

Thank you for submitting your revised manuscript entitled "Enhancer adoption by an LTR retrotransposon generates viral-like particles causing developmental limb phenotypes" (NG-A65483R). It has now been seen by the original referees and their comments are below. The reviewers find that the paper has improved in revision, and therefore we'll be happy in principle to publish it in Nature Genetics, pending minor revisions to satisfy the referees' final requests and to comply with our editorial and formatting guidelines.

We will now be performing detailed checks on your paper and will send you a checklist detailing our editorial and formatting requirements soon. Please do not upload the final materials and make any revisions until you receive this additional information from us.

Thank you again for your interest in Nature Genetics. Please do not hesitate to contact me if you have any questions.

Congratulations!

Sincerely,

Tiago

Tiago Faial, PhD
Chief Editor
Nature Genetics
<https://orcid.org/0000-0003-0864-1200>

Reviewer #1 (Remarks to the Author):

In this revised version of the manuscript by Glaser et al., the authors have added a lot of new experiments including a careful analysis of the timing and order of the pathogenic events they observe within the mutant limb bud, requiring some very pretty HCR RNA fluorescent in situ experiments. There are also several new ChIP-seq experiments and the generation and analyses of a number of new mutant knock-in mouse lines.

All of this this data is compelling and greatly strengthens their elegant study.

My only point is that there is a missing the word, likely loop, in line 373.

Reviewer #2 (Remarks to the Author):

We would like to commend Glaser et al. for addressing every point of our review and thoughtful consideration in their replies. Quite a few responses went above and beyond what we had requested and appreciate how they made this second review very clear and concise. The mutated gag knock-in is impeccable and conclusion is eloquently conveyed. Overall, I believe the revision is more than acceptable and we recommend this version of the manuscript for publication. I only have a very minor concern regarding the readout of Fgf8 expression using bulk RNA-seq and scRNA-seq/WISH, what could be the possible cause of the observed discrepancy? I do not think this will undermine any of the conclusions the authors are trying to draw in the manuscript but believe it requires some clarification in the final version. This updated version is much more complete and the great efforts the authors made have elevated the importance and impact of this manuscript.

Reviewer #3 (Remarks to the Author):

I am satisfied with the authors' response to my comments and recommend publication. Minor point: Could authors show a more unbiased analysis of differentially expressed genes from bulk RNA-seq in addition to a heatmap of pre-selected genes in Extended Data Fig. 8?

Reviewer #4 (Remarks to the Author):

I thank the authors for addressing my comments. I don't have further critiques.

Point-by-point response to the referees:

We sincerely thank the reviewers for their fair evaluation of our work. We were pleased to read the positive comments from the reviewers and feel that the requested changes are constructive and help to improve our manuscript. We have now performed additional experiments that we hope will address their concerns. We provide here a point-by-point response to their questions/comments. Below, are the original comments **in black** and our responses **in blue**. Relevant changes or additions are highlighted **in yellow** in the revised manuscript.

Reviewer #1:

Remarks to the Author:

The manuscript by Glaser et al sets out to characterise the molecular basis of a spontaneously occurring mouse mutation Dactylaplasia, first mapped in 1995. Much of the early part of this study uses modern molecular sequencing techniques to confirm data from the 2005 study by Kano (PNAS 104 (48) 19034-19039)- that phenotypes results from insertion of a MusD TE, that the active (phenotype causing) allele is unmethylated while that from C57 is heavily methylated, that MusD elements are expressed ectopically at the apical ectodermal ridge of limb buds and in the mutants FGF8 expression in the AER is downregulated. The conclusion from the Kano paper was that decreased expression of Fgf8 in the Dactylaplasia mouse is thought to be caused by degeneration of the AER, however now, given that the insertion is close to FGF8 it is not unreasonable to suspect that chromatin conformation and regulation of FGF8 is being disrupted.

The paper from *Kano et al.* published in 2007 reported that the *Dac* allele responsible for the Dactylaplasia phenotype is an LTR retrotransposon from the MusD family, and it described the epigenetic regulation of this element by the modifier of *Dac* (*Mdac*) located in a 9.4Mb interval on chromosome 13. However, the molecular mechanism behind the phenotype was still unknown and the paper by *Kano et al.* does not rule out that the *Dac* insertions affect gene regulation.

While our work corroborates some of the conclusions drawn by *Kano et al.*, we were able to show for the first time the mechanism by which MusD insertion at the *Fgf8* locus causes a developmental phenotype. First, we revealed co-expression of *Dac1J* and *Fgf8* in the AER cells that we demonstrate is due to enhancer adoption. Importantly, we show that the enhancer adoption mechanism happens at other developmental loci, demonstrating a new mechanism of how transposable elements interact with host genomic landscapes. Second, we show that the *Dac1J* insertion does not affect local gene regulation but we report the production of viral-like particles in the AER, associated with apoptotic cell death. Third, we carefully dissected which part of the MusD life cycle can cause a phenotype, highlighting how transposable elements' lifecycle can impact normal biology. We thus believe that our findings are novel compared to the previous work from *Kano et al.*

In the current study, nice Hi-C data shows an ectopic loop from the insertion into the neighbouring TAD which is absent in the inactive methylated allele. KI of the 5'LTR driving LacZ recapitulates this chromatin loop but not the digit phenotype showing that this alone isn't causative. 4Cseq does show peaks of increased interactions between the insertion and the region containing *Fgf8* enhancers (but not always exactly matching the enhancers) and lacZ staining of KIs show an FGF8 like expression pattern, demonstrating the adoption of endogenous enhancers.

The phenotype of mutant embryos is similar to that obtained in late stage surgical AER removal in chicken and experiments in carefully staged embryos show that regulation of FGF8 (and other AER markers) is the same as in wildtype at E9.5 and E10.5 and only differ in E11.5 embryos, suggesting a rapid and late loss of AER cells. This was confirmed by IF for cleaved Caspase3, the presence of apoptotic bodies in EM and upregulation of cell death genes in RNA-seq). By making a series of KIs carrying mutant MusD-Dac1J sequences (2 without functioning pol genes, and one with a disrupted primer binding site) all of which largely rescue the mutant phenotypes, the authors are able to demonstrate that the observed cell death results from the replicative viral like particles, perhaps causing genome instability. However, this appears to be tissue-dependent, as a wildtype KI in the adjacent TAD under the control of *Lbx1* leads to Gag protein expression but no apparent cell death. So, despite the possibility raised in the manuscript I'm unconvinced how generally applicable this mechanism may prove to be.

The mechanism that we are referring to is the enhancer adoption of the MusD 5'LTR (see discussion lines 363-365 and 381-382). We showed that this is not restricted to the *Fgf8* locus, suggesting that any unmethylated MusD can be faithfully expressed according to the regulatory domain where it is located. As written in lines 83-84, we believe that "enhancer adoption by LTR elements could be a more common phenomenon responsible for developmental malformation.". Regarding cell death, we indeed did not detect it in response to VLP production in the muscle progenitors of the developing limb when MusD expression was driven by *Lbx1* regulatory elements.

This might have different explanations, such as tissue-specificity, dose-dependent effect, or might depend on the surrounding genomic loci (as discussed in the discussion line 425). Yet, we showed that the cell death mechanism is not restricted to the AER cells as observed in the branchial arches, suggesting that other developmental tissue can be sensitive to VLPs.

I'm not sure what more information is added by the insertion of the B-globin driven LacZ than is shown by the 5'LTR lacZ experiment?

The β globin-LacZ construct allowed us to conclude that the 5'LTR-*Dac1J* promoter acts as a sensor and can capture the regulatory activity surrounding its insertion, just like a minimal promoter does. This is a property that, to our knowledge, was not known for retrotransposons promoters. This construct is important to decipher whether the 5'LTR promoter can drive specific expression in embryos or capture the expression of the surrounding regulatory activity. Here, we show that it behaves exactly like a sensor.

Some of the figures suggest that the mutant hand plates are already an abnormal size and shape by E11. Can the authors comment on how/ when the observed differences fit with their molecular timeline?

We show that the *Dac1J* mRNA is expressed in the AER from E9.5 (Figure 2) and observed MusD-Gag proteins and intact AER at E10.5 but apoptosis at E11.0 (Figure 4). In a new set of experiments, we now show increased DNA damage in *Dac1J/Dac1J* mutants compared to wild-type AER cells from E10.5 which is accentuated at E11.0 (Figure 4). We have also performed fluorescent in situ hybridization (FISH) data measuring *Fgf8* mRNA at four precisely staged embryos (E10.0, 30 somites; E10.5, 35 somites; E11.0, 40 somites; and E11.5, 46 somites; see new Extended data Figure 7c-d). This confirmed that *Fgf8* expression and AER cells are mostly not affected until E10.5. At E11.0, we observed the signature of apoptotic cells (fragmented nuclei), which was also observed with electron microscopy (Figure 4). We thus conclude that AER cells in *Dac1J/Dac1J* embryos undergo DNA damage in response to VLPs production at E10.5, and this likely leads to apoptotic cell death by E11.0, leading to the elimination of most of the AER cells by E11.5 (see schematic Figure 4h and below).

The use of *Dac* *+/+* and *+/-* to reflect the presence or absence of the insertion is confusing (and in mouse nomenclature terms the wrong way round) and should be changed.

This has now been changed in the manuscript text and figures to *Dac1J/WT* for the heterozygotes and *Dac1J/Dac1J* for the homozygotes.

In the figures- the known *Fgf8* enhancers are in two different shades of blue- but what this signifies is never explained.

The dark blue bars represent enhancers that have activities in the AER and the light blue bars represent enhancers that have any other *Fgf8* activity (as published in *Marinic et al. 2013*). We have now added an explanation in the legend of Figure 3a (“Published *Fgf8* enhancers are indicated in light blue and dark blue (specific AER enhancers)”).

Reviewer #2:

Remarks to the Author:

It has long been recognized that transposable elements (TEs) can be deleterious when their insertion causes genome instability or gene disruption. However, it remains unclear whether products of an active TE insertion per se can play pathological roles in addition to its genomic context. In this manuscript, Glaser et al. addresses this intriguing question with an affirmative answer. Specifically, they dissected the etiology of mouse dactylaplasia limb malformation caused by *Dac1J* allele – an insertion of an evolutionarily young TE MusD (type D-related mouse provirus-like) in the intergenic region near *Fgf8*. By examining the molecular role of MusD-*Dac1J* at various levels, they present multiple lines of evidence that MusD viral proteins are responsible for the dactylaplasia phenotype. The various animal models (a real Tour de Force) clearly corroborate the main claims and give validity for this phenomenon being a general mechanism in the certain developmental niche. This work is important due to how overlooked such a mechanism is, based on literature surrounding similar genetic elements.

Indeed, the opposite is typically reported, where a regulatory information embedded in the LTR of a TE influences nearby gene expression, splicing, etc.

The highlight of this study is that it presents a compelling example of how TE per se, in a pathological context, affects mammalian development without being solely functionally dependent of its nearby genes. The icing on the case is the discovery of reverse transcription of MusD-Dac1J being the primary cause of the dactylaplasia phenotype, which broadens our understanding of how TE lifecycle impacts normal biology. It is unclear how often these criteria might be found in the human population, but this mechanism demonstrates a novel way of approaching the complex relationship between TE and host.

As much as I enjoyed reviewing the manuscript, my enthusiasm for the work is somewhat dampened by an imprudent mechanistic interpretation of how different retroviral components contribute to the dactylaplasia etiology. Specifically, while the study provides solid evidence that disrupting reverse transcription of MusD-Dac1J significantly rescues the dactylaplasia phenotype, it does not examine the functional contribution of gag/gag-pol polyprotein to the phenotype at all with the rigorous loss-of-function experiments done on reverse transcription. These results are insufficient to support one of the main propositions made in the manuscript that “full Dac1J-derived MusD VLPs with replication potential are required to cause Dactylaplasia phenotype.” (line 287-288) – both mice with full length and gag-pol only MusD-Dac1J show dactylaplasia phenotype, though to different degrees. Nevertheless, the data presented are of high quality, and the manuscript is generally well written. I understand the difficulties that associate with the often-multifaceted roles of TEs in physiological context, and the presented discovery and characterization are overall sufficient to warrant publication. Therefore, I would consider the manuscript ready for publication with a minor revision that addresses the data interpretation issue along with few suggestions to improve the manuscript.

We thank the reviewer for the thoughtful comments. As described in detail later in response to the comments, we have now generated two new mutants affecting the Gag of the MusD-Dac1J and changed some of the conclusions/interpretations accordingly. We hope that this will address the reviewer’s concerns.

Major comments:

1. In line 110-111, the MusD insertion is 7.4 kb while the deletion made in figure 1e and 1f is a 16 kb deletion. Since this mechanism relies on the local epigenetic environment there might be cause for concern that the surrounding region has been disturbed. I think my minor comment 2 and 4 about Mdac and local epigenetic profiling might be useful here. Also, it might be informative to do the same experiment post deletion. Another way of asking this would be to add two additional bisulfite cloning result. These two additional panels wouldn’t perfectly match up to the current panels (LTR promoter region) but they would clue us into what is happening before > during (B6) > during (129) > after (deletion) of the insertion.

We initially generated a bigger deletion as sgRNAs further away had a better score than the ones closer to the *Dac1J* insertion. However, we now successfully generated a new deletion which is ~8.3kb and only 446 and 101 bp upstream and downstream of the MusD-*Dac1J* insertion, respectively. This deletion still rescues the phenotype (see below and new Figure 1e). This allowed us to perform CpG DNA methylation with the few CpG present upstream and downstream of the insertion in the different genotypes. Only 5 and 8 CpGs were used upstream and downstream of the insertion, respectively, because the region is CpG-poor. We show that the region is highly methylated and that the insertion of *Dac1J* does not seem to affect this DNA methylation pattern. This demonstrates that only the *Dac1J* 5’LTR but not the local epigenetic environment is affected (see below and Figure 1d and g). We have also added a track of ATAC, H3K27me3, H3K4me3, CTCF-ChIP, and RNA sequencing from wild-type forelimbs at E10.5 (Extended data 1f). This shows that neither the *Dac1J* insertion nor our deletion seems to disrupt any important element.

2. In figure 2d and 2e, only expression *Fgf8* is shown. What about other genes that are proximal to *Dac1J*? It would be helpful to show expression profiles of nearby genes, especially those that are topologically associated. We have slightly re-organized the paragraphs (lines 125-128 and 152-159) and extended data figures regarding the local gene expression. First, we now describe the expression of all genes in the *Lbx1* and *Fgf8* TADs (*Lbx1*, *Btrc*, *Poll*, *Dpcd*, *Fbxw4*, and *Fgf8*) measured by bulk (early E11.5) from forelimb. None of the genes at the locus are affected except for *Fgf8* (bulk RNA-seq from 3 biological replicates; see **Extended data Figure 4a and below**). Yet, *Fgf8* expression by whole mount in situ hybridization does not seem to be affected at E10.5 (**Extended data Figure 4b**).

Second, we use single-cell RNA-seq (forelimbs, E9.5 to E11.5) to decipher how the transcripts are affected over time. We show the dynamics of *Fgf8* expression (**Figure 2d, Extended data Figure 3c and 4a**) and correlate it with changes in the expression of other AER genes (**Extended data Figure 4b**). Finally, we now show the expression of the five other genes at the locus (*Lbx1*, *Btrc*, *Poll*, *Dpcd*, and *Fbxw4*) using our single-cell data (**Extended data Figure 4c and below**). We thought that those data fit better in the Extended data rather than in the main figure as they mostly show no change of expression. We believe that the most important message here is that the AER cells, rather than local gene expression, are affected.

3. In figure 3h, does the minimal β -globin promoter also introduce ectopic TAD similar to that of MusD 5' LTR? If not, does that mean that adaptation of nearby enhancer activity by 5' LTR is independent of chromatin topology? This is a relevant question but we have not performed any capture-HiC to measure a potential ectopic chromatin loop in the β -globin-LacZ embryos as this is a very time-consuming and costly experiment and we think it will not add to our main conclusions. Following reviewer 4's comment, we have now performed the CTCF ChIP-sequencing from limb buds at E11.5 (same stage as for the capture-HiC data). At this stage, we did not recapitulate the CTCF peak that we observed at E10.5 at the position where the 5LTR is inserted. This suggests that CTCF binding plays a minor role in the formation of the ectopic loop which is likely driven by transcription, similar to the mechanism shown by Zhang *et al.* with HERVH (<https://doi.org/10.1038/s41588-019-0479-7>). We can thus hypothesize that the minimal β -globin promoter might also introduce an ectopic chromatin loop as for the MusD 5'LTR and that the adoption of the nearby enhancers is independent of the chromatin topology but rather a property of the MusD 5'LTR.

4. In figure 5d and line 281, could the other 9 complete MusD copies be involved instead of ETNs? Furthermore, are there other MusD copies that potentially express pol? It would be helpful to show the expression (or non-expression, if that is the case) of these copies. This is very relevant as it may explain the partially rescued phenotype and strengthen the claim that replication-competent VLPs are responsible for the etiology. This is an interesting comment and we indeed did not consider that other active MusD in the mouse genome could participate in the phenotype. In our scRNA-seq data, we used the *Dac1J* sequence added to the *mm10* transcriptome to filter the reads that map 100% to the *Dac1J* (unique mapping). Some of the other MusD in the genome being highly similar in sequence, we would have expected to see expression in WT cells, which we did not observe. This suggests that other MusD elements with similar sequences to MusD are not expressed. However, the number of cells we sequenced and the repetitive nature of these elements do not allow us to rule out that other MusD are lowly expressed. We have now discussed this possibility in the discussion, lines 411-412. Yet, in the *Dac1J* mut-PBS knock-in, the animals show a similar 20% rescue of the phenotype as in the *Dac1J* mut-pol despite no reverse transcription (Figure 5). We thus believe that if other MusD pol genes play a role in the phenotype, it is only a minor one.

5. Main text line 286-288: "Altogether, the partial rescues of MusD-Dac1J element with disrupted Pol protein or tRNA binding site suggest that full Dac1J-derived MusD VLPs with replication potential are required to cause Dactylaplasia phenotype." – this is not entirely correct. If Dac1J is solely responsible for the RT activity of MusD as concerned above, then the result does not suggest that intact VLPs are required for the dactylaplasia phenotype. Instead, the result suggests that reverse transcription of MusD accounts for most of the dactylaplasia phenotype, while GAG solely, although to a much lesser extent, still causes missing digits. I feel strongly that the distinction between VLP formation and reverse transcription needs to be addressed more carefully in this context, as study on HIV has shown that VLPs can form effectively with 80% truncated gag (Accola *et al.*, J Virol. 2000). We agree that our conclusions were not entirely precise. By "*MusD VLPs with replication potential*" we meant MusD VLPs that can undergo reverse transcription. Yet, both the *Dac1J* mut-pol and mut-PBS show ~20% of the animals with a phenotype (although milder than what was observed in *Dac1J/Dac1J* where most of the affected

animals display a one-digit phenotype). Our results thus indeed suggest reverse transcription of the *Dac1J*-MusD (solely the *Dac1J* element as the *Dac1J* mut-PBS mutant did not inhibit MusD RT but only the *Dac1J*RT) account for most of the phenotype but that the Gag capsid or RNA alone can drive a milder phenotype. This is now discussed in the manuscript lines 412-418, given our new results of the Gag mutant KI (see **response to comments 6.**).

6. Main text line 315-318: “Overall, our results indicate that the cell death response to MusD VLPs during development can affect embryonic cells differently and highlight that mouse post-implantation development can also proceed with the presence of endogenous retrovirus capsid.” – same as above, the manuscript does not present evidence supporting that VLPs are responsible for cell death. The reviewer feels strongly that a knock-in experiment with gag mutated *Dac1J* is needed to make such statement.

We have now generated two versions of a *Dac1J* knock-in with a mutated *gag*: one with a complete deletion of the *gag* gene (mut-gag, see **Extended data Figure 9a and below**) and one with a deletion of the capsid + zinc finger domains of the Gag protein but where the matrix is still present (mut-gag(CA), see **Extended data Figure 9a and below**).

For the mut-gag(CA) mutant, polyclonal MusD-Gag antibody can still detect Gag expression in the embryos (**Figure 5c and Extended data Figure 9b and below**). However, in both cases, the phenotype was fully rescued as none of the observed E18.5 embryos show a phenotype (**Figure 5d and below**). This shows that the Gag capsid of the *Dac1J* element is required for the phenotype. Yet we showed with our *Dac1J* mut-pol and mut-PBS that Gag alone is not sufficient to lead to a complete phenotype (**Figure 5d and below**). We believe this new result suggests that both Gag capsid and reverse transcription play a major role in the phenotype. We indeed cannot exclude that inefficient reverse transcription happening outside of the *Dac1J* Gag capsid is also the cause of the complete rescue. We discussed that lines 409-418.

7. General: the methylation status between 129 and B6 is very interesting and perhaps outside the scope of the present work. It would at least be helpful to speculate on what might be causing such a drastic discrepancy between strains in the discussion and whether something akin to the IAP element within the agouti locus (differential methylation pattern/metastable epiallele) could possibly be at play. Is it possible “stress” or some “environmental” biosensor is triggered?

When measuring DNA methylation of the *Dac1J* 5’LTR, we did not see variable levels but rather full hypermethylation or complete hypomethylation. From the study of *Kano et al.* (2007), the modifier region responsible for the differential DNA methylation between mouse strains has been mapped to a large locus on chromosome 13, containing a cluster of KZFP. We have added a sentence lines 104-105 to clarify that part: “Moreover, the phenotype was shown to be dependent on the presence of an epigenetic modifier (*Mdac*) which is polymorphic between mouse strains and has been refined to a 9.4-Mb region on mouse chromosome 13”. It is believed that this modifier is a KZFP, which is missing in the 129sv mouse strain. However, this still needs to be demonstrated, and it is the topic of a separate and ongoing manuscript. We would, therefore, prefer not to elaborate

too much in our current manuscript which is rather focused on the mechanism of MusD-driven developmental malformation.

Minor comments:

1. In figure 1a, what is the unlabeled region just upstream of the 3' LTR? Is it supposed to be Env? If so, it might be helpful to indicate that this is mutated or missing.

The unlabeled region is not the *env* gene. MusD elements have lost the *env* gene (see Mager and Freeman 2000, doi: 10.1128/jvi.74.16.7221-7229.2000.) as we wrote line 98. This 3'UTR region contains several truncated parts of MusD/ETn elements. We have now labeled this region with the Repeatmasker names in **Figure 1a** (see below and figure legend).

2. In line 103-104, the manuscript mentions “However, Dactylaplasia is subject to polymorphisms between mouse strains (Fig. 1c) as the MusD-*Dac1J* 5'LTR promoter exhibits differential DNA methylation in the limb21 (Fig. 1d).” For better readability, it would be helpful to expand the background a bit more by briefly introduce that methylation of *Dac1J* is dependent on the presence of M_{Dac} (Kano et al., Proc. Natl. Acad. Sci.).

We have now added a sentence discussing the modifier of *Dac* located on chromosome 13 and refer to *Kano et al.* (104-105).

3. In figure 1b, what strain is “wildtype” specifically?

Both C56BL/6 and 129sv wild-type display normal 5-digit phenotype, skeletal prep have been done with both lines. Regarding the CpG DNA methylation, we now show the level of methylation upstream and downstream the insertion in both mouse strains.

4. In figure 1d, would it be possible to investigate the methylation status around the area where the MusD insertion would have been in the WT? Perhaps this could be a clue about the local epigenetic environment “pre-insertion”. Perhaps this can be done for both B6 and 129 just to see whether there was something different/unique about this location in these strains.

This has been done as detailed in major comment 1.

5. In figure 2a, when printed, the details of the limb bud are difficult to resolve. I think enlarging just this side of the image would help.

Done.

6. In figure 2b, similarly, I completely missed the “immune cells” population on my first look of this image. A little rearranging and enlarging will be helpful.

Done.

7. In line 122, AER has already been defined. No need to have full name here.

8. In figure 2c, text on the right is very hard to see. Please consider larger font. Keep consistent with the rest of the panel (same font and size).

Ok, corrected.

9. In figure 2c and line 121-133, how would these populations change if you were to compare WT to another WT? What is the individual variance among different embryos? (Replicates?) I am not familiar with the nuances of single cell analysis and just wonder whether a replicate would demonstrate a similar change. Alternatively, is it possible to present the data without the Mesenchyme, as this signal dominates the profile, making the changes you are interested in small and hard to visualize/interpret.

We do not have any replicates for this study but we have several WT limbs scRNA-seq data available from previous independent studies from our lab (<https://doi.org/10.1038/s41586-021-03208-9> and <https://doi.org/10.1038/s41467-023-37057-z>) where the proportion of AER cells in WT was never observed below 0.6% of total cells. There is probably variability between samples, even between WT, but the extremely low number of AER cells present at E11.5 in *Dac1J/Dac1J* (**Figure 2c** and **Extended data Figure 3b**) mutants seems

to be specific. As suggested, we now show both genotypes at each of the three embryonic stages without the mesenchyme cells (**Figure 2c and below**).

10. In line 139, was “E10.25” intentional?
Corrected.

11. In line 144, I was under the impression that mapping to a specific TE using single cell RNA-seq was difficult, but given the age of the insertion and about 100 nearly identical copies, is unique mapping even possible? Is this at the subfamily level?

In our scRNA-seq data, we used the *Dac1J* sequence added to the *mm10* transcriptome to filter the reads that map 100% to the *Dac1J* (unique mapping). We might get some reads that are coming from other very similar MusD in the genome but as no expression is detected in the wild-type, we can be confident that the detected expression is specific to MusD-*Dac1J*.

12. In line 156, minor type? I think the word “and” was meant to mean something else. “are”?
Corrected.

13. In figure 3a, “Fgf8 enhancers” text is way too small.
Corrected. Size is now the same as the gene names.

14. In line 168, can you expand a little bit on how increased insulation is determined and what this might impact? I think indicating how this is determined on figure 3e and 3f would help people who are not too familiar with these types of graphs/outputs.

Those figures show the capture-hic subtraction signal between mutant and wild-type. Red and blue signals mean more and less interaction in the mutant than in the wild-type, respectively. We have now clarified that in the text, lines 184-186, and added the text “subtraction map” in the figures. We also try to make it clear in the figure legend, and **Figure 3** is now reshuffled so that the subtraction map appears below the related capture-HiC map.

15. In figure 3e, *Dac1J* locus shows increased interaction asymmetrically spanning over *Lbx1* and *Fgf8* comparing with wildtype, while the same cannot be observed with 5’ LTR-LacZ. Do the authors have an interpretation for this? Personally, I think it is interesting and worth mentioning briefly in the manuscript.

We think that these contact lines correspond to what has been called “architectural stripes”. Architectural stripes are a frequent feature of developmental three-dimensional genome architecture often associated with active enhancers (*Kraft et al.* 2019. DOI: 10.1038/s41556-019-0273-x). The fact that this stripe is not observed with the 5’LTR-LacZ could reflect less interaction with the surrounding enhancers, less CTCF binding, or could exist but would be detected only with deeper sequencing as this stripe is rather subtle in the *Dac1J/Dac1J* mutant. We now mention it in the text (lines 199-200).

16. Figure 4d legend, shouldn’t it be “Zoom-in view (right)”?
Corrected.

17. In figure 5a, would it be better to title this something like “MusD endogenous retroviral cycle” to distinguish this from a current “viral” cycle of an exogenous retrovirus? Also, the font for LTR, gag, pro, etc... are quite hard to see in the current version of the figure. It might be a quality issue when the figures were submitted but there is an overall drop in quality of the various texts (some letters are thinner or seem to be missing “ink”, if that makes sense).

Corrected.

18. In figure 6, do *Dac1J* KI/KI @ *Lbx1*-TAD embryos develop to full term? Do they show any phenotype?

We have not generated a mouse line for this mutant (embryo generation via tetraploid morula aggregation of the mESC's clone) but we looked at embryos at E18.5 (from 2 different experiments) and never saw any obvious phenotype. We of course cannot rule out that there is a subtle muscle-related phenotype. We have added a sentence about that in the text (line 355).

19. In extended data figure 6, typo in title "Extended".

Corrected.

20. In discussion, is there any link between this phenotype and the human condition, or would this be specifically associated with the 129 mouse line?

The human condition is more similar to the phenotype of the *Dac1J* heterozygotes mice, characterized by "split" digits (see our previous publication <https://doi.org/10.1038/s41467-023-37057-z>). None of the SHFM3 patients have such a strong phenotype as the homozygotes *Dac1J* mice. The human condition is known to be caused by duplication or inversion at the *Lbx1-Fgf8* locus which we have shown to lead to ectopic expression of *Lbx1* and *Btrc* in the AER. In mice, this never led to cell death of the AER and we could never observe a phenotype as strong as the *Dac1J* mice (even in the heterozygote). We only observed a mild and non-penetrant split digit phenotype in the mice with the patient's inversion. We cannot exclude that in human patients, the *Btrc* and *Lbx1* ectopic expression leads to a certain level of cell death as the MusD does. However, it seems that the pathomechanism of the SHFM3 patients is not linked to retrotransposon insertion. Other developmental malformations in human, including some affecting the digits are still to be understood and our study hints that retrotransposons could have a role in their patho-mechanism.

Reviewer #3:

Remarks to the Author:

In this manuscript, Glaser et al. characterize genetic determinants of limb malformation in a dactylaplasia mouse model and highlight the importance of transposon control during development. The authors show that in the 129sv mouse strain with the dactylaplasia phenotype (*Dac1J*), a MusD ERV element inserted at the *Fbxw4-Fgf8* locus escapes DNA methylation and is activated in the apical ectodermal ridge (AER) cells via hijacking local cis-regulatory elements. The MusD activity does not appear to affect *Fgf8* expression but instead leads to apoptosis of AER cells, which authors suggest is due to viral-like particles (VLPs) produced by the MusD. Consequently, deletion of the entire MusD element, mutation of its coding sequence or of the tRNA primer-binding site, all result in rescue of the limb phenotype.

Overall, this is a well-executed study, with results that are of interest to the developmental biology and transposable element fields. The results are clearly presented in figures that make understanding of the manuscript straightforward. The mouse genetics approach to the dissection of the MusD element viral cycle is very elegant, and clearly implicates ERV proviral sequences in mediating the limb phenotypes. I think this latter point is the most novel aspect of the work (rather than adoption of local enhancers by the TE promoters, which is not surprising, and in fact a basis of many sensors of enhancer activity).

While I am generally positive about the study and its suitability for Nature Genetics readership, one quite unsatisfying point in the manuscript is its lack of insight into the mechanism behind the questions of why the MusD-expressing AER cells are undergoing apoptosis and why is the MusD activity temporarily decoupled from apoptosis (seems to occur at least a couple of days prior to the cell death). Given that loss of AER cell population due to MusD activity is the cornerstone of the manuscript, some insight into these questions would greatly strengthen the study. For example, at which point (if at all) following MusD expression do authors see DNA damage or innate immunity activated in AER? Is it restricted to AER cells within the developing limb? To what extent can the kinetics of the viral life cycle explain the observed delay? Some simple stainings may provide interesting, even if correlative, clues as to the underlying mechanism.

We thank the reviewer for the constructive feedback on our manuscript. We agree that answering those questions will improve our study and we made some additions to the manuscript that we hope will provide the answers:

1) we performed anti-gammaH2AX immuno-fluorescence at E10.5 and E11.0 in wild-type and mutant to detect DNA damage (see **Figure 4f-g, Extended data Figure 7b and below**). A few AER cells in wild-type conditions do show DNA damage but the intensity and number of cells in *Dac1J/DacJ* mutant is clearly higher at both stages. This suggests that DNA damage precedes cell death as we did not observe such a level of cleaved-caspase3 at E10.5 where the cells seem rather healthy. This also shows that the toxic effects of the VLPs are already present at E10.5. We summarize the cellular event as follows: *a) Dac1J*-MusD is co-expressed with *Fgf8*

from E9.5, *b*) by E10.5 we can observe VLPs and DNA damage in the AER cells, *c*) at E11.0, the AER cells undergo apoptosis and *d*) at E11.5 most of the AER cells are eliminated (see **scheme Figure 4h and below**).

2) We performed immuno-staining to detect a potential immune response (anti-F4/80 as a murine macrophage marker, anti-S9.6 to detect immunogenic RNA:DNA hybrid and different immune cell markers) but were not able to detect a signal. As this could also be technical, we looked at the expression of many interferon-stimulated genes from bulk RNA-sequencing at E11.0 but most of the gene expression were unchanged (see **Extended Figure 8c**). Importantly, because MusD elements have lost the envelop (*env*) gene, they cannot assemble into infectious particles. Endogenous retrovirus (ERV) activation has been linked to the activation of immune response in different contexts (for example <https://doi.org/10.1016/j.cell.2021.05.020> and <https://doi.org/10.1038/s41467-023-43728-8>) but always (to our knowledge) implicating ERVs with an *env* gene. From our current sets of data, it thus seems that no immune response is involved in the apoptotic process in response to MusD VLPs assembly in the AER of the *Dac1J/Dac1J* mice.

3) We performed fluorescent mRNA in situ staining (HCRTM RNA-FISH Technology) using an *Fgf8* probe at four carefully staged (using somites count) embryonic stages: E10.0, E10.5, E11.0, and E11.5 to clarify how the AER cells and *Fgf8* expression evolves (see **Extended data Figure 7c-d and below**). We can observe proper *Fgf8* activation and shape of the AER at E10.0 and E10.5 (although *Fgf8* might be a bit lower at E10.5, which could be correlated with our observation of DNA damage). At E11.0, *Fgf8* expression persists but the cells are unhealthy before being severely affected at E11.5. To reconcile why the AER cells seem to not be affected before E10.5, we speculate that this might be linked to the dose of MusD-*Dac1J* expression/ VLPs. *Fgf8* was shown to be most highly expressed at E10.5 (DOI: 10.1002/dvdy.394); perhaps MusD VLPs follow this dynamic and start to be damaging for the cells at that time point.

In addition, there are several other points that deserve further study and clarification.

Major comments:

-The authors focus on *Fgf8*, but there is another gene closer to the *MusD* element, named *Fbxw4*, for which the impact of the *MusD* (*Dac1J*) presence has not been addressed, but which has been previously implicated in dactylaplasia-like phenotypes. Please confirm that expression of *Fbxw4* is not affected by *MusD* activity.

We now show the measurement of *Fbxw4* expression using bulk at early E11.5 (see Extended data Figure 2a) and single-cell RNA-sequencing at E9.5, E10.5 and E11.5 (see below and Extended data Figure 3c, violin plots and Extended data Figure 4c, dot plot). The decrease that we see in the AER at E10.5 is observed for other genes (see Extended data Figure 4) and we believe this represents AER cells starting to be unhealthy because of DNA damage.

Thus, *Fbxw4* activation does not seem to be affected by the *Dac1J* insertion. This gene was believed to be implicated in the Dactylaplasia phenotype when studying an independent *MusD* insertion causing Dactylaplasia, *Dac2J*, which is located in an intron of *Fbxw4* (DOI: [10.1038/12709](https://doi.org/10.1038/12709)).

-Figure 1E: Why did the authors delete ~8 extra kb (3.3 upstream and 5.3 kb downstream) of the MusD element? Are these extra deleted DNA base pairs affecting the phenotype? Please show as a supplementary figure the epigenomic landscape (e.g. ATAC-seq, histone modifications) of the deleted genomic region to visualize if any cis-regulatory elements have been inadvertently deleted. WT data should suffice.

We initially generated a bigger deletion as sgRNAs further away had a better score than the ones closer to the *Dac1J* insertion but we have now successfully generated a new deletion, which is ~8.3kb with only 446 and 101bp upstream and downstream of the *Dac1J* insertion (see **Figure 1e and below**). This deletion still rescues the phenotype.

e Δ *Dac1J* in *Dac1J/Dac1J* (129s2/Sv)

We have also added a track of available ATAC, H3K27me3, H3K4me3, CTCF-ChIP, and RNA sequencing from forelimbs at E10.5 (see **Extended data 1f and below**). This shows that neither the *Dac1J* insertion nor our deletion seems to disrupt any important element.

f chr19:45,570,000-45,750,000

-Which markers were used to call the different cell clusters in Figure 2B? Please provide more details, especially for the AER cells. Which markers and specific expression levels are used as cut-off for considering a cell AER at the different stages?

We have added a table (**Extended data Table 1**) showing all marker genes for all 7 cell clusters from our single-cell RNAseq data. **Extended data Figure 3a** (see **below**) also shows violin plots for two marker genes per cluster.

- The authors should exclude a possibility that the depletion of the AER population at E11.5 is the consequence of a consistently lower *Fgf8* expression at E9.5 and E10.5. For example, in Figure 2D *Fgf8* expression appears lower in *Dac1j*^{+/+} than in WT, though it could be because some cells have already been lost. Please show *Fgf8* (and *Fbxw4*) expression in MusD-positive cells using Featureplot and violin plots. This could help visualize the results in a more quantitative manner.

We indeed believe that this lower expression of *Fgf8* in the AER at E10.5 results from some cells being already unhealthy. Indeed, we show in **Figure 4f** that DNA damage increases in the AER of the *Dac1J/Dac1J* mutant

compared to wild-type. Moreover, the decrease seen for *Fgf8* in **Figure 2d** can also be observed for other AER genes, as shown in **Extended data Figure 4b** (and below).

We have also added violin plots for *Fgf8* and *Fbxw4* in **Extended data Figure 3c** and feature plots showing the AER and dorso-ventral ectoderm cells (the only cluster where *Fgf8* is expressed) appear next to the dot plots in **Figure 2d**. Finally, we think we can exclude the possibility that the depletion of the AER population at E11.5 is the consequence of a consistently lower *Fgf8* expression at E9.5 and E10.5 as heterozygote mutants for *Fgf8* (i.e. only 50% of *Fgf8* expression in the AER) were shown to not have any phenotype (see <https://doi.org/10.1038/82609> and <https://doi.org/10.1038/82601>).

-How confident are the authors that the transcription in Fig. 2E and F is mostly (or solely) due to activity of the MusD element at the *Fgf8* locus? Is there evidence that other MusD elements are also active in these cells? Transposon mapping at the single insertion level is challenging with the 10x data.

In our scRNA-seq data, we used the *Dac1J* sequence added to the *mm10* transcriptome to filter the reads that map 100% to *Dac1J* (unique mapping). Some of the other MusD in the genome being highly similar in sequence, we would have expected to see expression in WT cells, which did not happen (see **Figure 2e**). This suggests that other MusD elements with similar sequences to MusD are not expressed. However, the number of cells we sequenced and the repetitive nature of these elements do not allow us to rule out that other MusD are lowly expressed. We have now discussed this possibility in lines 411-412.

-In Extended Data Figure 3 the authors perform bulk RNA-seq concluding that no other gene at the locus is affected. Showing browser captures is not enough to claim this. The authors should perform formal quantitative differential gene expression analysis to explore potential gene expression changes driving AER loss within and outside the TAD.

We now show a histogram representing rpkm expression of the 6 genes at the locus from bulk RNA-seq data from forelimbs at early E11.5 with 3 replicates. None of the 5 genes except *Fgf8* is affected by the VLPs at that stage (see **Extended data Figure 2a** and below). In **Extended data Figure 7d**, we show a heatmap from the same data with many genes important for limb development and indicate with an asterisk the ones that are significantly differentially expressed, not including the 5 genes at the locus (see below).

a Early E11.5 forelimbs, bulk RNA-sequencing

Minor comments:

-Line 76: Current reference 18 does not show expression at E3.5. Please add this reference too PMID: 2580305.
Thanks, done.

-Figure legend 1D, typo: promotor should be promoter.
Corrected.

-In Figure 3C and D, the authors show CTCF ChIP-seq. What is the numerical scale of the zoom-ins? It seems that WT and *Dac1j*^{+/+} are displayed at different scales favoring a larger CTCF peak at the *MusD* element. **Extended data Figure 5d** now shows these data (with other CTCF Chip-seq in response to one of reviewer 4's comment) with the same scale (0-3) for both the zoom-in and non-zoom-in tracks and it is indicated.

-*MusD* VLPs do not bud off the cells due to lack of envelope. Please indicate it somewhere in the text, as some readers may expect these particles to act in trans.
Ok, added lines 242-243.

-The authors did not provide a token for the reviewers to assess the quality of the sequencing data deposited at GEO. Apologies for forgetting that. The token is ankxyueohvutnax.

Reviewer #4:

Remarks to the Author:

Glaser et al investigated the mechanism by which a ERV insertion (*MusD-Dac1J*) near *Fgf8* gene causes Dactylaplasia, a genetic limb malformation. The study rejected the hypothesis that *Dac1J* disrupts *Fgf8* expression; rather, the authors discovered a novel mechanism where ERVs have the capability to take on the surround epigenomic landscape and express viral-like particles, which lead to cell death. The authors systematically show the role of *DAC1J* in limb malformation through the co-expression of *Fgf8* with *DAC1J* and the changes in TAD structure in mutant despite observing no transcriptional changes associated with this change. Since *Dac1J* has the machinery for autonomous proviruses they investigate viral-like particles (VLPs) and the potential implications. They found VLPs were associated with cell death and using mutated *DAC1J* they identified POL protein as a key driver in digit phenotype. They generalized their finding to show that *MusD* insertions can adapt the expression pattern of nearby genes using LTR-LacZ constructs.

Overall, reading this manuscript has been a treat for me. The authors presented an excellent scientific design and addressed many possible alternatives thoroughly. The authors not only made a novel discovery, but also defined a rigorous paradigm for testing hypothesis using genetics and genomics approaches. I have a few suggestions to hopefully help enhance the rigor of the science, which is already quite strong.

We thank the reviewer for the positive feedback on our manuscript and the mindful evaluation of our work.

1. Because this study investigates a young transposable element that has multiple copies in the genome and is polymorphic, any experiment that requires sequence specificity (CRISPR guide RNA design, bisulfite sequencing primer design, RNA-seq read mapping, etc.) needs to be carefully designed and scrutinized. The authors should provide a clear description of these designs, including analysis of off-target effect. How do they ensure that they are measuring/targeting the exact *Dac1J* copy that they intend to measure/target?

This is indeed a relevant point to consider. We have detailed below our strategy for the different techniques:

- **CRISPR guide:** all sgRNAs used in this study are available in **Extended Table 3**. They were carefully designed using Benchling and chosen to have as few off-target as possible. Except for one sgRNA, none are located in repeated regions. The only sgRNA located in which we could not avoid having off-targets is the "Deletion *Dac1J-Pol 5*" (see **Extended Table 3**, **Extended data Figure 9a** and **below**). We chose the sgRNA with the best off-target score within the *pol* region. We have now added in the methods a description of the design (lines 636-637) and a table of the off-targets (**Extended Table 4**). Yet, this sgRNA was used for deletion together with a sgRNA within a non-repetitive region. We also used two independent mESC clones for aggregation and phenotyping and found similar phenotypes with both clones. Finally, the phenotype found in this *Delta-pol* deletion was validated with our *Dac1J*-mutPol KI, which uses a different method (knock-in and no deletion), without any sgRNA in repeated regions.

- **Bisulfite-sequencing:** for both cloning-sequencing and pyrosequencing, we used one of the two primers located on the genomic, non-repetitive region. For *Dac1J* 5'LTR, the PCR was done as follows (see **schematic below**): For cloning-sequencing: nested PCRs with outer F+R and then from this PCR, inner F+R. For pyrosequencing: PCR with outer F+ Pyroseq_R(Btn) and then Pyroseq sequencing used for the sequencing in the pyrosequencer. The amplified regions are indicated in red in **Figure 1b** and in **Extended data Figure 5c**.

- **RNA-seq read mapping:** For bulk RNA-seq, we did not consider repeated regions as we only looked at genes. For scRNA-seq data, we used the *Dac1J* sequence added to the *mm10* transcriptome to filter the reads that map 100% to *Dac1J* (unique mapping). Some of the other MusD in the genome being highly similar in sequence, we would have expected to see expression in WT cells, which did not happen (see **Figure 2e**). This suggests that other MusD elements with similar sequences to MusD are not expressed. However, the number of cells we sequenced and the repetitive nature of these elements do not allow us to rule out that other MusD are lowly expressed. As also suggested by other reviewers, we have now discussed this possibility in lines 411-412

2. The differential methylation of *Dac1J* in C57BL/6 and 129sv is a key evidence for the claim that unmethylated *Dac1J* is required for the phenotype. However, to establish a real causal relationship, DNA methylation could be manipulated directly in C57BL/6 and 129sv to rule out other factors associated with their genetic background. Later in Fig 5b, the methylation status of the mutants should be provided. The assumption is that they are not methylated, but it should be confirmed.

The observation that the phenotype depends on mouse strains and is related to differential DNA methylation was already known (*Kano et al. 2007*). It has also been shown that other mouse strains than 129sv (for example BALBc and SM7B/SC) show a phenotype when having the *Dac1J* insertion when the epigenetic modifier *Mdac* is not present. We have added the sentence to illustrate that “*Moreover, the phenotype was shown to be dependent on the presence of an epigenetic modifier (Mdac) which is polymorphic between mouse strains and has been refined to a 9.4-Mb region on mouse chromosome 13*” lines 104-105. *Mdac* is unknown and this is outside of the scope of this manuscript and is currently under investigation for an independent story. For the knock-in mutants in **Figure 5b**, we have added a graph in **Extended data Figure 9b** (see **below**) showing the methylation status (measured by pyrosequencing, see methods) of several animals with or without a phenotype that we included in the phenotype analysis. This shows that the CpG DNA methylation status of the 5'LTR of the *Dac1J* knock-in is hypomethylated as for the *Dac1J/Dac1J* in the 129sv background and that is independent of the phenotype.

d CpG DNA methylation

3. To generalize the finding, the authors inserted the 5'LTR-LacZ construct into the *Lbx1*, *Shh*, and *Sox9*

regulatory domains and showed that the LacZ expression recapitulated those genes' expression. It is important to measure 3D genome changes in these contexts, in order to generalize the principle. Does the 5'LTR-LacZ insertion also form an ectopic chromatin loop like the 5'LTR-LacZ insertion upstream of *Fgf8* gene?

Our results at the *Fgf8* locus argue that the ectopic loop formed by the insertion of the *Dac1J* is likely not causing the observed expression in the AER. We apologize that this was unclear in the previous version. We have now performed new CTCF ChIP-sequencing (as asked and detailed in **comment 7.**), and we conclude that CTCF is not the main driver of the ectopic loop and that transcription is likely to be (as shown for a human ERV in <https://doi.org/10.1038/s41588-019-0479-7>). We also performed CTCF ChIP-seq in the 5LTR-LacZ KI at the *Fgf8* locus and did not see a clear CTCF peak. Thus, the ectopic contact we observed appears to be a consequence of transcriptional and/or enhancer activity, which appears not to play a role in the phenotype. Instead, the adoption of the surrounding regulatory elements by the MusD-*Dac1J* promotor, (see the experiment with the beta-globin-LacZ, **Figure 2g**) is driving the *Dac1J*-MusD expression.

4. Considering the limit number of cells be captured and presented using scRNA-seq and its relatively large variations in detecting cell numbers, it would be better if the authors can estimate the number of AER cells using another approach such as FACS or immunostaining. This could provide more direct and quantitative evidence on AER cell apoptosis at different time points.

We agree that an approach such as FACS will be ideal. However, to our knowledge, there is no good marker for sorting specifically AER cells using FACS. This would require generating a new mouse line with an AER-reporter which then needs to be crossed with our *Dac1J* mouse line. To avoid this major effort, we opted for a less quantitative but qualitative method: we used fluorescent mRNA in situ staining (HCR™ RNA-FISH Technology) using an *Fgf8* probe (as a marker of AER cells) at four carefully staged (using somites count) embryonic stages: E10.0, E10.5, E11.0, and E11.5 to clarify how the AER cells and *Fgf8* expression evolves (see **Extended data Figure 7c-d and below**). We observed proper *Fgf8* activation and shape of the AER at E10.0 and E10.5 (although *Fgf8* might be a bit lower at E10.5, which could be correlated with our new observation of DNA damage, see **Figure 4f**). At E11.0, *Fgf8* expression persists but the cells are unhealthy before being severely affected at E11.5. These new results are discussed in lines 258-262.

5. According to scRNA-seq data, AER cells only present a very small population at E10.5 and E11.5, which indicate that most of the CHIP-seq and cHi-C signal are captured from other cell types (i.e., mesenchymal cells), which means that MusD-*Dac1J* are likely to be epigenetically activated in those cells. However there is no strong expression of MusD-*Dac1J* transcripts as well as production of VLP particles in other cell types. Are there any explanations for that?

The AER cells are indeed only a small population of the total developing limb bud (3.5-1% of the forelimb according to our single-cell data). The epigenetic status (i.e. CpG DNA methylation) of the MusD-*Dac1J* is the same in every cell of the organism (as shown by measuring CpG DNA methylation in tissues from different germ layers (**Figure 1d and Extended data Figure 1e**)). The expression of MusD-*Dac1J*, specifically in the AER cells is solely due to the activation by the surrounding regulatory elements. To confirm this, we inserted the MusD-*Dac1J* element in the neighboring regulatory domain (*Lbx1* TAD) and show that it is now expressed according to

the regulatory information in an *Lbx1*-like pattern. The presence of regulatory elements activating the *MusD-Dac1J* promoter explains its expression as well as the production of VLP specifically in the AER cells and no other cell types of the limb.

We think the capture HiC signal could originate only from the *MusD-Dac1J*-expressing cells. It is possible to get such a signal even when coming from less than 5% of the cells as we are looking at a gain of signal.

6. Extended data Fig. 2 showed a large cell number decrease for immune cells at E10.5 and cell number decrease for DV ectoderm at E11.5. Why cell number decreased in *Dac1J* +/-? Are those cells affected by the *Dac1J* insertion, although *Dac1J* did not express in those cells?

The mentioned figure (in the new version, **Extended data Figure 3b**) shows raw cell numbers. We think that to make such a statement, the percentage of cell type measurement is more accurate as the total number of cells also varies between the samples. The percentage of cells is shown in **Figure 2c** and the table used to generate this figure is shown **below**. Immune cells are indeed decreased by half in the *Dac1J/Dac1J* mutant compared to WT at E10.5 and E11.5. This could be technical variability existing in clusters with very few cells. Alternatively, this could have a biological explanation where the resident immune cells of the limb are affected by VLPs production and cell death. However, we did not show any marker of innate immunity or inflammatory response (see response to the first comment). Regarding the decrease in DV-ectoderm cells at E11.5 in the *Dac1J/Dac1J* mutant, we mentioned that in the main text, lines 138-139. *Fgf8* (and accordingly *Dac1J* is expressed in many of these cells (see **Figure 2d,e**) so this is expected that those cells are also affected by VLPs production.

Wild-type	E9.5	E10.5	E11.5	Dac1J/Dac1J	E9.5	E10.5	E11.5
Immune cell	0,1%	1,0%	0,8%	Immune cell	0,08%	0,5%	0,4%
Erythrocytes	1,7%	2,2%	1,4%	Erythrocytes	2,8%	1,6%	1,1%
Vascular cells	7,5%	2,3%	0,8%	Vascular cells	4,54%	3,4%	0,7%
AER	3,5%	1,2%	0,9%	AER	4,90%	1,1%	0,1%
DV ectoderm	7,2%	4,4%	3,7%	DV ectoderm	9,01%	5,7%	0,7%
Myocytes	3,4%	8,7%	4,2%	Myocytes	4,83%	6,0%	8,3%
Mesenchyme	76,6%	80,2%	88,2%	Mesenchyme	73,82%	81,8%	88,7%

7. Fig. 3a-b showed the Hi-C data from wild-type and *Dac1J* +/- E11.5 mouse limb buds. However, the Fig.3c-d used the ChIP-seq tracks from E10.5 mouse limb buds. Please used the ChIP-seq tracks from same stages as the Hi-C data to support the TADs.

We have now produced new ChIP-seq data at E11.5 for WT, *Dac1J* (C57BL/6), *Dac1J* (129sv/S2) and 5LTR-LacZ @*Fgf8*. These data do not show a clear ectopic CTCF binding site at the *MusD-Dac1J* insertion site at E11.5 (see **Extended data Figure 5d and below**). It thus suggests that if CTCF is involved in the formation of the ectopic loop, it plays only a minor role (see text lines 188-189)

Apart from CTCF, transcriptional activation was shown to drive ectopic loop and this was notably shown for HERV retrotransposons elements (see <https://doi.org/10.1038/s41588-019-0479-7>). To test whether transcription of the *MusD-Dac1J* can be responsible for the ectopic loop, we used Capture-HiC data we generated from a tissue where *Fgf8* and *MusD-Dac1J* are not expressed (heart). In this tissue, the *MusD-Dac1J* promoter is fully unmethylated (see **Extended data Figure 5e**) but *MusD-Dac1J* is not transcribed. We did not observe any ectopic loop formation in heart samples from *Dac1J/Dac1J* mutants (see **Figure 3e and below**). When subtracting the WT forelimb from the *Dac1J/Dac1J* heart capture HiC maps, we did not see any gain of contact (see **Figure 3f and below**). This suggests that transcriptional activation of the *MusD-Dac1J*, rather than solely CTCF binding,

leads to the observed ectopic loop. It also corroborates our conclusion that the observed ectopic loop does not have any role per se but is likely a by-product of transcriptional activation. We re-wrote the results part lines 187-195 accordingly to these data.

Dac1J/Dac1J, E11.5 hearts

8. To measure the methylation status of 5' LTR promoter from *Dac1J*, the authors measured 19 CpGs of 5' LTR in Fig. 1, 21 CpGs of 5' LTR in Extended Data Fig. 4 and 7 CpGs in Extended Data Fig. 8. Please make these consistent.

We measured CpG DNA methylation of the *Dac1J* 5'LTR promoter by two different techniques. Bisulfite cloning-sequencing (see methods) allows the measurement of single molecule DNA methylation and shows the level of each DNA fragment in a cell population (CpG represented with black and white lollipop as in **Figure 1d** and **Extended data Figure 1e, 5e**). The PCR-amplified product we sequenced is 390bp and measured all the 19 CpG present in the 5'LTR (see **Extended data Figure 9e** and **below**). The 21 previously indicated CpGs included CpG outside of the 5'LTR in the genomic region and we have now corrected that mistake. All CpG methylation measured with cloning-sequencing represents 19 CpGs. Pyrosequencing (see methods) measures the global level of each CpG in a population. We used this method to quantify a larger number of samples in **Extended data Figure 9d** which would have been too time-consuming and costly by cloning sequencing. Pyrosequencing requires a shorter PCR product. We thus measured only 7 CpGs (179bp PCR product) as shown in **Extended data Figure 9e** and **below**. Data in **Extended data Figure 9d** shows the average of 7 CpGs for each animal.

5'LTR

9. The proper name of the B6 mice is "C57BL/6", not "C57B16". Also, "wild type" is used rather leisurely throughout the manuscript. Specific genetic background should be provided.

Ok, changes have been made for C57BL/6 throughout the manuscript and specific background is indicated in figures or figure legends.

Reviewer #1:

Remarks to the Author:

In this revised version of the manuscript by Glaser et al., the authors have added a lot of new experiments including a careful analysis of the timing and order of the pathogenic events they observe within the mutant limb bud, requiring some very pretty HCR RNA fluorescent in situ experiments. There are also several new ChIP-seq experiments and the generation and analyses of a number of new mutant knock-in mouse lines.

All of this this data is compelling and greatly strengthens their elegant study.

My only point is that there is a missing the word, likely loop, in line 373.

Thank you for the positive feedback. This has now been corrected.

Reviewer #2:

Remarks to the Author:

We would like to commend Glaser et al. for addressing every point of our review and thoughtful consideration in their replies. Quite a few responses went above and beyond what we had requested and appreciate how they made this second review very clear and concise. The mutated gag knock-in is impeccable and conclusion is eloquently conveyed. Overall, I believe the revision is more than acceptable and we recommend this version of the manuscript for publication. I only have a very minor concern regarding the readout of *Fgf8* expression using bulk RNA-seq and scRNA-seq/WISH, what could be the possible cause of the observed discrepancy? I do not think this will undermine any of the conclusions the authors are trying to draw in the manuscript but believe it requires some clarification in the final version. This updated version is much more complete and the great efforts the authors made have elevated the importance and impact of this manuscript.

Thank you for the positive feedback. Below, a clarification regarding the expression of *Fgf8* using bulk RNA-seq, scRNA-seq and WISH:

Bulk RNAseq data on **Extended data Figure 2a** are from early E11.5 embryonic limbs. We observed that *Fgf8* expression is affected but that the expression of other genes at the locus is unchanged.

On **Extended data Figure 2b**, we used WISH to look at *Fgf8* expression from E10.5 and E11.5 embryos. At E10.5, we do not see any changes of expression but at E11.5, the expression pattern is different, showing less expression of *Fgf8*. That confirms the bulk RNA-seq data.

Finally, on **Figure 2d-e** and **Extended data Figure 3c and 4a**, we show scRNA-seq. This shows that expression of *Fgf8* is not affected at E9.5 and barely at E10.5 while there is almost no *Fgf8*-expressing cells at E11.5. This observed lack of *Fgf8*-expressing cells from the single-cell data explains the previous results from bulk and WISH: we detected a decrease of *Fgf8* expression at E11.5 because the *Fgf8*-expressing cells are gone. We, however, note in **Extended data Figure 4a** that the few *Fgf8*-expressing cells present at E11.5 are still expressing *Fgf8*. In sum, our three readouts for *Fgf8* expression corroborate and we can conclude on the timing of *Fgf8* expression (confirmed by the HCR staining on Extended data 7c-d):

- E9.5: not affected (shown by scRNAseq, Figure 2d)
- E10.5: not or very mildly affected (shown by scRNAseq, Figure 2d and WISH, Extended data Figure 2b)
- E11.5: affected (shown by bulk RNAseq and WISH, Extended data Figure 2a-b) because of a lack of *Fgf8*-expressing cells (shown by scRNAseq, Figure 2c and Extended data Figure 3c). Yet, the very few cells present at E11.5 still express *Fgf8* so that the proportion of expression is not affected (shown by scRNAseq, Extended data Figure 4a).

Reviewer #3:

Remarks to the Author:

I am satisfied with the authors' response to my comments and recommend publication. Minor point: Could authors show a more unbiased analysis of differentially expressed genes from bulk RNA-seq in addition to a heatmap of pre-selected genes in extended data Fig. 8?

Thank you for the positive feedback. We have now added the table displaying the rpkm expression of all genes from this bulk RNA-seq (see **Supplementary Table 2**).

Reviewer #4:

Remarks to the Author:

I thank the authors for addressing my comments. I don't have further critiques.

Thank you for the positive feedback.